# Slicing Unbalanced Optimal Transport

**Clément Bonet**[*]                                                         *clement.bonet@ensae.fr*
*CREST, ENSAE, IP Paris, Palaiseau, France*

**Kimia Nadjahi**[*]                                                         *kimia.nadjahi@ens.fr*
*CNRS, ENS, Paris, France*

**Thibault Séjourné**[*]                                                     *thibault.sejourne@epfl.ch*
*LTS4, EPFL, Lausanne, Switzerland*

**Kilian Fatras**                                                            *kilian.fatras@mila.quebec*
*Mila, McGill University, Montreal, Canada*

**Nicolas Courty**                                                           *nicolas.courty@irisa.fr*
*IRISA, Université Bretagne-Sud, Vannes, France*

**Reviewed on OpenReview:** *https://openreview.net/forum?id=AjJTg5MOr8*

## Abstract

Optimal transport (OT) is a powerful framework to compare probability measures, a fundamental task in many statistical and machine learning problems. Substantial advances have been made in designing OT variants which are either computationally and statistically more efficient or robust. Among them, sliced OT distances have been extensively used to mitigate optimal transport's cubic algorithmic complexity and curse of dimensionality. In parallel, unbalanced OT was designed to allow comparisons of more general positive measures, while being more robust to outliers. In this paper, we bridge the gap between those two concepts and develop a general framework for efficiently comparing positive measures. We notably formulate two different versions of sliced unbalanced OT, and study the associated topology and statistical properties. We then develop a GPU-friendly Frank-Wolfe like algorithm to compute the corresponding loss functions, and show that the resulting methodology is modular as it encompasses and extends prior related work. We finally conduct an empirical analysis of our loss functions and methodology on both synthetic and real datasets, to illustrate their computational efficiency, relevance and applicability to real-world scenarios including geophysical data.

## 1 Introduction

Many machine learning tasks involve aligning objects such as images, graphs, datasets or their representations after transformations. This is particularly relevant in transfer learning tasks like domain adaptation (Fatras et al., 2021) or multimodal machine learning (Baltrušaitis et al., 2018). These objects can be conveniently represented as positive measures, *i.e.*, a set of samples associated with non-negative weights. Aligning then consists in minimizing a distance or discrepancy between two measures. It is crucial to choose a meaningful discrepancy that has desirable statistical, robustness and computational properties. In particular, some settings require comparing arbitrary positive measures, *i.e.*, measures whose total mass can have an arbitrary value, as opposed to probability distributions whose total mass is equal to 1. In cell biology, for instance, measures are used to represent and compare gene expressions of cell populations, and the total mass corresponds to the population size (Schiebinger et al., 2019).

---

[*] Equal contribution

**(Unbalanced) Optimal Transport.** Optimal transport (OT) has been frequently chosen as a loss function to align objects. OT defines a distance between two positive measures $\alpha$ and $\beta$ of the same mass ($m(\alpha) = m(\beta)$) by moving the mass of $\alpha$ toward the mass of $\beta$ with the least possible effort. However, in some applications, the mass equality constraint is not satisfied, *i.e.*, $m(\alpha) \neq m(\beta)$. It can still be enforced by a re-normalization of the mass, which is potentially spurious and makes the problem less interpretable. This setting has motivated the development of a new OT framework, called *unbalanced OT* (UOT), that can naturally compare measures of different masses by softly relaxing the mass conservation constraints (Kondratyev et al., 2016; Liero et al., 2018; Chizat et al., 2018b). An appealing outcome of this new OT variant is its robustness to outliers which is achieved by discarding them before transporting $\alpha$ to $\beta$. UOT has been useful for many theoretical and practical applications, *e.g.*, theory of deep learning (Chizat & Bach, 2018; Rotskoff et al., 2019), biology (Schiebinger et al., 2019; Demetci et al., 2022) and domain adaptation (Fatras et al., 2021). We refer to (Séjourné et al., 2022a) for an extensive survey of UOT. Computing UOT requires to solve a linear program whose complexity is cubical in the number $n$ of samples ($\mathcal{O}(n^3 \log n)$) (Pele & Werman, 2009; Peyré et al., 2019). Besides, accurately estimating UOT distances through empirical distributions is challenging as they suffer from the curse of dimension (Dudley, 1969). A common workaround is to rely on variants with lower complexities and better statistical properties. Among the most popular, we can list entropic OT (Cuturi, 2013; Pham et al., 2020) or minibatch OT (Fatras et al., 2020; 2021). In this paper, we focus on developing sliced UOT approaches.

**Sliced Optimal Transport.** Sliced OT (SOT) defines an alternative metric by leveraging the closed-form solution of OT between univariate measures (Rabin et al., 2012; Bonneel et al., 2015). It averages the OT cost between projections of $(\alpha, \beta)$ on 1D subspaces of $\mathbb{R}^d$. For 1D data, the OT solution can be computed through a sort algorithm, leading to an appealing $\mathcal{O}(n \log(n))$ complexity (Peyré et al., 2019). Furthermore, it has been shown to lift useful topological and statistical properties of OT from 1-dimensional to multi-dimensional settings (Nadjahi et al., 2020b; Bayraktar & Guo, 2021; Goldfeld & Greenewald, 2021). It therefore helps to mitigate the curse of dimensionality making SOT-based algorithms theoretically grounded, statistically efficient, and practical to solve even on large-scale settings. These appealing properties motivated the development of several variants and generalizations, *e.g.*, by considering different types or distributions of projections (Kolouri et al., 2019; Deshpande et al., 2019; Nguyen et al., 2020; Ohana et al., 2023; Nguyen et al., 2023) or manifold data (Bonet et al., 2023a;b;c). Fast computations of partial OT (a particular case of UOT) between univariate measures (Bonneel & Coeurjolly, 2019; Bai et al., 2023) or more generally on trees (Sato et al., 2020; Le & Nguyen, 2021), have been developed, so that slicing partial OT benefits from these efficient implementations and allows to compare large unnormalized measures. However, while (sliced) partial OT allows to compare measures with different masses, it assumes that each input measure is discrete and supported on points that all share the same mass (typically 1). In contrast, the Gaussian-Hellinger-Kantorovich (GHK) distance (Liero et al., 2018), another popular formulation of UOT, allows to compare measures with different masses *and* supported on points with varying masses, and has not been studied jointly with slicing.

**Contributions.** In this paper, we present the first general framework combining UOT and slicing between arbitrary distributions. Our main contribution is the introduction of two novel sliced variants of UOT, called *Sliced UOT* (SUOT) and *Unbalanced Sliced OT* (USOT). SUOT and USOT are both defined as regularized OT problems which leverage one-dimensional projections, but differ on how they relax the mass preservation constraint: USOT essentially performs a global reweighting of the inputs measures $(\alpha, \beta)$, while SUOT reweights each projection of $(\alpha, \beta)$. We provide a theoretical analysis of SUOT and USOT, which reveals that they share topological properties with UOT while being statistically more efficient in high-dimensional regimes, thanks to the slicing operation. Additionally, we propose fast and GPU-friendly algorithms to compute SUOT and USOT, based on the (non-trivial) dual derivation of our SUOT and USOT losses and a Frank-Wolfe strategy (Séjourné et al., 2022b). Finally, we illustrate the efficiency of our framework on various experiments: we deploy SUOT and USOT to compare distributions on non-Euclidean hyperbolic manifolds, classify documents, transfer image colors and aggregate large-scale geophysical data, and discuss their advantages over existing approaches.

**Outline.** In Section 2, we provide background knowledge on unbalanced OT and sliced OT. In Section 3, we introduce our new loss functions (SUOT and USOT) and prove their metric, topological, statistical and

duality properties in wide generality. We then explain in Section 4 how to compute SUOT and USOT via a Frank-Wolfe-based approach. We finally analyze the performance of SUOT and USOT on different practical tasks in Section 5.

## 2 Background

In this section, we first state our notations. Then, we provide the necessary background on unbalanced optimal transport and sliced optimal transport.

### 2.1 Notations

In what follows, $\mathcal{M}_+(\mathbb{R}^d)$ denotes the set of all positive Radon measures of finite mass on $\mathbb{R}^d$. For any $\alpha \in \mathcal{M}_+(\mathbb{R}^d)$, $\mathrm{supp}(\alpha)$ is the support of $\alpha$, and $m(\alpha) = \int_{\mathbb{R}^d} \mathrm{d}\alpha(x) < +\infty$ is the mass of $\alpha$. For $\alpha \in \mathcal{M}_+(\mathbb{R}^d)$ and a map $T : \mathbb{R}^d \to \mathbb{R}^p$, $T_\# \alpha$ is the pushforward measure of $\alpha$ by $T$, defined for all $A \subset \mathbb{R}^d$ as $T_\# \mu(A) = \mu\big(T^{-1}(A)\big)$. Let $\delta_z$ be the Dirac measure at $z$ and for $n \geq 1$, define the empirical measure $\hat{\alpha}_n = \sum_{i=1}^n w_i \delta_{Z_i}$, where $(Z_i)_{i=1}^n$ are $n$ independent and identically distributed (i.i.d.) samples from $\alpha \in \mathcal{M}_+(\mathbb{R}^d)$, and $w_i > 0$. For any convex function $\varphi : \mathbb{R} \to \mathbb{R} \cup \{+\infty\}$, we denote by $\varphi^*$ its Legendre transform, $i.e.$, for $x \in \mathbb{R}$, $\varphi^*(x) = \sup_{y \geq 0} \; xy - \varphi(y)$. We will also use the notation $\varphi^\circ(x) = -\varphi^*(-x)$. For $\alpha, \beta \in \mathcal{M}_+(\mathbb{R}^d)$, $\alpha \otimes \beta$ is the product measure, and for $f, g : \mathbb{R}^d \to \mathbb{R}$, denote by $f \oplus g : \mathbb{R}^d \times \mathbb{R}^d \to \mathbb{R}$ the mapping defined as $(f \oplus g)(x, y) = f(x) + g(y)$ for all $x, y \in \mathbb{R}^d$. $\mathbb{S}^{d-1} \triangleq \{\theta \in \mathbb{R}^d \; : \; \|\theta\| = 1\}$ is the unit sphere, and for $\theta \in \mathbb{S}^{d-1}$, $\theta^* : \mathbb{R}^d \to \mathbb{R}$ is the mapping defined as $\theta^*(x) = \langle \theta, x \rangle$ for all $x \in \mathbb{R}^d$.

### 2.2 Unbalanced Optimal Transport

We recall the static formulation of unbalanced OT proposed by Liero et al. (2018), which uses $\varphi$-*divergences* as penalty terms.

**Definition 2.1** ($\varphi$-divergences)**.** *Let* $(\alpha, \beta) \in \mathcal{M}_+(\mathbb{R}^d) \times \mathcal{M}_+(\mathbb{R}^d)$*. Let* $\varphi : \mathbb{R} \to \mathbb{R} \cup \{+\infty\}$ *be an* entropy function, $i.e.$, $\varphi$ *is convex, lower semicontinuous,* $\mathrm{dom}(\varphi) \triangleq \{x \in \mathbb{R}, \; \varphi(x) < +\infty\} \subset [0, +\infty)$ *and* $\varphi(1) = 0$. *Denote* $\varphi'_\infty \triangleq \lim_{x \to +\infty} \frac{\varphi(x)}{x}$. *The* $\varphi$-divergence *between* $\alpha$ *and* $\beta$ *is*

$$\mathrm{D}_\varphi(\alpha|\beta) \triangleq \int_{\mathbb{R}^d} \varphi\left(\frac{\mathrm{d}\alpha}{\mathrm{d}\beta}(x)\right) \mathrm{d}\beta(x) + \varphi'_\infty \int_{\mathbb{R}^d} \mathrm{d}\alpha^\perp(x), \tag{1}$$

*where* $\alpha^\perp$ *is defined as* $\alpha = \frac{\mathrm{d}\alpha}{\mathrm{d}\beta}\beta + \alpha^\perp$.

**Definition 2.2** (Unbalanced OT (Liero et al., 2018))**.** *Let* $(\varphi_1, \varphi_2)$ *be a pair of entropy functions and* $\mathrm{C}_d : \mathbb{R}^d \times \mathbb{R}^d \to \mathbb{R}$ *a cost function. The* unbalanced OT *problem between* $(\alpha, \beta) \in \mathcal{M}_+(\mathbb{R}^d) \times \mathcal{M}_+(\mathbb{R}^d)$ *reads*

$$\mathrm{UOT}(\alpha, \beta) \triangleq \inf_{\pi \in \mathcal{M}_+(\mathbb{R}^d \times \mathbb{R}^d)} \int \mathrm{C}_d(x, y) \mathrm{d}\pi(x, y) + \mathrm{D}_{\varphi_1}(\pi_1|\alpha) + \mathrm{D}_{\varphi_2}(\pi_2|\beta), \tag{2}$$

*where* $\pi_1$ *and* $\pi_2$ *denote the marginal distributions of* $\pi$ *with respect to (w.r.t.) the first and second variable respectively.*

When $\varphi_1 = \varphi_2$ and $\varphi_1(x) = 0$ for $x = 1$, $\varphi_1(x) = +\infty$ otherwise, (2) boils down to the Kantorovich formulation of OT (or *balanced OT*), denoted by $\mathrm{OT}(\alpha, \beta)$. Indeed, in that case, $\mathrm{D}_{\varphi_1}(\pi_1|\alpha) = \mathrm{D}_{\varphi_2}(\pi_2|\beta) = 0$ if $\pi_1 = \alpha$ and $\pi_2 = \beta$, $\mathrm{D}_{\varphi_1}(\pi_1|\alpha) = \mathrm{D}_{\varphi_2}(\pi_2|\beta) = +\infty$ otherwise.

Under other suitable choices of entropy functions $\varphi_1$ and $\varphi_2$, $\mathrm{UOT}(\alpha, \beta)$ is more robust than $\mathrm{OT}(\alpha, \beta)$, since it can discard outliers and compare $\alpha$ and $\beta$ with different masses. We refer to (Séjourné et al., 2022a, Section 4.2) for a detailed discussion on the choice of entropies and its consequences on the transport plan computed by UOT. Two common choices are $\varphi_i(x) = \rho\,|x - 1|$ and $\varphi_i(x) = \rho(x\log(x) - x + 1)$, where $\rho > 0$ is a characteristic radius w.r.t. $\mathrm{C}_d$. They respectively correspond to $\mathrm{D}_{\varphi_i} = \rho\mathrm{TV}$ (total variation distance (Chizat et al., 2018a)) and $\mathrm{D}_{\varphi_i} = \rho\mathrm{KL}$ (Kullback-Leibler divergence), and operate differently: KL smooths out geometric outliers, while TV either keeps or removes samples (Séjourné et al., 2022a). The GHK distance corresponds to (2) with $\mathrm{C}_d(x, y) = ||x - y||^2$ and $\mathrm{D}_{\varphi_i} = \rho_i\mathrm{KL}$ (Liero et al., 2018).

One can obtain an equivalent formulation of UOT by deriving the dual of (2) and proving strong duality. We recall this result below.

**Proposition 2.3** (Corollary 4.12 in (Liero et al., 2018))**.** *The* UOT *problem (2) can equivalently be written as* $\mathrm{UOT}(\alpha, \beta) = \sup_{f \oplus g \leq C_d} \mathcal{D}(f, g; \alpha, \beta)$*, with*

$$\mathcal{D}(f, g; \alpha, \beta) \triangleq \int \varphi_1^\circ\big(f(x)\big)\mathrm{d}\alpha(x) + \int \varphi_2^\circ\big(g(y)\big)\mathrm{d}\beta(y), \tag{3}$$

*where for* $i \in \{1, 2\}$*,* $\varphi_i^\circ(x) \triangleq -\varphi_i^*(-x)$ *with* $\varphi_i^*(x) \triangleq \sup_{y \geq 0} xy - \varphi_i(y)$ *the* Legendre transform *of* $\varphi_i$*, and* $f \oplus g \leq C_d$ *means that for* $(x, y) \sim \alpha \otimes \beta$*,* $f(x) + g(y) \leq C_d(x, y)$*.*

When clear from the context, we will omit the dependence on $(\alpha, \beta)$ and write $\mathcal{D}(f, g)$ instead of $\mathcal{D}(f, g; \alpha, \beta)$. The Legendre transform of $\varphi_i$ is well known for typical choices of $\varphi_i$-divergences. For example, if $\mathrm{D}_{\varphi_i} = \rho_i\mathrm{KL}$, then $\varphi_i^*(x) = \rho_i(e^{x/\rho_i} - 1)$.

Based on Proposition 2.3, one can compute $\mathrm{UOT}(\alpha, \beta)$ by optimizing a pair of continuous functions $(f, g)$. However, $\mathrm{UOT}(\alpha, \beta)$ is known to be computationally intensive (Pham et al., 2020), which motivates the development of methods able to scale to the large dimensions and sample sizes encountered in ML applications.

### 2.3 Sliced Optimal Transport

Among the many workarounds that have been proposed to overcome the OT computational bottleneck (Peyré et al., 2019), Sliced OT (Rabin et al., 2012) has attracted a lot of attention due to its computational benefits and theoretical guarantees.

**Definition 2.4** (Sliced OT)**.** *Let* $\mathbb{S}^{d-1} \triangleq \{\theta \in \mathbb{R}^d \ : \ \|\theta\| = 1\}$ *be the unit sphere in* $\mathbb{R}^d$*. For* $\theta \in \mathbb{S}^{d-1}$*, denote by* $\theta^\star : \mathbb{R}^d \to \mathbb{R}$ *the linear map such that for* $x \in \mathbb{R}^d$*,* $\theta^\star(x) \triangleq \langle \theta, x \rangle$*. Let* $\boldsymbol{\sigma}$ *be the uniform probability over* $\mathbb{S}^{d-1}$*. Consider* $(\alpha, \beta) \in \mathcal{M}_+(\mathbb{R}^d) \times \mathcal{M}_+(\mathbb{R}^d)$*. The* Sliced OT *problem is defined as*

$$\mathrm{SOT}(\alpha, \beta) \triangleq \int_{\mathbb{S}^{d-1}} \mathrm{OT}(\theta_\sharp^\star \alpha, \theta_\sharp^\star \beta)\mathrm{d}\boldsymbol{\sigma}(\theta), \tag{4}$$

*where for any measurable function* $f$ *and* $\xi \in \mathcal{M}_+(\mathbb{R}^d)$*,* $f_\sharp \xi$ *is the* push-forward measure *of* $\xi$ *by* $f$*, i.e., for any measurable set* $A \subset \mathbb{R}$*,* $f_\sharp \xi(A) \triangleq \xi(f^{-1}(A))$*,* $f^{-1}(A) \triangleq \{x \in \mathbb{R}^d : f(x) \in A\}$*.*

Since $(\theta_\sharp^\star \alpha, \theta_\sharp^\star \beta)$ are two measures supported on $\mathbb{R}$, $\mathrm{OT}(\theta_\sharp^\star \mu, \theta_\sharp^\star \nu)$ is defined in terms of a cost function $\mathrm{C}_1 : \mathbb{R} \times \mathbb{R} \to \mathbb{R}$, and can be efficiently computed. Therefore, $\mathrm{SOT}(\alpha, \beta)$ can provide significant computational advantages over $\mathrm{OT}(\alpha, \beta)$ in large-scale settings. In practice, if $\alpha = \sum_{i=1}^n \alpha_i \delta_{x_i}$ and $\beta = \sum_{i=1}^n \beta_i \delta_{y_i}$ are discrete measures, the standard procedure for approximating $\mathrm{SOT}(\alpha, \beta)$ consists in sampling $m$ i.i.d. samples $\{\theta_j\}_{j=1}^m$ from $\boldsymbol{\sigma}$, then computing $\mathrm{OT}\big((\theta_j^\star)_\sharp \alpha, (\theta_j^\star)_\sharp \beta\big)$ for $j = 1, \ldots, m$. This second step involves sorting the $n$ support points of $\alpha$ and $\beta$ (Peyré et al., 2019, Section 2.6), thus involves $\mathcal{O}(n \log n)$ operations per $\theta_j$.

$\mathrm{SOT}(\alpha, \beta)$ relies on the Kantorovich formulation of OT, thus $\mathrm{SOT}(\alpha, \beta) < +\infty$ only when $m(\alpha) = m(\beta)$, and may not provide meaningful comparisons in presence of outliers. To overcome such limitations, prior works have proposed slicing a particular instance of UOT that is partial OT (Bonneel & Coeurjolly, 2019; Bai et al., 2023), for which $\mathrm{D}_\varphi$ is the total variation distance. More precisely, noting POT the UOT problem with $\mathrm{D}_\varphi = \rho\mathrm{TV}$, they consider for $\alpha, \beta \in \mathcal{M}_+(\mathbb{R}^d)$ the problem

$$\mathrm{SPOT}(\alpha, \beta) \triangleq \int_{\mathbb{S}^{d-1}} \mathrm{POT}(\theta_\sharp^\star \alpha, \theta_\sharp^\star \beta)\mathrm{d}\boldsymbol{\sigma}(\theta). \tag{5}$$

For the 1D partial OT problem, Bonneel & Coeurjolly (2019) solve a one dimensional injective partial assignment in quasilinear complexity, but which does not allow for mass destruction in the source measure, while Bai et al. (2023) proposed an efficient procedure with a quadratic worst case complexity. However, their algorithms only apply to measures whose samples have constant mass (*e.g.*, $\alpha_i = \beta_j = 1$). In the next section, we generalize their line of work and propose a new way of combining sliced OT and unbalanced OT.

# 3 Sliced Unbalanced OT and Unbalanced Sliced OT

We present two new scalable and robust OT problems, by combining the unbalanced and slicing strategies in two different ways. We conduct a theoretical analysis of both strategies and provide a comparison of the two. For ease of exposition, all proofs of the results in this section are provided in Appendix A.

First, we propose to *slice the unbalanced OT problem*: we average the UOT problem over different projections of the compared measures, similar to the approach of sliced partial OT (Bonneel & Coeurjolly, 2019; Bai et al., 2023). We refer to this problem as Sliced Unbalanced OT (SUOT) and introduce it in Section 3.1. Next, we explore the reverse strategy, *i.e.*, we *unbalance the sliced OT problem*: the weights of SUOT are penalized to introduce imbalance, analogous to how UOT relates to OT. We call this method *Unbalanced Sliced OT* (USOT) and present it in Section 3.2.

## 3.1 Sliced Unbalanced Optimal Transport

Our first strategy consists in slicing the unbalanced OT problem and leads to the following definition.

**Definition 3.1** (Sliced Unbalanced OT). *Let $(\varphi_1, \varphi_2)$ be a pair of entropy functions and $\mathrm{C}_1 : \mathbb{R} \times \mathbb{R} \to \mathbb{R}$ a cost function. The* sliced unbalanced OT *problem between $(\alpha, \beta) \in \mathcal{M}_+(\mathbb{R}^d) \times \mathcal{M}_+(\mathbb{R}^d)$ reads*

$$\mathrm{SUOT}(\alpha, \beta) \triangleq \int_{\mathbb{S}^{d-1}} \mathrm{UOT}(\theta_\sharp^\star \alpha, \theta_\sharp^\star \beta) \mathrm{d}\boldsymbol{\sigma}(\theta) \,, \tag{6}$$

*where for $\theta \sim \boldsymbol{\sigma}$, $\mathrm{UOT}(\theta_\sharp^\star \alpha, \theta_\sharp^\star \beta) = \inf_{\pi_\theta \in \mathcal{M}_+(\mathbb{R} \times \mathbb{R})} \int_{\mathbb{R} \times \mathbb{R}} \mathrm{C}_1(x, y) \mathrm{d}\pi_\theta(x, y) + \mathrm{D}_{\varphi_1}\big((\pi_\theta)_1 | \theta_\sharp^\star \alpha\big) + \mathrm{D}_{\varphi_2}\big((\pi_\theta)_2 | \theta_\sharp^\star \beta\big)$ with $(\pi_\theta)_1, (\pi_\theta)_2$ the marginal distributions of $\pi_\theta$.*

By definition, SUOT is a specific instance of the class of sliced probability divergences (Nadjahi et al., 2020a), where the *base divergence* is chosen as UOT. SUOT can also be interpreted as a general expression of the sliced partial OT problem (Bonneel & Coeurjolly, 2019; Bai et al., 2023): while the latter imposes $\mathrm{D}_{\varphi_i} = \rho_i \mathrm{TV}$, SUOT allows for the use of arbitrary $\varphi$-divergences.

In the following, we establish a set of theoretical properties for SUOT with different choices of $\varphi$-divergences and cost functions $\mathrm{C}_1$. First, we identify sufficient conditions for which the solution of (6) exists.

**Proposition 3.2** (SUOT: Existence of solutions). *Assume that $\mathrm{C}_1$ is lower-semicontinuous and that either (i) $\varphi'_{1,\infty} = \varphi'_{2,\infty} = +\infty$, or (ii) $\mathrm{C}_1$ has compact sublevels on $\mathbb{R} \times \mathbb{R}$ and $\varphi'_{1,\infty} + \varphi'_{2,\infty} + \inf \mathrm{C}_1 > 0$. Then, the solution of $\mathrm{SUOT}(\alpha, \beta)$ exists, in the sense that for any $\theta \sim \boldsymbol{\sigma}$, there exists $\pi_\theta^* \in \mathcal{M}_+(\mathbb{R} \times \mathbb{R})$ attaining the infimum in $\mathrm{UOT}(\theta_\sharp^\star \alpha, \theta_\sharp^\star \beta)$.*

The assumptions of Proposition 3.2 are met for some settings of interest, including $\mathrm{D}_{\varphi_1} = \mathrm{D}_{\varphi_2} = \mathrm{KL}$ (since $\varphi'_\infty = +\infty$), or $\mathrm{D}_{\varphi_1} = \mathrm{D}_{\varphi_2} = \mathrm{TV}$ and $C_1(x, y) = |x - y|^p$ $(p \geq 1)$ (since $\varphi'_\infty = 1$): see (Séjourné et al., 2022a, Section 2.1) for more details.

Next, we show some topological properties of SUOT. In the next proposition, we prove that SUOT preserves the metric properties of UOT, which is consistent with (Nadjahi et al., 2020a, Proposition 1). In Section 3.3, we study the metrization of the weak*-topology with SUOT.

**Proposition 3.3** (SUOT: Metric properties). *Suppose* UOT *is non-negative, symmetric and/or definite on $\mathcal{M}_+(\mathbb{R}) \times \mathcal{M}_+(\mathbb{R})$. Then,* SUOT *is respectively non-negative, symmetric and/or definite on $\mathcal{M}_+(\mathbb{R}^d) \times \mathcal{M}_+(\mathbb{R}^d)$. If there exists $p \in [1, +\infty)$ s.t. for any $\alpha, \beta, \gamma \in \mathcal{M}_+(\mathbb{R})$, $\mathrm{UOT}^{1/p}(\alpha, \beta) \leq \mathrm{UOT}^{1/p}(\alpha, \gamma) + \mathrm{UOT}^{1/p}(\gamma, \beta)$, then $\mathrm{SUOT}^{1/p}(\alpha, \beta) \leq \mathrm{SUOT}^{1/p}(\alpha, \gamma) + \mathrm{SUOT}^{1/p}(\gamma, \beta)$.*

By Proposition 3.3, establishing the metric axioms of UOT between *univariate* measures (as detailed in (Séjourné et al., 2022a, Section 3.3.1)) is sufficient to prove the metric properties of SUOT between *multivariate* measures. For example, since GHK is a metric for the order $p = 2$ (Liero et al., 2018), so is the induced SUOT.

We move on to the statistical aspects and study the sample complexity of SUOT. For $(\alpha, \beta) \in \mathcal{M}_+(\mathbb{R}^d) \times \mathcal{M}_+(\mathbb{R}^d)$, we establish the speed of convergence of $\mathrm{SUOT}(\hat{\alpha}_n, \hat{\beta}_n)$ toward $\mathrm{SUOT}(\alpha, \beta)$, where $\hat{\alpha}_n, \hat{\beta}_n$ denote

the empirical measures supported on $n$ independent samples from $\alpha, \beta$ respectively (as defined in Section 2.1). We prove below that SUOT extends the sample complexity of UOT from one-dimensional settings to multi-dimensional ones.

**Theorem 3.4** (SUOT: Sample complexity). (i) *Assume for $(\mu, \nu) \in \mathrm{M} \times \mathrm{M}$ with $\mathrm{M} \subset \mathcal{M}_+(\mathbb{R})$, $\mathbb{E}|\mathrm{UOT}(\mu, \nu) - \mathrm{UOT}(\hat{\mu}_n, \hat{\nu}_n)| \leq \kappa(n)$. Then, for $(\alpha, \beta) \in \widetilde{\mathrm{M}} \times \widetilde{\mathrm{M}}$ with $\widetilde{\mathrm{M}} \triangleq \{\eta \in \mathcal{M}_+(\mathbb{R}^d) : \forall \theta \in \mathbb{S}^{d-1}, \theta^\star_\sharp \eta \in \mathrm{M}\}$, $\mathbb{E}|\mathrm{SUOT}(\alpha, \beta) - \mathrm{SUOT}(\hat{\alpha}_n, \hat{\beta}_n)| \leq \kappa(n)$.*

(ii) *Assume for $\mu \in \mathrm{M}$ with $\mathrm{M} \subset \mathcal{M}_+(\mathbb{R})$, $\mathbb{E}|\mathrm{UOT}(\mu, \hat{\mu}_n)| \leq \xi(n)$. Then, for $\alpha \in \widetilde{\mathrm{M}}$ with $\widetilde{\mathrm{M}} \triangleq \{\eta \in \mathcal{M}_+(\mathbb{R}^d) : \forall \theta \in \mathbb{S}^{d-1}, \theta^\star_\sharp \eta \in \mathrm{M}\}$, $\mathbb{E}|\mathrm{SUOT}(\alpha, \hat{\alpha}_n)| \leq \xi(n)$.*

Note that the expectations in Theorem 3.4 are taken with respect to the samples of the empirical measures, which are random. Theorem 3.4 shows that SUOT enjoys a *dimension-free* sample complexity, even when comparing multivariate measures. This advantage is recurrent of sliced divergences (Nadjahi et al., 2020b) and further motivates their use on high-dimensional settings. The sample complexity rates $\kappa(n)$ or $\xi(n)$ can be deduced from the literature on UOT for univariate measures. For instance, in the GHK setting, the rate is given by $\kappa(n) \propto n^{-1/2}$ for measures with compact, convex support and continuously differentiable densities (Vacher & Vialard, 2023, Corollary 3.4), and a suitable class $\widetilde{\mathrm{M}}$ can be defined.

Finally, we derive the dual formulation of SUOT and prove that strong duality holds. This result has important practical implications, as we will leverage it in Section 4 to develop a methodology for computing SUOT. Note that the computation of SUOT involves integration with respect to $\boldsymbol{\sigma}$, which generally cannot be done in closed from, as is the case for most sliced divergences. Since our goal is to develop a practical and implementable method, we will consider the Monte Carlo approximation commonly used by practitioners to compute sliced divergences (Nadjahi et al., 2020a): we approximate $\mathrm{SUOT}(\alpha, \beta)$ as $\int_{\mathbb{S}^{d-1}} \mathrm{UOT}(\theta^\star_\sharp \alpha, \theta^\star_\sharp \beta) \mathrm{d}\hat{\boldsymbol{\sigma}}_K(\theta)$, where $\hat{\boldsymbol{\sigma}}_K$ is a discrete distribution supported on $K$ i.i.d. samples drawn from $\boldsymbol{\sigma}$.

**Theorem 3.5** (SUOT: Strong duality). *For $i \in \{1, 2\}$, let $\varphi_i$ be an entropy function such that $\mathrm{dom}(\varphi_i^*) \cap \mathbb{R}_-$ is non-empty, and either $0 \in \mathrm{dom}(\varphi_i)$ or $m(\alpha), m(\beta) \in \mathrm{dom}(\varphi_i)$. Let $\mathcal{E} \triangleq \{(f_\theta, g_\theta)_{\theta \in \mathrm{supp}(\hat{\boldsymbol{\sigma}}_K)} : \forall \theta \in \mathrm{supp}(\hat{\boldsymbol{\sigma}}_K), f_\theta \oplus g_\theta \leq \mathrm{C}_1\}$. Then,*

$$\mathrm{SUOT}(\alpha, \beta) = \sup_{(f_\theta, g_\theta) \in \mathcal{E}} \int_{\mathbb{S}^{d-1}} \mathcal{D}(f_\theta, g_\theta; \theta^\star_\# \alpha, \theta^\star_\# \beta) \mathrm{d}\hat{\boldsymbol{\sigma}}_K(\theta). \tag{7}$$

## 3.2 Unbalanced Sliced Optimal Transport

As a second strategy to make unbalanced OT scalable, we propose to unbalance sliced OT. To this end, we start with the following formulation of UOT (Liero et al., 2018),

$$\mathrm{UOT}(\alpha, \beta) = \inf_{(\pi_1, \pi_2) \in \mathcal{M}_+(\mathbb{R}^d) \times \mathcal{M}_+(\mathbb{R}^d)} \mathrm{OT}(\pi_1, \pi_2) + \mathrm{D}_{\varphi_1}(\pi_1 | \alpha) + \mathrm{D}_{\varphi_2}(\pi_2 | \beta), \tag{8}$$

and we replace UOT by its sliced counterpart, SOT. This yields the following definition:

**Definition 3.6** (Unbalanced Sliced OT). *Let $(\varphi_1, \varphi_2)$ be a pair of entropy functions. The* unbalanced sliced OT *problem between $(\alpha, \beta) \in \mathcal{M}_+(\mathbb{R}^d) \times \mathcal{M}_+(\mathbb{R}^d)$ reads*

$$\mathrm{USOT}(\alpha, \beta) \triangleq \inf_{(\pi_1, \pi_2) \in \mathcal{M}_+(\mathbb{R}^d) \times \mathcal{M}_+(\mathbb{R}^d)} \mathrm{SOT}(\pi_1, \pi_2) + \mathrm{D}_{\varphi_1}(\pi_1 | \alpha) + \mathrm{D}_{\varphi_2}(\pi_2 | \beta). \tag{9}$$

This approach is entirely novel since, to the best of our knowledge, it has never been studied in prior work, even for specific choices of entropy. To gain a better grasp of this new object, USOT, we examine how the theoretical properties discussed in the previous section apply here.

We first prove that the solution of (9) exists under the same conditions as those for SUOT outlined in Proposition 3.2.

**Proposition 3.7** (USOT: Existence of solutions). *Assume that $\mathrm{C}_1$ is lower-semicontinuous and that either (i) $\varphi'_{1,\infty} = \varphi'_{2,\infty} = +\infty$, or (ii) $\mathrm{C}_1$ has compact sublevels on $\mathbb{R} \times \mathbb{R}$ and $\varphi'_{1,\infty} + \varphi'_{2,\infty} + \inf \mathrm{C}_1 > 0$. Then, the solution of $\mathrm{USOT}(\alpha, \beta)$ exists: there exists $(\pi_1^*, \pi_2^*) \in \mathcal{M}_+(\mathbb{R}^d) \times \mathcal{M}_+(\mathbb{R}^d)$ attaining the infimum in (9).*

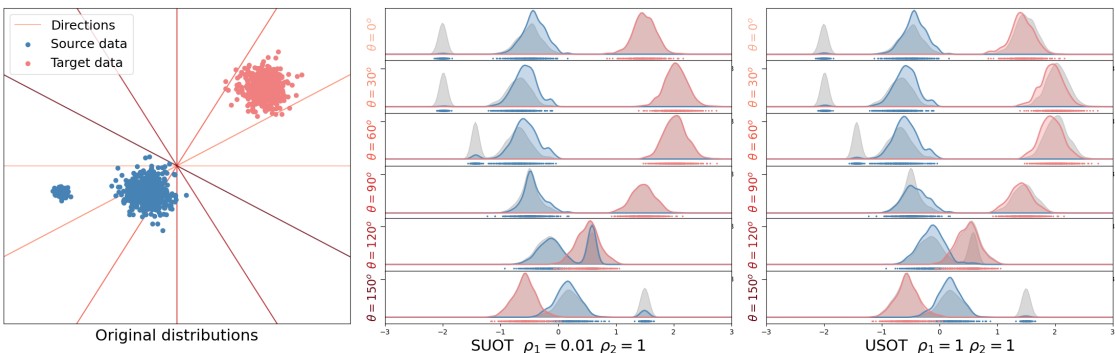

Figure 1: **Toy illustration** on the behaviors of SUOT and USOT. *(left)* Original 2D samples and slices used for illustration. KDE density estimations of the projected samples: grey, original distributions, colored, distributions reweighed by SUOT *(center)*, and reweighed by USOT *(right)*.

We then review the conditions under which USOT is a (pseudo-)metric, and we prove that strong duality holds. Similar to SUOT, the dual formulation that we derive will enable the design of an algorithm for effectively computing USOT in practice.

**Proposition 3.8** (USOT: Metric properties). *For any $(\alpha, \beta) \in \mathcal{M}_+(\mathbb{R}^d) \times \mathcal{M}_+(\mathbb{R}^d)$, $\mathrm{USOT}(\alpha, \beta) \geq 0$. If $\varphi_1 = \varphi_2$, $\mathrm{USOT}$ is symmetric. If $D_{\varphi_1}$ and $D_{\varphi_2}$ are definite, then $\mathrm{USOT}$ is definite. If $\mathrm{C}_1(x, y) = |x - y|$ and $D_{\varphi_1} = D_{\varphi_2} = \rho \mathrm{TV}$, then, $\mathrm{USOT}$ satisfies the triangle inequality.*

**Theorem 3.9** (USOT: Strong duality). *For $i \in \{1, 2\}$, let $\varphi_i$ be an entropy function such that $\mathrm{dom}(\varphi_i^*) \cap \mathbb{R}_-$ is non-empty, and either $0 \in \mathrm{dom}(\varphi_i)$ or $m(\alpha), m(\beta) \in \mathrm{dom}(\varphi_i)$. Let $\mathcal{E} \triangleq \{(f_\theta, g_\theta)_{\theta \in \mathrm{supp}(\hat{\boldsymbol{\sigma}}_K)} : \forall \theta \in \mathrm{supp}(\hat{\boldsymbol{\sigma}}_K), f_\theta \oplus g_\theta \leq \mathrm{C}_1\}$. Then,*

$$\mathrm{USOT}(\alpha, \beta) = \sup_{(f_\theta, g_\theta) \in \mathcal{E}} \mathcal{D}\left(\int_{\mathbb{S}^{d-1}} f_\theta \circ \theta^* \mathrm{d}\hat{\boldsymbol{\sigma}}_K(\theta), \int_{\mathbb{S}^{d-1}} g_\theta \circ \theta^* \mathrm{d}\hat{\boldsymbol{\sigma}}_K(\theta); \alpha, \beta\right). \tag{10}$$

Since USOT does not belong to the class of sliced divergences, establishing its sample complexity is more challenging compared to SUOT. Based on the literature, one standard technique involves deriving covering number bounds on the space of the dual potentials of USOT. This theoretical question is highly non-trivial given the complex structure of $\mathcal{E}$, and as such is out of the scope of this paper. Nevertheless, we investigate the sample complexity on empirical settings: our experimental results presented in Appendix C.5 suggest that USOT might also enjoy a dimension-free rate.

### 3.3 Comparative Analysis of Sliced Unbalanced and Unbalanced Sliced Optimal Transport

In addition to the theoretical analysis previously conducted for SUOT and USOT independently, this section provides further insights to better grasp the differences between these two strategies.

First, by comparing Definition 3.1 with Definition 3.6, SUOT and USOT clearly differ at the conceptual level. Specifically, $\mathrm{SUOT}(\alpha, \beta)$ penalizes the marginals of $\pi_\theta$ for $\theta \sim \boldsymbol{\sigma}$, where $\pi_\theta$ is the coupling that transports mass from $\theta_\sharp^* \alpha$ to $\theta_\sharp^* \beta$. In contrast, $\mathrm{USOT}(\alpha, \beta)$ directly regularizes the marginals of the coupling between $\alpha$ and $\beta$. To illustrate this difference, we consider $(\alpha, \beta) \in \mathcal{M}_+(\mathbb{R}^2) \times \mathcal{M}_+(\mathbb{R}^2)$ with $\alpha$ contaminated by outliers, then compute $\mathrm{SUOT}(\alpha, \beta)$ and $\mathrm{USOT}(\alpha, \beta)$. We plot $(\alpha, \beta)$ and the sampled projections $(\theta_k)_k$ (Figure 1, left), the marginals of $(\pi_{\theta_k})_k$ obtained with $\mathrm{SUOT}(\alpha, \beta)$ (Figure 1, center), and the marginals of $((\theta_k^*)_\sharp \pi)_k$ with $\mathrm{USOT}(\alpha, \beta)$ (Figure 1, right). We observe that the source outliers in $\alpha$ have been successfully removed by $\mathrm{USOT}(\alpha, \beta)$ for all $\theta_k$, while they may still appear with $\mathrm{SUOT}(\alpha, \beta)$ (*e.g.*, Figure 1, center: note the bimodal marginal in blue for $\theta = 120°$). This difference is due to the marignal penalization terms in $\mathrm{USOT}(\alpha, \beta)$, which operate directly w.r.t. $(\alpha, \beta)$ rather than their projections $(\theta_\sharp^* \alpha, \theta_\sharp^* \beta)$, unlike $\mathrm{SUOT}(\alpha, \beta)$.

A question of particular interest regarding probability divergences is how they relate to each other, specifically whether they yield equivalent topologies. We explore this question for SUOT and USOT. To do so, we

consider a notion of equivalence that is frequently studied in the literature, as in (Bayraktar & Guo, 2021, Theorem 2.3.(i)).

We start by proving a first set of inequalities that relate SUOT, USOT and UOT: our next theorem shows that USOT is always greater than SUOT and that, for appropriate choices of cost functions, USOT is upper-bounded by UOT.

**Theorem 3.10.** *For any* $(\alpha, \beta) \in \mathcal{M}_+(\mathbb{R}^d) \times \mathcal{M}_+(\mathbb{R}^d)$,

$$\mathrm{SUOT}(\alpha, \beta) \leq \mathrm{USOT}(\alpha, \beta) \,. \tag{11}$$

*Moreover, suppose that,* $\forall (x, y) \in \mathbb{R}^d \times \mathbb{R}^d, \forall \theta \in \mathbb{S}^{d-1}, \mathrm{C}_1\big(\theta^\star(x), \theta^\star(y)\big) \leq \mathrm{C}_d(x, y)$. *Then, for any* $(\alpha, \beta) \in \mathcal{M}_+(\mathbb{R}^d) \times \mathcal{M}_+(\mathbb{R}^d)$,

$$\mathrm{SUOT}(\alpha, \beta) \leq \mathrm{USOT}(\alpha, \beta) \leq \mathrm{UOT}(\alpha, \beta) \,. \tag{12}$$

In particular, Theorem 3.10 holds for the following common choice of costs: $\forall (s, t) \in \mathbb{R}^2, \mathrm{C}_1(s, t) = |s - t|^p$ and $\forall (x, y) \in \mathbb{R}^d \times \mathbb{R}^d, \mathrm{C}_d(x, y) = \|x - y\|^p$, with $p \in [1, +\infty)$.

Next, we prove that $\mathrm{UOT}(\alpha, \beta)$ can be upper-bounded by a functional of $\mathrm{SUOT}(\alpha, \beta)$ when $(\alpha, \beta)$ have compact supports, by adapting the reasoning from Bonnotte (2013, Lemma 5.1.4) to our setting and considering the duals of UOT (Proposition 2.3) and SUOT (Theorem 3.5) instead of the dual of OT and SOT. Most arguments in (Bonnotte, 2013) adapt well to our setting, but establishing a Lipschitz condition on the integrand of the dual required a more technical approach. To this end, we prove Lemma A.13, which results in a different constant value, denoted as $c(m(\alpha), m(\beta), \rho, R)$.

**Theorem 3.11.** *Let* $\mathsf{X} \subset \mathbb{R}^d$ *be a compact set with radius* $R$. *Define the cost functions as* $\mathrm{C}_1(s, t) = |s - t|^p, (s, t) \in \mathbb{R}^2$, *and* $\mathrm{C}_d(x, y) = \|x - y\|^p, (x, y) \in \mathbb{R}^d \times \mathbb{R}^d$, *with* $p \in [1, +\infty)$. *Assume either, (i)* $D_{\varphi_1} = D_{\varphi_2} = \rho \mathrm{KL}$; *or (ii)* $p = 1$ *and* $D_{\varphi_1} = D_{\varphi_2} = \rho \mathrm{TV}$. *Then, for any* $(\alpha, \beta) \in \mathcal{M}_+(\mathsf{X}) \times \mathcal{M}_+(\mathsf{X})$,

$$\mathrm{UOT}(\alpha, \beta) \leq c(m(\alpha), m(\beta), \rho, R) \, \mathrm{SUOT}(\alpha, \beta)^{\frac{1}{d+1}} \,,$$

*where* $c(m(\alpha), m(\beta), \rho, R)$ *is a constant depending on* $m(\alpha), m(\beta), \rho, R$, *which is non-decreasing in* $m(\alpha)$ *and* $m(\beta)$.

We show the equivalence of SUOT, USOT and UOT by combining Theorem 3.11 and Theorem 3.10, assuming that the constant $c(m(\alpha), m(\beta), \rho, R)$ does not depend on $m(\alpha), m(\beta)$. This occurs, for example, when the masses of $\alpha$ and $\beta$ are uniformly bounded; that is, there exists $M \in \mathbb{R}_+$ such that $m(\alpha) \leq M$ and $m(\beta) \leq M$.

The equivalence of SUOT, USOT and UOT is a key result for proving that SUOT and USOT *metrize weak\* convergence*, provided that UOT does (as in the GHK setting (Liero et al., 2018, Theorem 7.25)). Recall that a sequence of positive measures $(\alpha_n)_{n \in \mathbb{N}^*}$ converges weakly to $\alpha \in \mathcal{M}_+(\mathbb{R}^d)$ (denoted by $\alpha_n \rightharpoonup \alpha$) if, for any continuous and bounded $f : \mathbb{R}^d \to \mathbb{R}, \lim_{n \to +\infty} \int f \mathrm{d}\alpha_n = \int f \mathrm{d}\alpha$.

**Theorem 3.12** (Metrizability of the weak\* topology by SUOT, USOT)**.** *Assume the conditions in Theorem 3.11 are met. Let* $(\alpha_n)_{n \in \mathbb{N}^*}$ *be a sequence of measures in* $\mathcal{M}_+(\mathsf{X})$ *and* $\alpha \in \mathcal{M}_+(\mathsf{X})$, *where* $\mathsf{X} \subset \mathbb{R}^d$ *is a compact set with radius* $R$. *Then,* SUOT *and* USOT *metrize the weak\* convergence, i.e.,* $\alpha_n \rightharpoonup \alpha \Leftrightarrow \lim_{n \to +\infty} \mathrm{SUOT}(\alpha_n, \alpha) = 0$, *and* $\alpha_n \rightharpoonup \alpha \Leftrightarrow \lim_{n \to +\infty} \mathrm{USOT}(\alpha_n, \alpha) = 0$.

The metrizability of weak\* convergence was not studied in related work, including in existing instances of our framework, such as partial OT (Bonneel & Coeurjolly, 2019; Bai et al., 2023). In addition to complementing prior work, our result paves the way for other research directions. For instance, it can be used to justify the well-posedness of approximating an unbalanced Wasserstein gradient flow (Ambrosio et al., 2005) using SUOT, as done for SOT in (Candau-Tilh, 2020; Bonet et al., 2022). Unbalanced Wasserstein gradient flows have been a key tool in deep learning theory, *e.g.*, to prove global convergence of one-hidden layer neural networks (Chizat & Bach, 2018; Rotskoff et al., 2019).

## 4 Computing SUOT and USOT with Frank-Wolfe algorithms

In this section, we propose two algorithms to compute SUOT and USOT in practice. The resulting procedures are given in Algorithms 2 and 3 respectively, and require smooth penalty terms $(D_{\varphi_1}, D_{\varphi_2})$. This condition

is satisfied in the GHK setting ($D_{\varphi_i} = \rho_i \text{KL}$), but not for sliced partial OT ($D_{\varphi_i} = \rho_i \text{TV}$, Bai et al. (2023)). Our strategy is inspired by Séjourné et al. (2022b), where they proposed to solve the unbalanced OT problem between univariate measures using the Frank-Wolfe algorithm (as recalled in Appendix B.2). More precisely, we apply FW to optimize translation-invariant forms of the dual problems derived in Theorems 3.5 and 3.9.

## 4.1 Background: Frank-Wolfe Algorithm and Application to One-Dimensional Unbalanced OT

FW is a popular iterative first-order optimization algorithm for solving $\max_{x \in \mathcal{E}} \mathcal{H}(x)$, where $\mathcal{E}$ is a compact convex set and $\mathcal{H} : \mathcal{E} \to \mathbb{R}$ a concave, differentiable function. The procedure consists in maximizing a linear approximation of $\mathcal{H}$ at each iteration: given the current iterate $x_t$, FW solves the *linear oracle* $r_{t+1} \in \arg\max_{r \in \mathcal{E}} \langle \nabla \mathcal{H}(x_t), r \rangle$, then performs $x_{t+1} = (1 - \gamma_{t+1}) x_t + \gamma_{t+1} r_{t+1}$ with stepsize $\gamma_{t+1}$ typically chosen as $\gamma_{t+1} = \frac{2}{2+t+1}$. We refer to this step as FWStep and report the pseudo-code in Appendix B.2.

Séjourné et al. (2022b) apply FW to solve a translation-invariant formulation of the dual of $\text{UOT}(\alpha, \beta)$ for $(\alpha, \beta) \in \mathcal{M}_+(\mathbb{R}) \times \mathcal{M}_+(\mathbb{R})$, and show that the linear oracle in FWStep is the dual of $\text{OT}(\alpha_t, \beta_t)$ where $(\alpha_t, \beta_t)$ are normalized versions of $(\alpha, \beta)$, *i.e.*, $m(\alpha_t) = m(\beta_t) = 1$. Therefore, computing UOT amounts to solve a sequence of OT problems, which can efficiently be done since $(\alpha_t, \beta_t)$ are univariate probability measures. The expression of $(\alpha_t, \beta_t)$ depend on the input measures $(\alpha, \beta)$, the current iterates $(f_t, g_t)$ and the penalty coefficients $(\rho_1, \rho_2)$.

## 4.2 Frank-Wolfe Solvers for Sliced Unbalanced and Unbalanced Sliced OT

**Translation-invariant duals.** We compute $\text{SUOT}(\alpha, \beta)$ and $\text{USOT}(\alpha, \beta)$ for any $(\alpha, \beta) \in \mathcal{M}_+(\mathbb{R}^d) \times \mathcal{M}_+(\mathbb{R}^d)$ by solving translation-invariant formulations of their duals with FW. By Theorems 3.5 and 3.9, and adapting the reasoning of Séjourné et al. (2022b), we prove that

$$\text{SUOT}(\alpha, \beta) = \sup_{(f_\theta, g_\theta) \in \mathcal{E}} \int_{\mathbb{S}^{d-1}} \mathcal{H}(f_\theta, g_\theta; \theta^\star_\# \alpha, \theta^\star_\# \beta) \mathrm{d}\hat{\boldsymbol{\sigma}}_K(\theta) \,, \tag{13}$$

$$\text{USOT}(\alpha, \beta) = \sup_{(f_\theta, g_\theta) \in \mathcal{E}} \mathcal{H}\left( \int_{\mathbb{S}^{d-1}} f_\theta \circ \theta^\star \mathrm{d}\hat{\boldsymbol{\sigma}}_K(\theta), \int_{\mathbb{S}^{d-1}} g_\theta \circ \theta^\star \mathrm{d}\hat{\boldsymbol{\sigma}}_K(\theta); \alpha, \beta \right) \tag{14}$$

where $\mathcal{H}(f, g; \alpha, \beta) \triangleq \sup_{\lambda \in \mathbb{R}} \mathcal{D}(f + \lambda, g - \lambda; \alpha, \beta)$. These alternative duals are translation-invariant since, for any $\lambda \in \mathbb{R}$, $\mathcal{H}(f + \lambda, g - \lambda; \alpha, \beta) = \mathcal{H}(f, g; \alpha, \beta)$. If $(\varphi_1^\circ, \varphi_2^\circ)$ are smooth and strictly concave, then the maximizer in $\mathcal{H}$, denoted by $\lambda^\star(f, g)$, exists and is unique. In particular, when $D_{\varphi_1} = \rho_1 \text{KL}$ and $D_{\varphi_2} = \rho_2 \text{KL}$, $\lambda^\star(f, g)$ admits an analytical expression, which is given in the normalization routine (Algorithm 1). This is convenient as it avoids the need for approximate solvers to compute $\mathcal{H}(f, g; \alpha, \beta)$.

**Frank-Wolfe iterations.** We then apply FW to solve (13) and (14). We show that each iteration consists in solving a particular sliced OT problem between probability measures that depend on the input $(\alpha, \beta)$ and the iterates. To clarify this point, we present below the updates of FWStep tailored for each problem, starting with SUOT.

**Proposition 4.1** (Frank-Wolfe iterations for SUOT). *Let* $(\alpha, \beta) \in \mathcal{M}_+(\mathbb{R}^d) \times \mathcal{M}_+(\mathbb{R}^d)$ *and consider solving (13) with FW. Assume that* $(\varphi_1^\circ, \varphi_2^\circ)$ *are smooth and strictly concave. Given current iterates* $(f_\theta^t, g_\theta^t)_{\theta \in \text{supp}(\hat{\boldsymbol{\sigma}}_K)} \in \mathcal{E}$, *the solutions of the linear oracle*

---

**Algorithm 1** $-\text{Norm}(\alpha, \beta, f, g, \rho_1, \rho_2)$

**Input:** $\alpha, \beta, f, g, \rho_1, \rho_2$
**Output:** Normalized measures $(\bar{\alpha}, \bar{\beta})$

$$\lambda^\star \leftarrow \frac{\rho_1 \rho_2}{\rho_1 + \rho_2} \log \left( \frac{\int e^{-f(x)/\rho_1} \mathrm{d}\alpha(x)}{\int e^{-g(y)/\rho_2} \mathrm{d}\beta(y)} \right)$$

$$\bar{\alpha} \leftarrow e^{-\frac{(f(x) + \lambda^\star)}{\rho_1}} \alpha$$

$$\bar{\beta} \leftarrow e^{-\frac{(g(y) - \lambda^\star)}{\rho_2}} \beta$$

Return $(\bar{\alpha}, \bar{\beta})$

---

$(r_\theta^t, s_\theta^t)_{\theta \in \text{supp}(\hat{\boldsymbol{\sigma}}_K)}$ *are the dual potentials of* $\int_{\mathbb{S}^{d-1}} \text{OT}(\theta^\star_\sharp \alpha_\theta^t, \theta^\star_\sharp \beta_\theta^t) \mathrm{d}\hat{\boldsymbol{\sigma}}_K(\theta)$, *where* $(\alpha_\theta^t, \beta_\theta^t)$ *are measures given by* $\alpha_\theta^t = \nabla \varphi^\circ(f_\theta^t + \lambda^\star(f_\theta^t, g_\theta^t)) \alpha$ *and* $\beta_\theta^t = \nabla \varphi^\circ(g_\theta^t - \lambda^\star(f_\theta^t, g_\theta^t)) \beta$.

Proposition 4.1 shows that each FW iteration for solving the translation-invariant dual of $\text{SUOT}(\alpha, \beta)$ reduces to solving a balanced sliced OT problem: by (Séjourné et al., 2022b, Proposition 1), the measures $(\alpha_\theta^t, \beta_\theta^t)$ have the same mass, *i.e.*, $m(\alpha_\theta^t) = m(\beta_\theta^t)$. When using KL-based penalty terms, the procedure for computing $(\alpha_\theta^t, \beta_\theta^t)$ is detailed in Algorithm 1, and reports the closed-form expression of $\lambda^\star(f_\theta^t, g_\theta^t)$.

---

**Algorithm 2** – SUOT

**Input:** $\alpha$, $\beta$, $F$, $(\theta_k)_{k=1}^K$, $\rho_1$, $\rho_2$
**Output:** $\text{SUOT}(\alpha,\beta)$, $(f_\theta, g_\theta)$

  $(f_\theta, g_\theta) \leftarrow (0,0)$
  **for** $t = 0, 1, \ldots, F - 1$ **do**
    **for** $\theta \in (\theta_k)_{k=1}^K$ **do**
      $(\alpha_\theta, \beta_\theta) \leftarrow \texttt{Norm}(\theta_\sharp^\star \alpha, \theta_\sharp^\star \beta, f_\theta, g_\theta, \rho_1, \rho_2)$
      $(r_\theta, s_\theta) \leftarrow \texttt{SlicedDual}(\alpha_\theta, \beta_\theta)$
      $f_\theta \leftarrow (1 - \gamma_t) f_\theta + \gamma_t r_\theta$  (FWStep)
      $g_\theta \leftarrow (1 - \gamma_t) g_\theta + \gamma_t s_\theta$  (FWStep)
    **end for**
  **end for**
  Return $\text{SUOT}(\alpha,\beta)$, $(f_\theta, g_\theta)$ as in (7)

---

**Algorithm 3** – USOT

**Input:** $\alpha$, $\beta$, $F$, $(\theta_k)_{k=1}^K$, $\rho_1$, $\rho_2$
**Output:** $\text{USOT}(\alpha,\beta)$, $(f_{avg}, g_{avg})$

  $(f_\theta, g_\theta, f_{avg}, g_{avg}) \leftarrow (0,0,0,0)$
  **for** $t = 0, 1, \ldots, F - 1$ **do**
    **for** $\theta \in (\theta_k)_{k=1}^K$ **do**
      $(\pi_1, \pi_2) \leftarrow \texttt{Norm}(\alpha, \beta, f_{avg}, g_{avg}, \rho_1, \rho_2)$
      $(r_\theta, s_\theta) \leftarrow \texttt{SlicedDual}(\theta_\sharp^\star \pi_1, \theta_\sharp^\star \pi_2)$
    **end for**
    $(r_{avg}, s_{avg}) \leftarrow \frac{1}{K} \sum_{k=1}^K r_{\theta_k}, \frac{1}{K} \sum_{k=1}^K s_{\theta_k}$
    $f_{avg} \leftarrow (1 - \gamma_t) f_{avg} + \gamma_t r_{avg}$  (FWStep)
    $g_{avg} \leftarrow (1 - \gamma_t) g_{avg} + \gamma_t s_{avg}$  (FWStep)
  **end for**
  Return $\text{USOT}(\alpha,\beta)$, $(f_{avg}, g_{avg})$ as in (10)

---

Each iteration requires computing the dual potentials of a sliced OT problem, which is non-trivial: previous implementations related to sliced OT only output the value of the loss, $\text{SOT}(\alpha,\beta)$, typically in the context of training generative models (Deshpande et al., 2019; Nguyen et al., 2020). We thus design two novel implementations in PyTorch (Paszke et al., 2019) to compute the dual potentials of sliced OT. The first one leverages that the gradient of $\text{OT}(\alpha,\beta)$ w.r.t. $(\alpha,\beta)$ are optimal $(f,g)$, which allows to backpropagate $\text{OT}(\theta_\sharp^\star \alpha, \theta_\sharp^\star \beta)$ w.r.t. $(\alpha,\beta)$ to obtain $(r_\theta, s_\theta)$. The second one computes them in parallel on GPUs using their closed form, which to the best of our knowledge, is a new sliced algorithm. We call $\texttt{SlicedDual}(\alpha,\beta)$ the step returning optimal $(r_\theta, s_\theta)$ solving $\text{OT}(\theta_\sharp^\star \alpha, \theta_\sharp^\star \beta)$ for all $\theta \in \text{supp}(\hat{\boldsymbol{\sigma}}_K)$, and refer to Appendix B.3 for the algorithms.

Building on Proposition 4.1 and the discussion above, we develop the FW methodology to compute $\text{SUOT}(\alpha,\beta)$ and detail it in Algorithm 2. Next, we derive the FW iterates for $\text{USOT}(\alpha,\beta)$.

**Proposition 4.2** (Frank-Wolfe iterations for USOT). *Let $(\alpha,\beta) \in \mathcal{M}_+(\mathbb{R}^d) \times \mathcal{M}_+(\mathbb{R}^d)$ and consider solving (14) with FW. Assume that $(\varphi_1^\circ, \varphi_2^\circ)$ are smooth and strictly concave. Given current iterates $(f_\theta^t, g_\theta^t)_{\theta \in \text{supp}(\hat{\boldsymbol{\sigma}}_K)} \in \mathcal{E}$, the solutions of the linear oracle $(r_\theta^t, s_\theta^t)_{\theta \in \text{supp}(\hat{\boldsymbol{\sigma}}_K)}$ are the dual potentials of $\text{SOT}(\bar{\alpha}^t, \bar{\beta}^t)$, where $(\bar{\alpha}_t, \bar{\beta}_t)$ are measures given by $\bar{\alpha}_t = \nabla \varphi^\circ(f_{avg} + \lambda^\star(f_{avg}, g_{avg}))\alpha$ and $\bar{\beta}_t = \nabla \varphi^\circ(g_{avg} - \lambda^\star(f_{avg}, g_{avg}))\beta$, with $f_{avg}(x) \triangleq \int_{\mathbb{S}^{d-1}} f_\theta^t(\theta^\star(x)) \mathrm{d}\hat{\boldsymbol{\sigma}}_K(\theta)$, $g_{avg}(y) \triangleq \int_{\mathbb{S}^{d-1}} g_\theta^t(\theta^\star(y)) \mathrm{d}\hat{\boldsymbol{\sigma}}_K(\theta)$.*

The resulting FW methodology, detailed in Algorithm 3, also leverages the $\texttt{Norm}$ and $\texttt{SlicedDual}$ routines. The key difference from $\text{SUOT}(\alpha,\beta)$ is in where the integral over $\theta \in \text{supp}(\hat{\boldsymbol{\sigma}}_K)$ is performed, leading to a different balanced sliced OT problem to solve.

**Marginals of UOT/USOT.** The optimal primal marginals of UOT and USOT are geometric normalizations of inputs $(\alpha,\beta)$ with discarded outliers. Their computation involves the $\texttt{Norm}$ routine detailed in Algorithm 1, using optimal dual potentials. This is how we compute marginals in Figure 1 and in the experiments of Section 5: see Appendix B.4 for more details.

### 4.3 Convergence Properties and Complexity

**Convergence and stochastic Frank-Wolfe.** Our theoretical setting verifies the assumptions of (Lacoste-Julien & Jaggi, 2015, Theorem 8), thus ensuring fast convergence of our methods. The number of FW iterations needed to converge remains low in our experiments. We give in Appendix B.5 empirical evidences that few iterations of FW ($F \leq 20$) suffice to reach numerical precision.

Formally, the preceding algorithms assume that the functional $\mathcal{H}$ is given through integrals over the hypersphere, describing the set of all possible directions $\theta$. However, in practice, SOT is computed by Monte-Carlo approximations, *i.e.*, drawing a fixed number $K$ of directions $(\theta_k)_{k=1}^K$ and solving independently the different 1D OT problems. In the specific case of SUOT, this does not change much: $K$ FW procedures are ran

Table 1: Accuracy on document classification

| | BBCSport | | Goodreads | | |
| --- | --- | --- | --- | --- | --- |
| | Acc | t ($\cdot 10^{-3}$s) | Acc (genre) | Acc (like) | t ($\cdot 10^{-3}$s) |
| OT | 94.55 | $3.12_{\pm 1.61}$ | 55.22 | 71.00 | $440.30_{\pm 250}$ |
| UOT | 96.73 | $243.39_{\pm 9.24}$ | - | - | - |
| SinkhUOT | 95.45 | $46.22_{\pm 2.17}$ | 53.55 | 67.81 | $2021.68_{\pm 356}$ |
| SOT | $89.39_{\pm 0.76}$ | $1.80_{\pm 0.22}$ | $50.09_{\pm 0.51}$ | $65.60_{\pm 0.20}$ | $4.49_{\pm 1.44}$ |
| SUOT | $90.12_{\pm 0.15}$ | $13.9_{\pm 1.21}$ | $50.15_{\pm 0.04}$ | $66.72_{\pm 0.38}$ | $14.32_{\pm 0.95}$ |
| USOT | $93.52_{\pm 0.04}$ | $14.37_{\pm 1.29}$ | $52.67_{\pm 0.62}$ | $67.78_{\pm 0.39}$ | $14.45_{\pm 0.88}$ |
| SUOT (CV on $\rho$) | $90.00_{\pm 0.59}$ | - | $49.67_{\pm 0.79}$ | $66.43_{\pm 0.44}$ | - |
| USOT (CV on $\rho$) | $92.61_{\pm 0.55}$ | - | $52.06_{\pm 7.20}$ | $66.61_{\pm 0.72}$ | - |

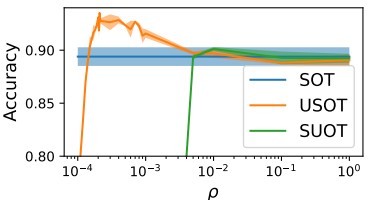

Figure 2: Ablation on BBCSport of $\rho$.

independently (eventually in parallel) over the fixed set of directions. The case of USOT relies on a global FW scheme, where $f_{avg}, g_{avg}$ are computed w.r.t. a fixed distribution $\hat{\boldsymbol{\sigma}}_K = (1/K) \sum_{k=1}^K \delta_{\theta_k}$. This empirical distribution of directions can be considered fixed throughout the FW iterations, or can be drawn independently for each iteration of the FW procedure. This actually corresponds to a *Stochastic FW* algorithm, which also converges as our setting verifies the assumptions of (Hazan & Luo, 2016, Theorem 3). We call this procedure *Stochastic* USOT, which corresponds to Algorithm 3 except that $(\theta_k)_{k=1}^K$ are sampled at each iteration. Since this procedure performs well in our experiments (*e.g.*, Table 4) and $\mathbb{E}_{\theta_k \sim \boldsymbol{\sigma}}[\hat{\boldsymbol{\sigma}}_K] = \boldsymbol{\sigma}$, this suggests the dual in Theorem 3.9 holds for $\boldsymbol{\sigma}$.

**Algorithmic complexity.** FW algorithms and its variants have been widely studied theoretically. Computing `SlicedDual` has theoretically a complexity $\mathcal{O}(KN \log N)$, where $N$ is the number of samples, and $K$ the number of projections of $\hat{\boldsymbol{\sigma}}_K$. However, we note that the sorting operation, which yields the super linear complexity, can be computed once for all FW iterations. Consequently, the overall complexity of SUOT and USOT is thus $\mathcal{O}(KN \log N + FN)$, where $F$ is the number of FW iterations needed to reach convergence, with a $\mathcal{O}(N)$ complexity. Thus, our formulation enjoy a similar complexity than SOT, which is particularly appealing. However, *Stochastic* USOT is more costly, as each iteration requires sorting data projected along newly-sampled $(\theta_k)_{k=1}^K$. Its complexity is therefore $\mathcal{O}(KFN \log N)$. We finally note that due to the independent nature of the treatments of every projections, computing both `Norm` and `SlicedDual` operations can be done in parallel, leveraging GPU computations when available.

**Extension to non-Euclidean settings.** Interestingly, our algorithms offer great modularity, in the sense they can easily be used to compute unbalanced versions of existing variants of SOT. Indeed, while such variants differ in the one-dimensional representations of $\alpha$ and $\beta$ they use, they all consist in solving 1D OT problems to compare $\alpha$ and $\beta$, which our FW strategy can solve. To illustrate this point, we combined our FW routine with hyperbolic SOT (Bonet et al., 2023c) to compare measures supported on hyperbolic spaces: see Appendix C.3.

## 5 Experiments

This section presents a set of numerical experiments, which illustrate the effectiveness, robustness and computational efficiency of USOT[1]. We first showcase the benefit of USOT over SUOT and SOT on a document classification task. Then, we consider experiments in very large scale settings such as color transfer on every pixels and the computation of barycenters of geophysical datasets.

### 5.1 Document classification

We first consider a document classification problem (Kusner et al., 2015). Documents are represented as distributions of words embedded with *word2vec* (Mikolov et al., 2013) in dimension $d = 300$. Let $D_k$ be the $k$-th document and $x_1^k, \ldots, x_{n_k}^k \in \mathbb{R}^d$ be the set of words in $D_k$. Then, $D_k = \sum_{i=1}^{n_k} w_i^k \delta_{x_i^k}$ where $w_i^k$ is the frequency of $x_i^k$ in $D_k$ normalized s.t. $\sum_{i=1}^{n_k} w_i^k = 1$. Given a loss function L, the document classification task is solved by computing the matrix $\left(\mathrm{L}(D_k, D_\ell)\right)_{k,\ell}$, then using a k-nearest neighbor classifier. The aim

---

[1]The code is available at https://github.com/clbonet/Slicing_Unbalanced_Optimal_Transport.

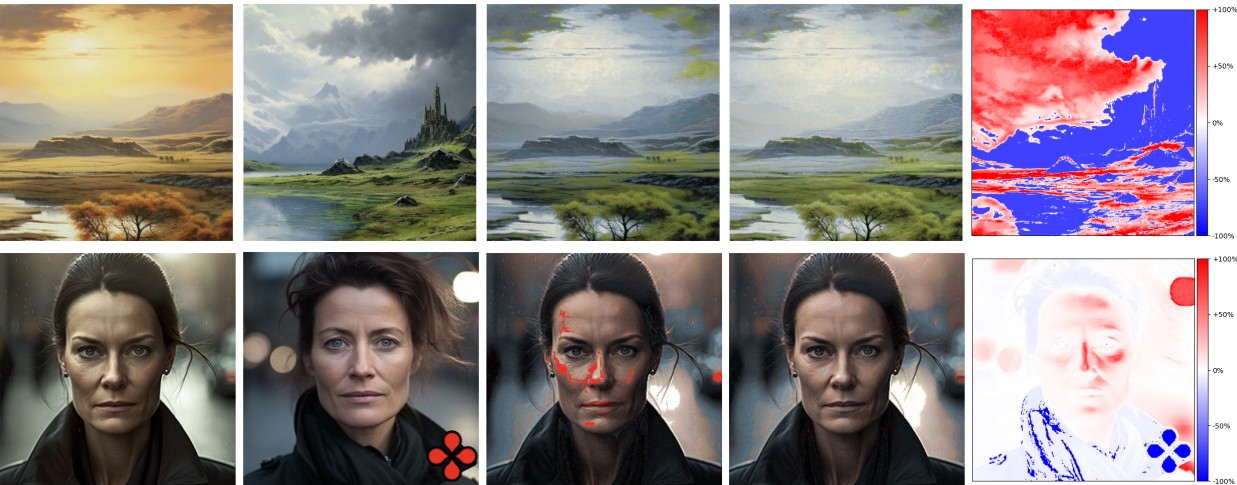

Figure 3: **Color transfer** between a source and a target image (*first and second columns*). We compare SOT gradient flows operated in the color space (*third column*) and the same procedure with a reweighing of the distributions by USOT (*fourth column*). The last column shows a percentage of mass change given by USOT, *i.e.*, $\frac{(\pi_2^* - \beta)}{\beta}$, where red indicates mass creation and blue mass destruction.

of this experiment is to show that by discarding possible outliers using a well chosen parameter $\rho$, USOT is able to outperform SOT and SUOT on this task. Since a word typically appears several times in a document, the measures are not uniform and sliced partial OT (Bonneel & Coeurjolly, 2019; Bai et al., 2023) cannot be used in this setting. We detail in Appendix C additional experiments without normalizing histograms to compare with (Bai et al., 2023) (*i.e.*, $\sum_{i=1}^{n_k} w_i^k$ is the sentence length). We consider the BBCSport dataset (Kusner et al., 2015), a standard benchmark with small documents for which OT can be used effectively, and the Goodreads dataset (Maharjan et al., 2017) on two tasks (genre and likability predictions), a dataset with large-scale documents for which the computational burden of performing OT and UOT is substantial. We report on Table 1 the accuracy and average runtimes of OT, UOT computed with the majorization minimization algorithm (Chapel et al., 2021) or approximated with the Sinkhorn algorithm (SinkhUOT) (Pham et al., 2020), as well as SOT, SUOT and USOT. All the benchmark methods are computed using the Python OT library (Flamary et al., 2021) on a Nvidia Tesla V100 GPU. For sliced methods, we average over 3 computations of the loss matrix and report the standard deviation in Table 1. The number of neighbors was selected via cross validation. The results for UOT, SinkhUOT, SUOT and USOT are reported for $\rho$ yielding the best accuracy among a grid (see Appendix C.1 for more details), and we display an ablation of this parameter on the BBCSport dataset in Figure 2. We also add on Table 1 the results obtained with a cross validation (CV) on $\rho$ for USOT and SUOT. Our findings demonstrate that our approaches surpass SOT in performance, incurring only a minor computational overhead. Moreover, our methods closely rival the performance of OT, while being 40 times faster on large-scale datasets. This highlight their practical significance, particularly when OT is computationally unfeasible.

## 5.2 Color transfer

Color transfer is a long-standing problem in OT, which dates back to the seminal work of Rabin et al. (2010). It consists in aligning the color distributions of two images. While previous works, *e.g.* (Ferradans et al., 2013; Bonneel et al., 2016), considered color palettes to deal with the complexity of OT, we illustrate the scalability of our methods by considering here the full distributions of pixels within images, in a way similar to (Bonneel & Coeurjolly, 2019). We express the color transfer as a gradient flow, where every pixel is a sample in the 3D RGB color space. Formally, let $\alpha(t) = \frac{1}{N} \sum_{i=1}^N \delta_{x_i(t)}, \beta = \frac{1}{M} \sum_{j=1}^M \delta_{y_j}$, where $\alpha$ (resp. $\beta$) represents the color distribution of the source (resp. target) image. The SOT gradient flow performing color transfer consists in iterating the following scheme: $X(t+1) = X(t) - \gamma \nabla_X \text{SOT}(\alpha(t), \beta)$, where $\alpha(t)$ is the color distribution of the source image at iterations $t$, supported by pixels from $X(t)$. One of the major

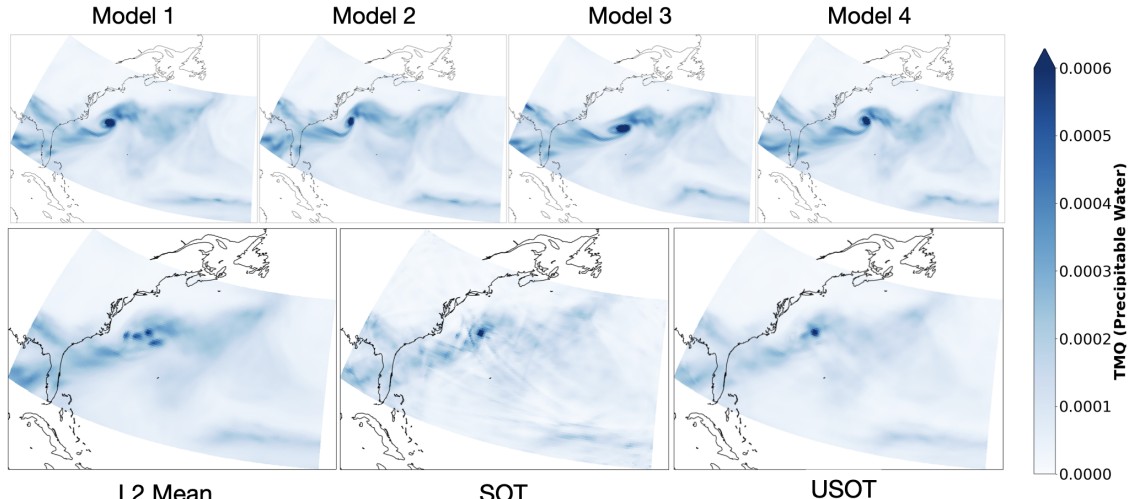

Figure 4: **Barycenter of geophysical data**. (*First row*) Simulated output of 4 different climate models depicting different scenarios for the evolution of a tropical cyclone (*Second row*) Results of different aggregation strategies.

problem is color bleeding: potential color artefacts are likely to appear since the object proportions between images are likely to differ. We propose to correct this issue in a two steps procedure, relying on USOT: *i)* we first obtain optimal marginals $(\pi_1^*, \pi_2^*)$ by solving (9) and using the `Norm` routine, then *ii)* we solve the classical SOT gradient flow with the measures $(\pi_1^*, \pi_2^*)$.

We present results on $300 \times 300$ images produced by the generative model Midjourney (Mid, 2023). The images correspond to two landscapes and photo-realistic portraits of two women (see first and second column of Figure 3). For every results, we iterate the gradient flow for 100 iterations, and the learning rate $\gamma$ is set to $10^{-2}$. The computation of the final result is produced in less than one minute with a commodity GPU, while it is out of reach for OT or UOT solvers on this distributions size ($90K$ pixels). The third column shows the color transfer using SOT. It reveals instances of color bleeding: green clouds appear in the sky of landscape scenes, and a red superimposed logo results in unwanted red pixels in the portrait. Contrasting, the fourth column displays the outcomes achieved through our approach, effectively addressing the color bleeding issue. We chose $\rho_1 = 10^4$ and $\rho_2 = 0.02$ for this task and we observe the resulting change in mass distribution on the target image in the last column of Figure 3. It provides insightful indications of the discarded colors (the darkened landscape in the first case and the removal of the logo in the second), which is only possible when all the pixels are considered in the transfer.

### 5.3 Barycenter of geophysical data

OT barycenters are an important topic of interest (Le et al., 2021) for their ability to capture mass changes and spatial deformations over several reference measures. In order to compute barycenters under the USOT geometry on a fixed grid, we employ a mirror-descent strategy similar to (Cuturi & Doucet, 2014a, Algorithm (1)) and described more in depth in Appendix C. We compute unbalanced sliced OT barycenter for climate model data. Ensembles of multiple models are commonly employed to reduce biases and evaluate uncertainties in climate projections (Sanderson et al., 2015; Thao et al., 2022). The commonly used Multi-Model Mean approach assumes models are centered around true values and averages the ensemble with equal or varying weights. However, spatial averaging may fail in capturing specific characteristics of the physical system at stake, and we propose to use USOT barycenter instead. We use the ClimateNet dataset (Prabhat et al., 2021), and more specifically the TMQ (precipitable water) indicator. The ClimateNet dataset is a human-expert-labeled curated dataset which captures tropical cyclones (TCs), among other things. To simulate the output of several climate models, we take a specific instant (first date of 2011) and apply the elastic deformation from TorchVision (Paszke et al., 2019) in an area close to the eastern part of the U.S.A. As a result, we obtain 4 different TCs, as shown in the first row of Figure 4. The classical L2 spatial mean is

displayed on the second row of Figure 4 and reveals 4 different TCs centers/modes, which is undesirable. As the total TMQ mass in the considered zone varies between the different models, a direct application of SOT is impossible, or requires a normalization of the mass that has undesired effect as can be seen on the second row. Finally, we show the result of the USOT barycenter with $\rho_1 = 1e1$ (related to the data) and $\rho_2 = 1e4$ (related to the barycenter). This barycenter has only one apparent mode, which is the expected behaviour. The considered measures have a size of $100 \times 200$, and we run the barycenter algorithm for 500 iterations (with $K = 64$ projections), which takes 3 minutes on a commodity GPU. UOT barycenters for this size of problems are intractable, and to the best of our knowledge, our experiment is the very first instance where unbalanced OT barycenters can be computed on such a large scale.

## 6 Conclusion

We proposed two losses merging unbalanced and sliced OT, with theoretical guarantees and an efficient and modular Frank-Wolfe algorithm. We illustrate the performance improvement over SOT on various experiments, and described novel applications of unbalanced OT barycenters of positive measures, with a new case study on geophysical data. These novel results and algorithms pave the way to numerous new applications of sliced variants of OT, and we believe that our contributions will motivate practitioners to further explore their use in general ML applications, without the cumbersome task of pre-processing probability measures.

### Acknowledgments

We thank the anonymous reviewers for their valuable comments. KF was supported by NSERC Discovery grant (RGPIN-2019-06512) and a Samsung grant. CB was supported by project DynaLearn from Labex CominLabs and Région Bretagne ARED DLearnMe, and by the ANR PEPR PDE-AI. NC was supported by the ANR AI Chair OTTOPIA ANR-20-CHIA-0030

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

# A Postponed proofs for Section 3

## A.1 Existence of minimizers: Proof of Proposition 3.2 and Proposition 3.7

We provide the formal statement and detailed proof on the existence of a solution for both SUOT and USOT, as mentioned in Section 3.

**Proposition A.1.** *(Existence of minimizers) Assume that $C_1$ is lower-semicontinuous and that either (i) $\varphi'_{1,\infty} = \varphi'_{2,\infty} = +\infty$, or (ii) $C_1$ has compact sublevels on $\mathbb{R} \times \mathbb{R}$ and $\varphi'_{1,\infty} + \varphi'_{2,\infty} + \inf C_1 > 0$. Then the solution of $\mathrm{SUOT}(\alpha, \beta)$ and $\mathrm{USOT}(\alpha, \beta)$ exist, i.e., the infimum in (6) and (9) is attained. More precisely, there exists $(\pi_1, \pi_2)$ which attains the infimum for $\mathrm{USOT}(\alpha, \beta)$ (see Equation 9). Concerning $\mathrm{SUOT}(\alpha, \beta)$, there exists for any $\theta \in \mathrm{supp}(\boldsymbol{\sigma})$ a plan $\pi_\theta$ attaining the infimum in $\mathrm{UOT}(\theta^\star_\sharp \alpha, \theta^\star_\sharp \beta)$ (see Equation 2).*

*Proof.* We leverage (Liero et al., 2018, Theorem 3.3) to prove this proposition. In the setting of SUOT, if such assumptions (i) or (ii) are satisfied for $(\alpha, \beta)$, then they also hold for $(\theta^\star_\sharp \alpha, \theta^\star_\sharp \beta)$ for any $\theta \in \mathbb{S}^{d-1}$. Hence, $\mathrm{UOT}(\theta^\star_\sharp \alpha, \theta^\star_\sharp \beta)$ admits a solution $\pi^\theta$.

Concerning USOT, note that one necessarily has $m(\pi_1) = m(\pi_2)$, otherwise $\mathrm{SOT}(\pi_1, \pi_2) = +\infty$. From (Liero et al., 2018, Equation (3.10)), for any admissible $(\pi_1, \pi_2, \pi)$, one has

$$\mathrm{USOT}(\alpha, \beta) \geq m(\pi) \inf C_1 + m(\alpha)\varphi_1\left(\tfrac{m(\pi)}{m(\alpha)}\right) + m(\beta)\varphi_2\left(\tfrac{m(\pi)}{m(\beta)}\right).$$

In both settings the above bounds implies coercivity of the functional of USOT w.r.t. the masses of the measures $(\pi_1, \pi_2, \pi)$. Thus there exists $M > 0$ such that $m(\pi_1) = m(\pi_2) = m(\pi) < M$, otherwise $\mathrm{USOT}(\alpha, \beta) = +\infty$. By the Banach-Alaoglu theorem, the set of bounded measures $(\pi_1, \pi_2)$ is compact, and the set of plans $\pi$ with such marginals is also compact because $\mathbb{R}^d$ is Polish and $C_1$ is lower-semicontinuous (Santambrogio, 2015, Theorem 1.7). Because the functional of USOT is lower-semicontinuous in $(\pi_1, \pi_2, \pi)$ and we can restrict optimization over a compact set, we have existence of minimizers for USOT by standard proofs of calculus of variations. $\square$

## A.2 Strong duality: Proof of Theorem 3.5 and Theorem 3.9

Note that the result for SUOT (Theorem 3.5) is proved in Lemma A.4. Thus we focus on the proof of duality for USOT.

*Proof of Theorem 3.9.* We start from the definition of USOT, reformulate it to apply the strong duality result of Proposition A.2 and obtain our reformulation. We first have that

$$\mathrm{USOT}(\alpha, \beta) = \inf_{(\pi_1, \pi_2) \in \mathcal{M}_+(\mathbb{R}^d)^2} \left\{ \mathrm{SOT}(\pi_1, \pi_2) + \mathrm{D}_{\varphi_1}(\pi_1 | \alpha) + \mathrm{D}_{\varphi_2}(\pi_2 | \beta) \right\},$$

$$= \inf_{(\pi_1, \pi_2) \in \mathcal{M}_+(\mathbb{R}^d)^2} \left\{ \int_{\mathbb{S}^{d-1}} \left[ \sup_{f_\theta \oplus g_\theta \leq C_1} \int f_\theta \mathrm{d}(\theta^\star_\sharp \pi_1) + \int g_\theta \mathrm{d}(\theta^\star_\sharp \pi_2) \right] \mathrm{d}\hat{\boldsymbol{\sigma}}_K(\theta) \right.$$

$$+ \sup_{\tilde{f} \in \mathcal{E}(\mathbb{R}^d)} \int \varphi_1^\circ(\tilde{f}(x)) \mathrm{d}\alpha(x) - \int \tilde{f}(x) \mathrm{d}\pi_1(x)$$

$$\left. + \sup_{\tilde{g} \in \mathcal{E}(\mathbb{R}^d)} \int \varphi_2^\circ(\tilde{g}(y)) \mathrm{d}\beta(y) - \int \tilde{g}(y) \mathrm{d}\pi_2(y) \right\},$$

$$= \inf_{(\pi_1, \pi_2) \in \mathcal{M}_+(\mathbb{R}^d)^2} \left\{ \sup_{f_\theta \oplus g_\theta \leq C_1} \int_{\mathbb{S}^{d-1}} \left[ \int f_\theta \mathrm{d}(\theta^\star_\sharp \pi_1) + \int g_\theta \mathrm{d}(\theta^\star_\sharp \pi_2) \right] \mathrm{d}\hat{\boldsymbol{\sigma}}_K(\theta) \right.$$

$$+ \sup_{\tilde{f} \in \mathcal{E}(\mathbb{R}^d)} \int \varphi_1^\circ(\tilde{f}(x)) \mathrm{d}\alpha(x) - \int \tilde{f}(x) \mathrm{d}\pi_1(x)$$

$$\left. + \sup_{\tilde{g} \in \mathcal{E}(\mathbb{R}^d)} \int \varphi_2^\circ(\tilde{g}(y)) \mathrm{d}\beta(y) - \int \tilde{g}(y) \mathrm{d}\pi_2(y) \right\},$$

where $\mathcal{E}(\mathbb{R}^d)$ denotes a set of lower-semicontinuous functions, and the last equality holds thanks to Lemma A.3.

We focus now on verifying that Proposition A.2 holds, so that we can swap the infimum and the supremum. Define the functional

$$
\mathcal{L}((\pi_1, \pi_2), ((f_\theta)_\theta, (g_\theta)_\theta, \tilde{f}, \tilde{g})) \triangleq \int_{\mathbb{S}^{d-1}} \left[ \int f_\theta \mathrm{d}(\theta^\star_\sharp \pi_1) + \int g_\theta \mathrm{d}(\theta^\star_\sharp \pi_2) \right] \mathrm{d}\hat{\boldsymbol{\sigma}}_K(\theta)
$$
$$
+ \int \varphi_1^\circ(\tilde{f}(x)) \mathrm{d}\alpha(x) - \int \tilde{f}(x) \mathrm{d}\pi_1(x)
$$
$$
+ \int \varphi_2^\circ(\tilde{g}(y)) \mathrm{d}\beta(y) - \int \tilde{g}(y) \mathrm{d}\pi_2(y) \, .
$$

One has that,

- For any $((f_\theta)_\theta, (g_\theta)_\theta, \tilde{f}, \tilde{g})$, $\mathcal{L}$ is linear (thus convex) and lower-semicontinuous.

- For any $(\pi_1, \pi_2)$, $\mathcal{L}$ is concave in $((f_\theta)_\theta, (g_\theta)_\theta, \tilde{f}, \tilde{g})$ because $\varphi_i^\circ$ is concave and thus $\mathcal{L}$ is a sum of linear or concave functions.

Furthermore, since we assumed that $0 \in \mathrm{dom}(\varphi)$, then

$$
\sup_{((f_\theta)_\theta, (g_\theta)_\theta, \tilde{f}, \tilde{g})} \inf_{(\pi_1, \pi_2) \in \mathcal{M}_+(\mathbb{R}^d)^2} \mathcal{L} \leq \mathrm{USOT}(\alpha, \beta) \leq \varphi_1(0) m(\alpha) + \varphi_2(0) m(\beta) \, ,
$$

because the marginals $(\pi_1, \pi_2) = (0, 0)$ are admissible and suboptimal. If we consider instead that $(m(\alpha), m(\beta)) \in \mathrm{dom}(\varphi)$, then we take the marginals $\pi_1 = \alpha/m(\alpha)$ and $\pi_2 = \beta/m(\beta)$, which yields an upper-bound by $m(\alpha)\varphi_1(\frac{1}{m(\alpha)}) + m(\beta)\varphi_2(\frac{1}{m(\beta)})$. Then we consider an anchor dual point $b^\star = ((f_\theta)_\theta, (g_\theta)_\theta, \tilde{f}, \tilde{g})$ to bound $\mathcal{L}$ over a compact set. We take $f_\theta = 0$, $g_\theta = 0$, which are always admissible since we take $\mathrm{C}_1(x, y) \geq 0$. Then, since we assume there exists $p_i \leq 0$ in $\mathrm{dom}(\varphi_i^*)$, we take $\tilde{f} = p_1$ and $\tilde{g} = p_2$. For these potentials one has:

$$
\mathcal{L}((\pi_1, \pi_2), b^\star) = \varphi_1^\circ(p_1) m(\alpha) - p_1 m(\pi_1) + \varphi_2^\circ(p_2) m(\alpha) - p_2 m(\pi_2).
$$

Note that the functional at this point only depends on the masses of the marginals $(\pi_1, \pi_2)$. Since $(p_1, p_2) \geq 0$ the set of $(\pi_1, \pi_2)$ such that $\mathcal{L}((\pi_1, \pi_2), b^\star) \leq \varphi_1(0) m(\alpha) + \varphi_2(0) m(\beta)$ is non-empty (at least in a neighbour-hood of $(\pi_1, \pi_2) = (0, 0)$, and that $(m(\pi_1), m(\pi_2))$ are uniformly bounded by some constant $M > 0$. By the Banach-Alaoglu theorem, such set of measures is compact for the weak* topology.

Therefore, Proposition A.2 holds and we have strong duality, *i.e.*,

$$
\mathrm{USOT}(\alpha, \beta) = \sup_{\left\{ \begin{array}{l} f_\theta \oplus g_\theta \leq \mathrm{C}_1 \\ (\tilde{f}, \tilde{g}) \in \mathcal{E}(\mathbb{R}^d) \end{array} \right\}} \inf_{(\pi_1, \pi_2) \in \mathcal{M}_+(\mathbb{R}^d)^2} \mathcal{L}((\pi_1, \pi_2), ((f_\theta)_\theta, (g_\theta)_\theta, \tilde{f}, \tilde{g})).
$$

To achieve the proof, note that taking the infimum in $(\pi_1, \pi_2)$ (for fixed dual variables) reads

$$
\inf_{\pi_1, \pi_2 \geq 0} \int \left( \int_{\mathbb{S}^{d-1}} f_\theta(\theta^\star(x)) \mathrm{d}\hat{\boldsymbol{\sigma}}_K(\theta) \right) \mathrm{d}\pi_1(x) - \int \tilde{f}(x) \mathrm{d}\pi_1(x)
$$
$$
+ \int \left( \int_{\mathbb{S}^{d-1}} g_\theta(\theta^\star(y)) \mathrm{d}\hat{\boldsymbol{\sigma}}_K(\theta) \right) \mathrm{d}\pi_2(y) - \int \tilde{g}(y) \mathrm{d}\pi_2(y).
$$

Note that we applied Fubini's theorem here, which holds here because all measures have compact support, thus all quantities are finite. It allows to rephrase the minimization over $\pi_1, \pi_2 \geq 0$ as the following constraint

$$
\int_{\mathbb{S}^{d-1}} f_\theta(\theta^\star(x)) \mathrm{d}\hat{\boldsymbol{\sigma}}_K(\theta) \geq \tilde{f}(x), \qquad \int_{\mathbb{S}^{d-1}} g_\theta(\theta^\star(y)) \mathrm{d}\hat{\boldsymbol{\sigma}}_K(\theta) \geq \tilde{g}(y),
$$

otherwise the infimum is $-\infty$. However, the function $\varphi^\circ$ is non-decreasing (see (Séjourné et al., 2019, Proposition 2)). Thus the maximization in $(\tilde{f}, \tilde{g})$ is optimal when the above inequality is actually an equality, *i.e.*,

$$\int_{\mathbb{S}^{d-1}} f_\theta(\theta^\star(x)) \mathrm{d}\hat{\boldsymbol{\sigma}}_K(\theta) = \tilde{f}(x), \qquad \int_{\mathbb{S}^{d-1}} g_\theta(\theta^\star(y)) \mathrm{d}\hat{\boldsymbol{\sigma}}_K(\theta) = \tilde{g}(y).$$

Plugging the above relation in the functional $\mathcal{L}$ yields the desired result on the dual of USOT and ends the proof.

$\square$

We mention a strong duality result which is very general and which we use in the proof of Theorem 3.9. This result is taken from (Liero et al., 2018, Theorem 2.4) which itself takes it from (Simons, 2006).

**Proposition A.2.** *(Liero et al., 2018, Theorem 2.4) Consider two sets $A$ and $B$ be nonempty convex sets of some vector spaces. Assume $A$ is endowed with a Hausdorff topology. Let $L : A \times B \to \mathbb{R}$ be a function such that*

1. *$a \mapsto L(a, b)$ is convex and lower-semicontinuous on $A$, for every $b \in B$*

2. *$b \mapsto L(a, b)$ is concave on $B$, for every $a \in A$.*

*If there exists $b_\star \in B$ and $\kappa > \sup_{b \in B} \inf_{a \in A} L(a, b)$ such that the set $\{a \in A, L(a, b_\star) < \kappa\}$ is compact in $A$, then*

$$\inf_{a \in A} \sup_{b \in B} L(a, b) = \sup_{b \in B} \inf_{a \in A} L(a, b)$$

We also consider the following to swap the supremum in the integral which defines sliced-UOT (and in particular sliced-OT). In what follows we note sliced potentials as functions $f_\theta(z)$ with $(\theta, z) \in \mathbb{S}^{d-1} \times \mathbb{R}$, such that

$$\mathrm{SUOT}(\alpha, \beta) = \int_{\mathbb{S}^{d-1}} \left[ \sup_{f_\theta \oplus g_\theta \leq \mathrm{C}_1} \int \varphi^\circ \circ f_\theta \mathrm{d}(\theta^\star_\sharp \alpha) + \int \varphi^\circ \circ g_\theta \mathrm{d}(\theta^\star_\sharp \beta) \right] \mathrm{d}\hat{\boldsymbol{\sigma}}_K(\theta).$$

Note that with the above definition, $z \mapsto f_\theta(z)$ is continuous for any $\theta$, but $\theta \mapsto f_\theta(z)$ is only $\hat{\boldsymbol{\sigma}}_K$-measurable.

**Lemma A.3.** *Consider two sets $X$ and $Y$, a measure $\sigma$ such that $\sigma(X) < +\infty$. Assume $Y$ is compact. Consider a function $\mathcal{F} : X \times Y \to \mathbb{R}$. Assume there exists a sequence $(y_n)$ in $Y$ such that $\mathcal{F}(\cdot, y_n) \to \sup_{y \in Y} \mathcal{F}(\cdot, y)$ uniformly. Then one has*

$$\sup_{y \in Y} \int_X \mathcal{F}(x, y) \mathrm{d}\sigma(x) = \int_X \sup_{y \in Y} \mathcal{F}(x, y) \mathrm{d}\sigma(x).$$

*Proof.* Define $\mathcal{G}(x) = \sup_{y \in Y} \mathcal{F}(x, y)$ and $\mathcal{H}(x, y) \triangleq \mathcal{G}(x) - \mathcal{F}(x, y)$. One has $\mathcal{H} \geq 0$ by definition, and the desired equality can be rewritten as

$$\sup_{y \in Y} \int_X \mathcal{F}(x, y) \mathrm{d}\sigma(x) = \int_X \sup_{y \in Y} \mathcal{F}(x, y) \mathrm{d}\sigma(x)$$
$$\Leftrightarrow \inf_{y \in Y} \int_X \mathcal{H}(x, y) \mathrm{d}\sigma(x) = 0.$$

Since the integral involving $\mathcal{H}$ is non-negative, the infimum is zero if and only if we have a sequence $(y_n)$ such that $\int_X \mathcal{H}(\cdot, y_n) \mathrm{d}\sigma \to 0$. By assumption, one has $\mathcal{F}(\cdot, y_n) \to \sup_{y \in Y} \mathcal{F}(\cdot, y)$ uniformly, *i.e.*, $||\mathcal{H}(\cdot, y_n)||_\infty \to 0$. This implies thanks to Holder's inequality that

$$0 \leq \int_X \mathcal{H}(\cdot, y_n) \mathrm{d}\sigma \leq \sigma(X) ||\mathcal{H}(\cdot, y_n)||_\infty$$

Thus by assumption one has $\int_X \mathcal{F}(\cdot, y_n)\mathrm{d}\sigma \to \int_X \mathcal{G}\mathrm{d}\sigma$, which indeed means that we have the desired permutation between supremum and integral.

$\square$

**Lemma A.4.** *Let $p \in [1, +\infty)$ and assume that $\mathrm{C}_1(x, y) = |x - y|^p$. Consider two positive measures $(\alpha, \beta)$ with compact support. Assume that the measure $\hat{\boldsymbol{\sigma}}_K$ is discrete, i.e., $\hat{\boldsymbol{\sigma}}_K = \frac{1}{K}\sum_{i=1}^{K}\delta_{\theta_i}$ with $\theta_i \in \mathbb{S}^{d-1}$, $i = 1, \ldots, n$. Then, one can swap the integral over the sphere and the supremum in the dual formulation of* SUOT, *such that*

$$\mathrm{SUOT}(\alpha, \beta) = \sup_{f_\theta \oplus g_\theta \leq \mathrm{C}_1} \int_{\mathbb{S}^{d-1}} \Big[ \int \varphi^\circ \circ f_\theta \mathrm{d}(\theta_\sharp^\star \alpha) + \int \varphi^\circ \circ g_\theta \mathrm{d}(\theta_\sharp^\star \beta) \Big] \mathrm{d}\hat{\boldsymbol{\sigma}}_K(\theta).$$

*In particular, this result is valid for* SOT.

*Proof.* The proof consists in applying Lemma A.3 for $(X, Y)$ chosen as $X = \mathrm{supp}(\hat{\boldsymbol{\sigma}}_K) \subset \mathbb{S}^{d-1}$ and

$$Y = \big\{ \forall \theta \in \mathrm{supp}(\hat{\boldsymbol{\sigma}}_K), \ f_\theta : \mathbb{R} \to \mathbb{R}, \ g_\theta : \mathbb{R} \to \mathbb{R}, \ f_\theta(x) + g_\theta(y) \leq \mathrm{C}_1(x, y) \big\}.$$

The functions in $Y$ are dual potentials, and by definition are continuous for any $\theta$. Let $\mathcal{F} : X \times Y \to \mathbb{R}$ be the functional defined as

$$\mathcal{F} : (\theta, (f_\theta)_\theta, (g_\theta)_\theta) \mapsto \int f_\theta \mathrm{d}(\theta_\sharp^\star \alpha) + \int g_\theta \mathrm{d}(\theta_\sharp^\star \beta).$$

Since the measures $(\alpha, \beta)$ have compact support, then by Lemma A.5, the supremum is attained over a subset of dual potentials of $Y$ such that for any fixed $\theta \in X$, $(f_\theta, g_\theta)$ are Lipschitz-continuous and bounded, thus uniformly equicontinuous functions (with constants independent of $\theta$). By the Ascoli-Arzela theorem, the set of uniformly equicontinuous functions is compact for the uniform convergence. Hence, for any $\theta \in X$, there exists a sequence of dual potentials $(f_{\theta,n}, g_{\theta,n})$ which uniformly converges to optimal dual potentials $(f_\theta, g_\theta)$ (up to extraction of subsequence). Besides, we have $\mathrm{OT}(\theta_\sharp^\star \alpha, \theta_\sharp^\star \beta) = \mathcal{F}(\theta, f_\theta, g_\theta)$ and $\mathcal{F}(\theta, (f_{\theta,n})_\theta, (g_{\theta,n})_\theta) \to \mathrm{OT}(\theta_\sharp^\star \alpha, \theta_\sharp^\star \beta)$ as $n \to +\infty$. Denote $\mathcal{F}_n(\theta) \triangleq \mathcal{F}(\theta, (f_{\theta,n})_\theta, (g_{\theta,n})_\theta)$ and $\mathrm{OT}(\theta) \triangleq \mathrm{OT}(\theta_\sharp^\star \alpha, \theta_\sharp^\star \beta)$. In order to apply Lemma A.3, we need to prove that the convergence of $(\mathcal{F}_n(\theta))_{n \in \mathbb{N}^*}$ to $\mathrm{OT}(\theta_\sharp^\star \alpha, \theta_\sharp^\star \beta)$ is uniform w.r.t. $\theta$, i.e., $\sup_{\theta \in X}|\mathcal{F}_n(\theta) - \mathrm{OT}(\theta)| \to 0$ as $n \to +\infty$.

First, note that for any $\theta \in X$,

$$|\mathcal{F}_n(\theta) - \mathrm{OT}(\theta)| \leq m(\alpha)\|f_{\theta,n} - f_\theta\|_\infty + m(\beta)\|g_{\theta,n} - g_\theta\|_\infty.$$

Since for a fixed $\theta \in X$, $(f_{\theta,n}, g_{\theta,n})_{n \in \mathbb{N}^*}$ uniformly converge to $(f_\theta, g_\theta)$, this means that

$$\forall \theta \in X, \ \forall \varepsilon > 0, \ \exists N(\varepsilon, \theta), \forall n \geq N(\varepsilon, \theta), \ m(\alpha)\|f_{\theta,n} - f_\theta\|_\infty + m(\beta)\|g_{\theta,n} - g_\theta\|_\infty < \varepsilon.$$

Since we assume that $\sigma$ is supported on a discrete set, then the cardinal of $X$ is finite and one can define $N(\varepsilon) \triangleq \max_{\theta \in X} N(\varepsilon, \theta)$. This yields,

$$\forall \varepsilon > 0, \ \exists N(\varepsilon), \forall n \geq N(\varepsilon), \sup_{\theta \in X}|\mathcal{F}_n(\theta) - \mathrm{OT}(\theta)| < \varepsilon.$$

which means that $\sup_{\theta \in X}|\mathcal{F}_n(\theta) - \mathrm{OT}(\theta)| \to 0$, thus concludes the proof.

$\square$

**Lemma A.5.** *Let $p \in [1, +\infty)$ and $\mathrm{C}_1(x, y) = |x - y|^p$. Consider two positive measures $(\alpha, \beta) \in \mathcal{M}_+(\mathbb{R}^d)$ whose support is such that $\mathrm{C}_d(x, y) = ||x - y||^p \leq R$. Then for any $\theta \in \mathbb{S}^{d-1}$, one can restrict without loss of generality the problem $\mathrm{UOT}(\theta_\sharp^\star \alpha, \theta_\sharp^\star \beta)$ as a supremum over dual potentials satisfying $f_\theta(x) + g_\theta(y) \leq \mathrm{C}_1(x, y)$, uniformly bounded by $M$ and uniformly $L$-Lipschitz, where $M$ and $L$ do not depend on $\theta$.*

*Proof.* We adapt the proof of (Santambrogio, 2015, Proposition 1.11), and focus on showing that the uniform boundedness and Lipschitz constant are independent of $\theta \in \mathbb{S}^{d-1}$ in this setting. Here we consider the translation-invariant formulation of UOT from (Séjourné et al., 2022b), *i.e.*, $\mathrm{UOT}(\alpha, \beta) = \sup_{f \oplus g \leq C_d} \mathcal{H}(f, g)$, where $\mathcal{H}(f, g) = \sup_{\lambda \in \mathbb{R}} \mathcal{D}(f + \lambda, g - \lambda)$. It is proved in (Séjourné et al., 2022b, Proposition 9) that the above problem has the same primal and is thus equivalent to optimize $\mathcal{D}$. By definition one has $\mathcal{H}(f, g) = \mathcal{H}(f + \lambda, g - \lambda)$ for any $\lambda \in \mathbb{R}$, *i.e.*, this formulation shares the same invariance as Balanced OT. Thus we can reuse all arguments from (Santambrogio, 2015, Proposition 1.11), such that for $\mathrm{UOT}(\alpha, \beta)$, one can use the constraint $f(x) + g(y) \leq C_d(x, y)$ and the assumption $C_d(x, y) \leq R$ to prove that without loss of generality, on can restrict to potentials such that $f(x) \in [0, R]$ and $g(y) \in [-R, R]$. Furthermore if the cost satisfies in $\mathbb{R}^d$

$$|C_d(x, y) - C_d(x', y')| \leq L(||x - x'|| + ||y - y'||),$$

then one can also restrict w.l.o.g. to potentials which are $L$-Lipschitz. For the cost $C_d(x, y) = ||x - y||^p$ with $p \geq 1$, this holds with constant $L = pR^{p-1}$ because the support is bounded and the gradient of $C_d$ is radially non-decreasing.

Regarding $\mathrm{OT}(\theta^\star_\sharp \alpha, \theta^\star_\sharp \beta)$, the bounds $(M_\theta, L_\theta)$ could be refined by considering the dependence in $\theta \in \mathbb{S}^{d-1}$. However we prove now these constants can be upper-bounded by a finite constant independent of $\theta$. In this setting we consider the cost

$$C_1(\theta^\star(x), \theta^\star(y)) = |\langle \theta, x - y \rangle|^p \leq ||\theta||^p ||x - y||^p \leq ||x - y||^p,$$

by Cauchy-Schwarz inequality. Therefore, if $(\alpha, \beta)$ have supports such that $||x - y||^p \leq R$, then $(\theta^\star_\sharp \alpha, \theta^\star_\sharp \beta)$ also have supports bounded by $R$ in $\mathbb{R}$. Similarly note that the derivative of $h(x) = x^p$ is non-decreasing for $p \geq 1$. Hence the cost $C_1(\theta^\star(x), \theta^\star(y))$ has a bounded derivative, which reads

$$p|\langle \theta, x - y \rangle|^{p-1} \leq p||\theta||^{p-1}||x - y||^{p-1} \leq p|x - y||^{p-1} \leq pR^{p-1}.$$

Thus on the supports of $(\theta^\star_\sharp \alpha, \theta^\star_\sharp \beta)$ one can also bound the Lipschitz constant of the cost $C_1(x, y) = |x - y|^p$ by the same constant $L$. $\qquad\square$

**Remark: Extending Theorem 3.9.** We conjecture that Theorem 3.9 also holds when $\boldsymbol{\sigma}$ is the uniform measures over $\mathbb{S}^{d-1}$, since the above holds for any $N \in \mathbb{N}^*$ and $\hat{\boldsymbol{\sigma}}_N$ converges weakly* to $\boldsymbol{\sigma}$. Proving this result would require that potentials $(f_\theta, g_\theta)$ are also regular (*i.e.*, Lipschitz and bounded) w.r.t $\theta \in \mathbb{S}^{d-1}$. This regularity is proved in (Xi & Niles-Weed, 2022) assuming $(\alpha, \beta)$ have densities, but remains unknown for discrete measures. Since discretizing $\boldsymbol{\sigma}$ corresponds to the computational approach, we assume it to be discrete, so that no additional assumption than boundedness on $(\alpha, \beta)$ is required. For instance, such result remains valid for semi-discrete UOT computation.

## A.3  Metric properties: Proof of Proposition 3.3 and Proposition 3.8

*Proof of Proposition 3.3.* **Metric properties of SUOT.** Symmetry and non-negativity are immediate. Assume $\mathrm{SUOT}(\alpha, \beta) = 0$. Since $\boldsymbol{\sigma}$ is the uniform distribution on $\mathbb{S}^{d-1}$, then for any $\theta \in \mathbb{S}^{d-1}$, $\mathrm{UOT}(\theta^\star_\sharp \alpha, \theta^\star_\sharp \beta) = 0$, and since UOT is assumed to be definite, then $\theta^\star_\sharp \alpha = \theta^\star_\sharp \beta$. By (Bogachev & Ruas, 2007, Proposition 3.8.6), this implies that $\alpha$ and $\beta$ have the same Fourier transform. By injectivity of the Fourier transform, we conclude that $\alpha = \beta$, hence SUOT is definite. The triangle inequality results from applying

the Minkowski inequality then the triangle inequality for $\mathrm{UOT}^{1/p}$ for $p \in [1, +\infty)$: for any $\alpha, \beta, \gamma \in \mathcal{M}_+(\mathbb{R}^d)$,

$$\mathrm{SUOT}^{1/p}(\alpha, \beta)$$

$$= \left( \int_{\mathbb{S}^{d-1}} \mathrm{UOT}(\theta_\sharp^\star \alpha, \theta_\sharp^\star \beta) \mathrm{d}\boldsymbol{\sigma}(\theta) \right)^{1/p}$$

$$\leq \left( \int_{\mathbb{S}^{d-1}} \left[ \mathrm{UOT}^{1/p}(\theta_\sharp^\star \alpha, \theta_\sharp^\star \gamma) + \mathrm{UOT}^{1/p}(\theta_\sharp^\star \gamma, \theta_\sharp^\star \beta) \right]^p \mathrm{d}\boldsymbol{\sigma}(\theta) \right)^{1/p}$$

$$\leq \left( \int_{\mathbb{S}^{d-1}} \left[ \mathrm{UOT}^{1/p}(\theta_\sharp^\star \alpha, \theta_\sharp^\star \gamma) \right]^p \mathrm{d}\boldsymbol{\sigma}(\theta) \right)^{1/p} + \left( \int_{\mathbb{S}^{d-1}} \left[ \mathrm{UOT}^{1/p}(\theta_\sharp^\star \gamma, \theta_\sharp^\star \beta) \right]^p \mathrm{d}\boldsymbol{\sigma}(\theta) \right)^{1/p}$$

$$= \mathrm{SUOT}^{1/p}(\alpha, \gamma) + \mathrm{SUOT}^{1/p}(\gamma, \beta).$$

$\square$

*Proof of Proposition 3.8.* **Metric properties of USOT.** Let $(\alpha, \beta) \in \mathcal{M}_+(\mathbb{R}^d)$. Non-negativity is immediate, as USOT is defined as a program minimizing a sum of positive terms. SOT is symmetric, thus when $\varphi_1 = \varphi_2$, we obtain symmetry of the functional w.r.t. $(\alpha, \beta)$. Assume $\mathrm{D}_\varphi$ is definite, *i.e.*, $\mathrm{D}_\varphi(\alpha|\beta) = 0$ implies $\alpha = \beta$. Assume now that $\mathrm{USOT}(\alpha, \beta) = 0$, and denote by $(\pi_1, \pi_2)$ the optimal marginals attaining the infimum in (9). $\mathrm{USOT}(\alpha, \beta) = 0$ implies that $\mathrm{SOT}(\pi_1, \pi_2) = 0$, $\mathrm{D}_\varphi(\pi_1|\alpha) = 0$ and $\mathrm{D}_\varphi(\pi_2|\beta) = 0$. These three terms are definite, which yields $\alpha = \pi_1 = \pi_2 = \beta$, hence the definiteness of USOT. The Partial OT setting (*i.e.*, $\mathrm{D}_\varphi = \rho\mathrm{TV}$) is treated in Appendix A.7.

$\square$

## A.4 Sample complexity: Proof of Theorem 3.4

Theorem 3.4 is obtained by adapting (Nadjahi et al., 2020b, Theorems 4 and 5). We provide the detailed derivations below.

*Proof of Theorem 3.4.* Let $(\alpha, \beta) \in \widetilde{\mathrm{M}} \times \widetilde{\mathrm{M}}$ with respective empirical approximations $\hat{\alpha}_n, \hat{\beta}_n$ over $n$ samples. By using the definition of SUOT, the triangle inequality and the assumed sample complexity of UOT for univariate measures, we show that

$$\mathbb{E} \left| \mathrm{SUOT}(\alpha, \beta) - \mathrm{SUOT}(\hat{\alpha}_n, \hat{\beta}_n) \right| \tag{15}$$

$$= \mathbb{E} \left| \int_{\mathbb{S}^{d-1}} \left\{ \mathrm{UOT}(\theta_\sharp^\star \alpha, \theta_\sharp^\star \beta) - \mathrm{UOT}(\theta_\sharp^\star \hat{\alpha}_n, \theta_\sharp^\star \hat{\beta}_n) \right\} \mathrm{d}\boldsymbol{\sigma}(\theta) \right| \tag{16}$$

$$\leq \mathbb{E} \left\{ \int_{\mathbb{S}^{d-1}} \left| \mathrm{UOT}(\theta_\sharp^\star \alpha, \theta_\sharp^\star \beta) - \mathrm{UOT}(\theta_\sharp^\star \hat{\alpha}_n, \theta_\sharp^\star \hat{\beta}_n) \right| \mathrm{d}\boldsymbol{\sigma}(\theta) \right\} \tag{17}$$

$$\leq \int_{\mathbb{S}^{d-1}} \mathbb{E} \left| \mathrm{UOT}(\theta_\sharp^\star \alpha, \theta_\sharp^\star \beta) - \mathrm{UOT}(\theta_\sharp^\star \hat{\alpha}_n, \theta_\sharp^\star \hat{\beta}_n) \right| \mathrm{d}\boldsymbol{\sigma}(\theta) \tag{18}$$

$$\leq \int_{\mathbb{S}^{d-1}} \kappa(n) \mathrm{d}\boldsymbol{\sigma}(\theta) = \kappa(n), \tag{19}$$

which completes the proof for the first setting.

Next, let $\alpha \in \widetilde{\mathrm{M}}$ with corresponding empirical approximation $\hat{\alpha}_n$. Then, using the definition of SUOT, the triangle inequality (w.r.t. integral) and the assumed convergence rate in UOT,

$$\mathbb{E} \left| \mathrm{SUOT}(\hat{\alpha}_n, \alpha) \right| \tag{20}$$

$$= \mathbb{E} \left| \int_{\mathbb{S}^{d-1}} \mathrm{UOT}(\theta_\sharp^\star \hat{\alpha}_n, \theta_\sharp^\star \alpha) \mathrm{d}\boldsymbol{\sigma}(\theta) \right| \leq \mathbb{E} \left\{ \int_{\mathbb{S}^{d-1}} \left| \mathrm{UOT}(\theta_\sharp^\star \hat{\alpha}_n, \theta_\sharp^\star \alpha) \right| \mathrm{d}\boldsymbol{\sigma}(\theta) \right\} \tag{21}$$

$$\leq \int_{\mathbb{S}^{d-1}} \mathbb{E} \left| \mathrm{UOT}(\theta_\sharp^\star \hat{\alpha}_n, \theta_\sharp^\star \alpha) \right| \mathrm{d}\boldsymbol{\sigma}(\theta) \leq \int_{\mathbb{S}^{d-1}} \xi(n) \mathrm{d}\boldsymbol{\sigma}(\theta) = \xi(n). \tag{22}$$

$\square$

**Corollary A.6.** *Assume for $\mu \in \mathcal{M}_+(\mathbb{R})$, $\mathbb{E}|\mathrm{UOT}(\mu, \hat{\mu}_n)| \leq \xi(n)$ and that for $p \geq 1$, $\mathrm{UOT}^{1/p}$ satisfies non-negativity, symmetry and the triangle inequality on $\mathcal{M}_+(\mathbb{R}) \times \mathcal{M}_+(\mathbb{R})$. Then, for $\alpha, \beta \in \mathcal{M}_+(\mathbb{R}^d)$,*

$$\mathbb{E}\left|\mathrm{SUOT}^{1/p}(\alpha, \beta) - \mathrm{SUOT}^{1/p}(\hat{\alpha}_n, \hat{\beta}_n)\right| \leq 2\xi(n)^{1/p}. \tag{23}$$

*Proof.* Since $\mathrm{UOT}^{1/p}$ satisfies non-negativity, symmetry and the triangle inequality on $\mathcal{M}_+(\mathbb{R}) \times \mathcal{M}_+(\mathbb{R})$, $\mathrm{SUOT}^{1/p}$ verifies these three metric properties on $\mathcal{M}_+(\mathbb{R}^d) \times \mathcal{M}_+(\mathbb{R}^d)$ by Proposition 3.3, and we can derive its sample complexity as follows. For any $\alpha, \beta$ in $\mathcal{M}_+(\mathbb{R}^d)$ with respective empirical approximations $\hat{\alpha}_n, \hat{\beta}_n$, applying the triangle inequality yields for $p \in [1, +\infty)$,

$$\left|\mathrm{UOT}^{1/p}(\alpha, \beta) - \mathrm{UOT}^{1/p}(\hat{\alpha}_n, \hat{\beta}_n)\right| \leq \mathrm{UOT}^{1/p}(\hat{\alpha}_n, \alpha) + \mathrm{UOT}^{1/p}(\hat{\beta}_n, \beta). \tag{24}$$

Taking the expectation of (24) with respect to $\hat{\alpha}_n, \hat{\beta}_n$ gives,

$$\mathbb{E}\left|\mathrm{SUOT}^{1/p}(\alpha, \beta) - \mathrm{SUOT}^{1/p}(\hat{\alpha}_n, \hat{\beta}_n)\right| \leq \mathbb{E}|\mathrm{SUOT}^{1/p}(\hat{\alpha}_n, \alpha)| + \mathbb{E}|\mathrm{SUOT}^{1/p}(\hat{\beta}_n, \beta)| \tag{25}$$

$$\leq \{\mathbb{E}|\mathrm{SUOT}(\hat{\alpha}_n, \alpha)|\}^{1/p} + \{\mathbb{E}|\mathrm{SUOT}(\hat{\beta}_n, \beta)|\}^{1/p} \tag{26}$$

$$\leq \xi(n)^{1/p} + \xi(n)^{1/p} = 2\xi(n)^{1/p}, \tag{27}$$

where (26) is immediate if $p = 1$, and results from applying Hölder's inequality on $\mathbb{S}^{d-1}$ if $p > 1$, and (27) follows from (22). $\square$

## A.5 Comparison of SUOT, USOT, SOT, and proof of Theorem 3.10 and Theorem 3.11

In this section, we establish several bounds to compare SUOT, USOT and SOT on the space of compactly-supported measures. We provide the detailed derivations and auxiliary lemmas needed for the proofs. The Partial OT setting (*i.e.*, $\mathrm{D}_\varphi = \rho\mathrm{TV}$) is treated in Appendix A.7.

**Proof of Theorem 3.10.** Theorem 3.10 is a direct consequence from Theorems A.7 and A.8.

**Theorem A.7.** *Let $(\alpha, \beta) \in \mathcal{M}_+(\mathbb{R}^d) \times \mathcal{M}_+(\mathbb{R}^d)$. Then, $\mathrm{SUOT}(\alpha, \beta) \leq \mathrm{USOT}(\alpha, \beta)$.*

*Proof.* To show that $\mathrm{SUOT}(\alpha, \beta) \leq \mathrm{USOT}(\alpha, \beta)$, we use a sub-optimality argument. Let $\pi$ be the solution $\mathrm{USOT}(\alpha, \beta)$ and denote by $(\pi_1, \pi_2)$ the marginals of $\pi$. For any $\theta \in \mathbb{S}^{d-1}$, denote by $\pi_\theta$ the solution of $\mathrm{OT}(\theta^\star_\sharp \pi_1, \theta^\star_\sharp \pi_2)$. By definition of USOT, the marginals of $\pi_\theta$ are given by $(\theta^\star_\sharp \pi_1, \theta^\star_\sharp \pi_2)$. Since the sequence $(\pi_\theta)_\theta$ is suboptimal for the problem $\mathrm{SUOT}(\alpha, \beta)$, one has

$$\mathrm{SUOT}(\alpha, \beta) \leq \int_{\mathbb{S}^{d-1}} \left\{ \int \mathrm{C}_1 \mathrm{d}\pi_\theta + \mathrm{D}_{\varphi_1}(\theta^\star_\sharp \pi_1 | \theta^\star_\sharp \alpha) + \mathrm{D}_{\varphi_2}(\theta^\star_\sharp \pi_2 | \theta^\star_\sharp \beta) \right\} \mathrm{d}\boldsymbol{\sigma}(\theta) \tag{28}$$

$$\leq \int_{\mathbb{S}^{d-1}} \int \mathrm{C}_1 \mathrm{d}\pi_\theta \mathrm{d}\boldsymbol{\sigma}(\theta) + \mathrm{D}_{\varphi_1}(\pi_1 | \alpha) + \mathrm{D}_{\varphi_2}(\pi_2 | \beta) \tag{29}$$

$$= \mathrm{USOT}(\alpha, \beta), \tag{30}$$

where the second inequality results from Lemma A.10, and the last equality follows from the definition of $\mathrm{USOT}(\alpha, \beta)$. $\square$

**Theorem A.8.** *Let $(\alpha, \beta) \in \mathcal{M}_+(\mathbb{R}^d) \times \mathcal{M}_+(\mathbb{R}^d)$. Additionally, let $p \in [1, +\infty)$ and assume that for all $x, y \in \mathbb{R}^d$, $\theta \in \mathbb{S}^{d-1}$, $\mathrm{C}_1\big(\theta^\star(x), \theta^\star(y)\big) \leq \mathrm{C}_d(x, y)$. Then, $\mathrm{USOT}(\alpha, \beta) \leq \mathrm{UOT}(\alpha, \beta)$.*

*Proof.* By (Bonnotte, 2013, Proposition 5.1.3), $\mathrm{SOT}(\mu, \nu) \leq \mathrm{OT}(\mu, \nu)$ as $\mathrm{C}_1\big(\theta^\star(x), \theta^\star(y)\big) \leq \mathrm{C}_d(x, y)$ for all $x, y \in \mathbb{R}^d$, $\theta \in \mathbb{S}^{d-1}$. Let $\pi$ be the solution of $\mathrm{UOT}(\alpha, \beta)$ with marginals $(\pi_1, \pi_2)$. These marginals are

sub-optimal for $\text{USOT}(\alpha, \beta)$, we have

$$\text{USOT}(\alpha, \beta) \leq \text{SOT}(\pi_1, \pi_2) + \text{D}_{\varphi_1}(\pi_1|\alpha) + \text{D}_{\varphi_2}(\pi_2|\beta), \tag{31}$$

$$\leq \text{OT}(\pi_1, \pi_2) + \text{D}_{\varphi_1}(\pi_1|\alpha) + \text{D}_{\varphi_2}(\pi_2|\beta), \tag{32}$$

$$= \text{UOT}(\alpha, \beta), \tag{33}$$

where the last equality is obtained because $\pi$ is optimal in $\text{UOT}(\alpha, \beta)$.

$\square$

**Proof of Theorem 3.11.**

**Theorem A.9.** *Let* $\mathsf{X}$ *be a compact subset of* $\mathbb{R}^d$ *with radius* $R$ *and consider* $\alpha, \beta \in \mathcal{M}_+(\mathsf{X})$. *Additionally, let* $p \in [1, +\infty)$ *and assume* $\text{C}_1(x, y) = |x - y|^p$ *for* $(x, y) \in \mathbb{R}$ *and* $\text{C}_d(x, y) = \|x - y\|^p$ *for* $(x, y) \in \mathbb{R}^d$. *Let* $\rho > 0$ *and assume* $D_{\varphi_1} = D_{\varphi_2} = \rho\text{KL}$. *Then,* $\text{UOT}(\alpha, \beta) \leq c\,\text{SUOT}(\alpha, \beta)^{1/(d+1)}$, *where* $c = c(m(\alpha), m(\beta), \rho, R)$ *is a non-decreasing function of* $m(\alpha)$ *and* $m(\beta)$.

*Proof.* We adapt the proof of (Bonnotte, 2013, Lemma 5.1.4), which establishes a bound between OT and SOT. The first step consists in bounding from above the distance between two regularized measures.

Let $\psi : \mathbb{R}^d \to \mathbb{R}_+$ be a smooth and radial function such that $\text{supp}(\psi) \subseteq B_d(\mathbf{0}, 1)$ and $\int_{\mathbb{R}^d} \psi(x)\text{dLeb}(x) = 1$. Let $\psi_\lambda(x) = \lambda^{-d}\psi(x/\lambda)$. For any function $f$ defined on $\mathbb{R}^s$ ($s \geq 1$), denote by $\mathcal{F}[f]$ the Fourier transform of $f$ defined for $x \in \mathbb{R}^s$ as $\mathcal{F}[f](x) = \int_{\mathbb{R}^s} f(w)e^{-i\langle w, x\rangle}\text{d}w$. Let $\alpha_\lambda = \alpha * \varphi_\lambda$ and $\beta_\lambda = \beta * \varphi_\lambda$ where $*$ is the convolution operator. Let $(f, g)$ such that $f \oplus g \leq \text{C}_d$. By using the isometry properties of the Fourier transform and the definition of $\psi_\lambda$, then representing the variables with polar coordinates, we have

$$\int_{\mathbb{R}^d} \varphi^\circ(f(x))\text{d}\alpha_\lambda(x) = \int_{\mathbb{R}^d} \mathcal{F}[\varphi^\circ \circ f](w)\mathcal{F}[\alpha](w)\mathcal{F}[\psi](\lambda w)\text{d}w \tag{34}$$

$$= \int_{\mathbb{S}^{d-1}} \int_0^{+\infty} \mathcal{F}[\varphi^\circ \circ f](r\theta)\mathcal{F}[\alpha](r\theta)\mathcal{F}[\psi](\lambda r)r^{d-1}\text{d}r\text{d}\theta. \tag{35}$$

Since $\varphi^\circ \circ f$ is a real-valued function, $\mathcal{F}[\varphi^\circ \circ f]$ is an even function, then

$$\int_{\mathbb{R}^d} \varphi^\circ(f(x))\text{d}\alpha_\lambda(x) \tag{36}$$

$$= \frac{1}{2}\int_{\mathbb{S}^{d-1}} \int_{\mathbb{R}} \mathcal{F}[\varphi^\circ \circ f](r\theta)\mathcal{F}[\alpha](r\theta)\mathcal{F}[\psi](\lambda r)|r|^{d-1}\text{d}r\text{d}\theta \tag{37}$$

$$= \frac{1}{2}\int_{\mathbb{S}^{d-1}} \int_{\mathbb{R}} \mathcal{F}[\varphi^\circ \circ f](r\theta)\mathcal{F}[\theta_\sharp^\star \alpha](r)\mathcal{F}[\psi](\lambda r)|r|^{d-1}\text{d}r\text{d}\theta \tag{38}$$

$$= \frac{1}{2}\int_{\mathbb{S}^{d-1}} \int_{\mathbb{R}} \mathcal{F}[\varphi^\circ \circ f](r\theta)\left(\int_{-R}^R e^{-iru}\text{d}\theta_\sharp^\star \alpha(u)\right)\mathcal{F}[\psi](\lambda r)\,|r|^{d-1}\,\text{d}r\text{d}\theta \tag{39}$$

$$= \frac{1}{2}\int_{\mathbb{S}^{d-1}} \int_{\mathbb{R}} \left(\int_{\mathbb{R}^d} \int_{-R}^R \varphi^\circ(f(x))e^{-ir(u+\langle\theta, x\rangle)}\text{d}\theta_\sharp^\star \alpha(u)\right)\mathcal{F}[\psi](\lambda r)\,|r|^{d-1}\,\text{d}x\text{d}r\text{d}\theta. \tag{40}$$

Equation 38 follows from the property of push-forward measures, (39) results from the definition of the Fourier transform and $u \in [-R, R]$, and (40) results from the definition of the Fourier transform and Fubini's theorem. By making a change of variables ($x$ becomes $x - u\theta$), we obtain

$$\int_{\mathbb{R}^d} \varphi^\circ(f(x))\text{d}\alpha_\lambda(x) \tag{41}$$

$$= \frac{1}{2}\int_{\mathbb{S}^{d-1}} \int_{\mathbb{R}} \int_{\mathbb{R}^d} \int_{-R}^R \varphi^\circ(f(x - u\theta))e^{-ir\langle\theta, x\rangle}\text{d}\theta_\sharp^\star \alpha(u)\mathcal{F}[\psi](\lambda r)\,|r|^{d-1}\,\text{d}x\text{d}r\text{d}\theta \tag{42}$$

$$= \frac{1}{2}\int_{\mathbb{S}^{d-1}} \int_{\mathbb{R}} \int_{B_d(\mathbf{0}, 2R+\lambda)} \int_{-R}^R \varphi^\circ(f(x - u\theta))e^{-ir\langle\theta, x\rangle}\text{d}\theta_\sharp^\star \alpha(u)\mathcal{F}[\psi](\lambda r)\,|r|^{d-1}\,\text{d}x\text{d}r\text{d}\theta, \tag{43}$$

where (43) follows from $\text{supp}(\alpha) \subseteq B_d(\mathbf{0}, R)$. Indeed, this implies that $\text{supp}(\alpha_\lambda) \subseteq B_d(\mathbf{0}, R + \lambda)$, thus the domain of $x \mapsto \varphi^\circ \circ f(x - u\theta)$ is contained in $B_d(\mathbf{0}, 2R + \lambda)$.

Similarly, one can show that

$$\int_{\mathbb{R}^d} \varphi^\circ(g(y)) \mathrm{d}\beta_\lambda(y) \tag{44}$$

$$= \frac{1}{2} \int_{\mathbb{S}^{d-1}} \int_{\mathbb{R}} \int_{B_d(\mathbf{0}, 2R+\lambda)} \int_{-R}^{R} \varphi^\circ(g(y - u\theta)) e^{-\mathrm{i}r\langle \theta, y \rangle} \mathrm{d}\theta_\sharp^\star \beta(u) \mathcal{F}[\psi](\lambda r) |r|^{d-1} \mathrm{d}y \mathrm{d}r \mathrm{d}\theta . \tag{45}$$

By (43) and (45), we obtain

$$\int_{\mathbb{R}^d} \varphi^\circ(f(x)) \mathrm{d}\alpha_\lambda(x) + \int_{\mathbb{R}^d} \varphi^\circ(g(y)) \mathrm{d}\beta_\lambda(y) \tag{46}$$

$$= \frac{1}{2} \int_{\mathbb{S}^{d-1}} \int_{\mathbb{R}} \int_{B_d(\mathbf{0}, 2R+\lambda)} \left\{ \int_{-R}^{R} \varphi^\circ(f(x - u\theta)) \mathrm{d}\theta_\sharp^\star \alpha(u) \right.$$

$$\left. + \int_{-R}^{R} \varphi^\circ(g(x - u\theta)) \mathrm{d}\theta_\sharp^\star \beta(u) \right\} e^{-\mathrm{i}r\langle \theta, x \rangle} \mathcal{F}[\psi](\lambda r) |r|^{d-1} \mathrm{d}x \mathrm{d}r \mathrm{d}\theta , \tag{47}$$

and,

$$\left| \int_{B_d(\mathbf{0}, 2R+\lambda)} \left\{ \int_{-R}^{R} \varphi^\circ(f(x - u\theta)) \mathrm{d}\theta_\sharp^\star \alpha(u) + \int_{-R}^{R} \varphi^\circ(g(x - u\theta)) \mathrm{d}\theta_\sharp^\star \beta(y) \right\} e^{-ir\langle \theta, x \rangle} \mathrm{d}x \right| \tag{48}$$

$$\leq \int_{B_d(\mathbf{0}, 2R+\lambda)} \mathrm{UOT}(\theta_\sharp^\star \alpha, \theta_\sharp^\star \beta) |e^{-ir\langle \theta, x \rangle}| \mathrm{d}x \tag{49}$$

$$\leq (2R + \lambda)^d \mathrm{UOT}(\theta_\sharp^\star \alpha, \theta_\sharp^\star \beta) , \tag{50}$$

where (49) is obtained by taking the supremum of (48) over the set of potentials $(\tilde{f}, \tilde{g})$ such that for $u \in [-R, R]$, $\exists (x, \theta) \in B_d(\mathbf{0}, 2R + \lambda) \times \mathbb{S}^{d-1}$, $\tilde{f}(u) = f(x - u\theta)$, $\tilde{g}(u) = g(x - u\theta)$, which is included in the set of potentials $(f', g')$ s.t. $f' : \mathbb{R} \to \mathbb{R}$, $g' : \mathbb{R} \to \mathbb{R}$ and $f' \oplus g' \leq \mathrm{C}_1$. Therefore,

$$\int_{\mathbb{R}^d} \varphi^\circ(f(x)) \mathrm{d}\alpha_\lambda(x) + \int_{\mathbb{R}^d} \varphi^\circ(g(y)) \mathrm{d}\beta_\lambda(y)$$

$$\leq \frac{1}{2} (2R + \lambda)^d \mathcal{A}(\mathbb{S}^{d-1}) \int_{\mathbb{S}^{d-1}} \mathrm{UOT}(\theta_\sharp^\star \alpha, \theta_\sharp^\star \beta) \mathrm{d}\boldsymbol{\sigma}(\theta) \int_{\mathbb{R}} \lambda^{-d} |\mathcal{F}[\psi](r) |r|^{d-1} | \mathrm{d}r \tag{51}$$

$$\leq c(2R + \lambda)^d \lambda^{-d} \mathrm{SUOT}(\alpha, \beta) , \tag{52}$$

where $c = \frac{1}{2} \mathcal{A}(\mathbb{S}^{d-1}) \int_{\mathbb{R}} |\mathcal{F}[\psi](r) |r|^{d-1} | \mathrm{d}r$. We deduce from the dual formulation of UOT (3) and (49) that,

$$\mathrm{UOT}(\alpha_\lambda, \beta_\lambda) \leq c(2R + \lambda)^d \lambda^{-d} \mathrm{SUOT}(\alpha, \beta) . \tag{53}$$

The last step of the proof consists in relating $\mathrm{UOT}(\alpha_\lambda, \beta_\lambda)$ with $\mathrm{UOT}(\alpha, \beta)$. For any $(f, g)$ such that $f \oplus g \leq \mathrm{C}_d$, we have

$$\int_{\mathbb{R}^d} \varphi^\circ(f(x)) \mathrm{d}\alpha(x) + \int_{\mathbb{R}^d} \varphi^\circ(g(y)) \mathrm{d}\beta(y) - \mathrm{UOT}(\alpha_\lambda, \beta_\lambda) \tag{54}$$

$$\leq \int_{\mathbb{R}^d} \varphi^\circ(f(x)) \mathrm{d}\alpha(x) + \int_{\mathbb{R}^d} \varphi^\circ(g(x)) \mathrm{d}\beta(x) - \int_{\mathbb{R}^d} \varphi^\circ(f(x)) \mathrm{d}\alpha_\lambda(x) - \int_{\mathbb{R}^d} \varphi^\circ(g(y)) \mathrm{d}\beta_\lambda(y) \tag{55}$$

$$\leq \int_{\mathbb{R}^d} \{\varphi^\circ(f(x)) - \psi_\lambda * \varphi^\circ(f(x))\} \mathrm{d}\alpha(x) + \int_{\mathbb{R}^d} \{\varphi^\circ(g(y)) - \psi_\lambda * \varphi^\circ(g(y))\} \mathrm{d}\beta(y) . \tag{56}$$

For $x \in \mathbb{R}^d$,

$$\varphi^\circ(f(x)) - \psi_\lambda * \varphi^\circ(f(x)) = \lambda^{-d} \int_{\mathbb{R}^d} \left( \varphi^\circ(f(x)) - \varphi^\circ(f(y)) \right) \psi\left( \frac{x-y}{\lambda} \right) \mathrm{d}y \tag{57}$$

$$\leq \lambda^{-d} \int_{\mathbb{R}^d} \left| \varphi^\circ(f(x)) - \varphi^\circ(f(y)) \right| \psi\left( \frac{x-y}{\lambda} \right) \mathrm{d}y \,, \tag{58}$$

Since $\mathrm{D}_\varphi = \rho\mathrm{KL}$, then for $z \in \mathbb{R}$, $\varphi^\circ(z) = \rho(1 - e^{-z/\rho})$, so for $(x,y) \in \mathbb{R}^d \times \mathbb{R}^d$,

$$\varphi^\circ(f(x)) - \varphi^\circ(f(y)) = \rho(e^{-f(y)/\rho} - e^{-f(x)/\rho}) \tag{59}$$

By Lemma A.13, the potentials $(f,g)$ are bounded by constants depending on $m(\alpha), m(\beta)$. Therefore, we can bound (59) as follows.

$$|\varphi^\circ(f(x)) - \varphi^\circ(f(y))| \leq e^{-\lambda^\star/\rho} \|x - y\| \,. \tag{60}$$

We thus derive the following upper-bound on (58).

$$\varphi^\circ(f(x)) - \psi_\lambda * \varphi^\circ(f(x)) \leq \lambda^{-d} e^{-\lambda^\star/\rho} \int_{\mathbb{R}^d} \|x - y\| \psi\left( \frac{x-y}{\lambda} \right) \mathrm{d}y \tag{61}$$

$$\leq \lambda^{-d+1} e^{-\lambda^\star/\rho} \int_{\mathbb{R}^d} \frac{\|x-y\|}{\lambda} \psi\left( \frac{x-y}{\lambda} \right) \mathrm{d}y \tag{62}$$

By doing the change of variables $z = (y-x)/\lambda$ and using the fact that $\psi$ is a radial function, we obtain

$$\varphi^\circ(f(x)) - \psi_\lambda * \varphi^\circ(f(x)) \leq \lambda e^{-\lambda^\star/\rho} \int_{\mathbb{R}^d} \|z\| \psi(z) \mathrm{d}z \,. \tag{63}$$

Similarly, using the bounds on $g$ in Lemma A.13, one can show that

$$|\varphi^\circ(g(x)) - \varphi^\circ(g(y))| \leq e^{(\lambda^\star + R)/\rho} \|x - y\| \,, \tag{64}$$

therefore,

$$\varphi^\circ(g(x)) - \psi_\lambda * \varphi^\circ(g(x)) \leq \lambda e^{(\lambda^\star + R)/\rho} \int_{\mathbb{R}^d} \|z\| \psi(z) \mathrm{d}z \,. \tag{65}$$

We conclude that,

$$\int_{\mathbb{R}^d} \varphi^\circ(f(x)) \mathrm{d}\alpha(x) + \int_{\mathbb{R}^d} \varphi^\circ(g(y)) \mathrm{d}\beta(y) - \mathrm{UOT}(\alpha_\lambda, \beta_\lambda) \leq \left( e^{-\lambda^\star/\rho} + e^{(\lambda^\star + R)/\rho} \right) \lambda \mathrm{M}_1(\psi) \,, \tag{66}$$

where $\mathrm{M}_1(\psi) \triangleq \int_{\mathbb{R}^d} \|z\| \psi(z) \mathrm{d}z$. Taking the supremum on both sides over $(f,g)$ such that $f \oplus g \leq \mathrm{C}_d$ yields,

$$\mathrm{UOT}(\alpha, \beta) - \mathrm{UOT}(\alpha_\lambda, \beta_\lambda) \leq \left( e^{-\lambda^\star/\rho} + e^{(\lambda^\star + R)/\rho} \right) \lambda \mathrm{M}_1(\psi) \,. \tag{67}$$

Finally, by combining (53) with the above inequality, we obtain

$$\mathrm{UOT}(\alpha, \beta) \leq \left( e^{-\lambda^\star/\rho} + e^{(\lambda^\star + R)/\rho} \right) \lambda \mathrm{M}_1(\psi) + c(2R + \lambda)^d \lambda^{-d} \mathrm{SUOT}(\alpha, \beta) \tag{68}$$

$$\leq c'\lambda\left( 1 + (2R + \lambda)^d \lambda^{-(d+1)} \mathrm{SUOT}(\alpha, \beta) \right), \tag{69}$$

where $c'$ is a constant satisfying $c' \geq c$ and $c' \geq \left( e^{-\lambda^\star/\rho} + e^{(\lambda^\star + R)/\rho} \right) \mathrm{M}_1(\psi)$. By choosing $\lambda = R^{d/(d+1)} \mathrm{SUOT}(\alpha, \beta)^{1/(d+1)}$, (69) becomes

$$\mathrm{UOT}(\alpha, \beta) \leq c' R^{d/(d+1)} \mathrm{SUOT}(\alpha, \beta)^{1/(d+1)} \left( 1 + (2R + \lambda)^d R^{-d} \right). \tag{70}$$

We conclude using that $\mathrm{SUOT}(\alpha, \beta)$ is bounded from above. Indeed, $\mathrm{SUOT}(\alpha, \beta) \leq \rho(m(\alpha) + m(\beta))$ since on the one hand, $\pi$ is suboptimal in (3) thus $\mathrm{UOT}(\alpha, \beta) \leq \rho(m(\alpha) + m(\beta))$, and on the other hand, $m(\alpha) = m(\theta^\star_\sharp \alpha)$ for any $\theta \in \mathbb{S}^{d-1}$. This yields $\lambda \leq R^{d/(d+1)} \rho^{1/(d+1)} (m(\alpha) + m(\beta))^{1/(d+1)}$, hence

$$\mathrm{UOT}(\alpha, \beta) \leq c(m(\alpha), m(\beta), \rho, R) \mathrm{SUOT}(\alpha, \beta)^{1/(d+1)} \,, \tag{71}$$

where $c(m(\alpha), m(\beta), \rho, R)$ is a non-decreasing function of $m(\alpha)$ and $m(\beta)$.

$\square$

**Additional Lemmas.**

**Lemma A.10.** *For any $\theta \in \mathbb{S}^{d-1}$ and $\alpha, \beta \in \mathcal{M}_+(\mathbb{R}^d)$, $D_\varphi(\theta^\star_\sharp \alpha | \theta^\star_\sharp \beta) \leq D_\varphi(\alpha | \beta)$.*

*Proof.* For $\alpha, \beta \in \mathcal{M}_+(\mathbb{R}^s)$ with $s \geq 1$, the dual characterization of $\varphi$-divergences reads ([Liero et al., 2018](), Theorem 2.7)

$$D_\varphi(\alpha | \beta) = \sup_{f \in \mathcal{E}(\mathbb{R}^s)} \int_{\mathbb{R}^s} \varphi^\circ(f(x)) \mathrm{d}\beta(x) - \int_{\mathbb{R}^s} f(x) \mathrm{d}\alpha(x),$$

where $\mathcal{E}(\mathbb{R}^s)$ denotes the space of lower semi-continuous functions from $\mathbb{R}^s$ to $\mathbb{R} \cup \{+\infty\}$. Therefore, for any $\theta \in \mathbb{S}^{d-1}$ and $\alpha, \beta \in \mathcal{M}_+(\mathbb{R}^d)$,

$$D_\varphi(\theta^\star_\sharp \alpha | \theta^\star_\sharp \beta) = \sup_{f \in \mathcal{E}(\mathbb{R})} \int_{\mathbb{R}} \varphi^\circ(f(t)) \mathrm{d}(\theta^\star_\sharp \beta)(t) - \int_{\mathbb{R}} f(t) \mathrm{d}(\theta^\star_\sharp \alpha)(t) \tag{72}$$

$$= \sup_{g: \mathbb{R}^d \to \mathbb{R} \, s.t. \, \exists f \in \mathcal{E}(\mathbb{R}), \, g = f \circ \theta^\star} \int_{\mathbb{R}^d} \varphi^\circ(g(x)) \mathrm{d}\beta(x) - \int_{\mathbb{R}^d} g(x) \mathrm{d}\alpha(x) \tag{73}$$

where (73) results from the definition of push-forward measures. We conclude the proof by observing that the supremum in (73) is taken over a subset of $\mathcal{E}(\mathbb{R}^d)$.

$\square$

**Lemma A.11.** *([Santambrogio, 2015](), Proposition 1.11) Let $p \in [1, +\infty)$ and assume $C_d(x, y) = \|x - y\|^p$. Let $\alpha, \beta$ with compact support, such that $C_d(x, y) \leq R^p$ for $(x, y) \in \mathrm{supp}(\alpha) \times \mathrm{supp}(\beta)$. Then without loss of generality the dual potentials $(f, g)$ of $\mathrm{UOT}(\alpha, \beta)$ satisfy $f(x) \in [0, R]$ and $g(y) \in [-R, R]$.*

**Lemma A.12.** *([Séjourné et al., 2022b](), Proposition 2) Define the translation-invariant dual formulation*

$$\mathrm{UOT}(\alpha, \beta) = \sup_{f \oplus g \leq C_d} \sup_{\lambda \in \mathbb{R}} \int \varphi^\circ_1(f + \lambda) \mathrm{d}\alpha + \int \varphi^\circ_2(g - \lambda) \mathrm{d}\beta. \tag{74}$$

*Let $\rho > 0$ and assume $D_{\varphi_1} = D_{\varphi_2} = \rho\mathrm{KL}$. Take optimal potentials $(f, g)$ in (74). Then optimal potentials in (3) are given by $(f + \lambda^\star(f, g), g - \lambda^\star(f, g))$, where the optimal translation $\lambda^\star$ reads*

$$\lambda^\star(f, g) \triangleq \frac{1}{2}\left[S^\beta_\rho(g) - S^\alpha_\rho(f)\right], \quad S^\alpha_\rho(f) \triangleq -\rho \log \int e^{-f/\rho} \mathrm{d}\alpha,$$

*and we call $S^\alpha_\rho(f)$ the soft-minimum of $f$. When $m(\alpha) = 1$ and $m \leq f(x) \leq M$, then $m \leq S^\alpha_\rho(f) \leq M$.*

**Lemma A.13.** *Assume $(\alpha, \beta)$ have compact support such that, for $(x, y) \in \mathrm{supp}(\alpha) \times \mathrm{supp}(\beta)$, $C(x, y) \leq R$. Then, without loss of generality, one can restrict the optimization of the dual formulation (3) of $\mathrm{UOT}(\alpha, \beta)$ over the set of potentials satisfying for $(x, y) \in \mathrm{supp}(\alpha) \times \mathrm{supp}(\beta)$,*

$$f(x) \in [\lambda^\star, \lambda^\star + R], \quad g(y) \in [-\lambda^\star - R, -\lambda^\star + R],$$

*where $\lambda^\star \in [-R + \frac{\rho}{2} \log \frac{m(\alpha)}{m(\beta)}, \frac{R}{2} + \frac{\rho}{2} \log \frac{m(\alpha)}{m(\beta)}]$. In particular, one has*

$$f(x) \in [-R + \frac{\rho}{2} \log \frac{m(\alpha)}{m(\beta)}, \frac{3R}{2} + \frac{\rho}{2} \log \frac{m(\alpha)}{m(\beta)}], \quad g(y) \in [-\frac{3R}{2} - \frac{\rho}{2} \log \frac{m(\alpha)}{m(\beta)}, 2R - \frac{\rho}{2} \log \frac{m(\alpha)}{m(\beta)}]$$

*Proof.* Consider the translation-invariant dual formulation (74): if $(f, g)$ are optimal, then for any $\lambda \in \mathbb{R}$, $(f + \lambda, g - \lambda)$ are also optimal. We leverage the structure of the dual constraint $f \oplus g \leq C_d$ with Lemma A.11. Since for $(x, y) \in \mathrm{supp}(\alpha) \times \mathrm{supp}(\beta)$, $C_d(x, y) \leq R$, then without loss of generality, $f(x) \in [0, R]$ and $g(y) \in [-R, R]$. The potentials $(f, g)$ are optimal for the translation-invariant dual energy, and we need a bound for the original dual functional (3). To this end, we leverage Lemma A.12 to compute the optimal

translation, such that $(f, g) = (f + \lambda^\star(f, g), g - \lambda^\star(f, g))$. Let $\bar{\alpha} = \alpha/m(\alpha)$ and $\bar{\beta} = \beta/m(\beta)$ be the normalized probability measures. The translation can be written as,

$$\lambda^\star(f, g) = \frac{1}{2}\left[S_\rho^{\bar{\beta}}(g) - S_\rho^{\bar{\alpha}}(f)\right] + \frac{\rho}{2}\log\frac{m(\alpha)}{m(\beta)}, \tag{75}$$

where the functional $S_\rho^\alpha$ is defined in Lemma A.12. Since $\bar{\alpha}$ and $\bar{\beta}$ are probability measures, then by (Genevay et al., 2019, Proposition 1), $f(x) \in [0, R]$ and $g(x) \in [-R, R]$ respectively imply $S_\rho^{\bar{\alpha}}(f) \in [0, R]$ and $S_\rho^{\bar{\beta}}(g) \in [-R, R]$. Combining these bounds on $S_\rho^{\bar{\alpha}}(f)$, $S_\rho^{\bar{\beta}}(g)$ with the expression of $\lambda^\star(f, g)$, (75) yields the desired bounds on the optimal potentials $(f, g)$ of the dual formulation (3).

$\square$

## A.6 Metrizing weak* convergence: Proof of Theorem 3.12

The Kullback-Leibler setting is treated here. The Partial OT setting (*i.e.*, $D_\varphi = \rho TV$) is treated in Appendix A.7.

*Proof.* Let $(\alpha_n)$ be a sequence of measures in $\mathcal{M}_+(X)$ and $\alpha \in \mathcal{M}_+(X)$, where $X \subset \mathbb{R}^d$ is compact with radius $R > 0$. First, we assume that $\alpha_n \rightharpoonup \alpha$. Then, by (Liero et al., 2018, Theorem 2.25), under our assumptions, $\alpha_n \rightharpoonup \alpha$ is equivalent to $\lim_{n \to +\infty} \mathrm{UOT}(\alpha_n, \alpha) = 0$. This implies that $\lim_{n \to +\infty} \mathrm{SUOT}(\alpha_n, \alpha) = 0$ and $\lim_{n \to +\infty} \mathrm{USOT}(\alpha_n, \alpha) = 0$, since by Theorem 3.11 and non-negativity of SUOT (Proposition 3.3),

$$0 \leq \mathrm{SUOT}(\alpha_n, \alpha) \leq \mathrm{USOT}(\alpha_n, \alpha) \leq \mathrm{UOT}(\alpha_n, \alpha).$$

Conversely, assume either that $\lim_{n \to +\infty} \mathrm{SUOT}(\alpha_n, \alpha) = 0$ or $\lim_{n \to +\infty} \mathrm{USOT}(\alpha_n, \alpha) = 0$. First assume there exists $M > 0$ such that for large enough $n \in \mathbb{N}^*, m(\alpha_n) \leq M$, then by Theorem 3.11, there exists $c > 0$ such that $\mathrm{UOT}(\alpha_n, \alpha) \leq c\big(\mathrm{SUOT}(\alpha_n, \alpha)\big)^{1/(d+1)}$. Since $c$ is doesn't depend on the masses $(m(\alpha_n), m(\alpha))$, it does not depend on $n$. By Theorem 3.11, it yields metric equivalence between SUOT, USOT and UOT, thus $\lim_{n \to +\infty} \mathrm{UOT}(\alpha_n, \alpha) = 0$. By (Liero et al., 2018, Theorem 2.25), we eventually obtain $\alpha_n \rightharpoonup \alpha$, which is the desired result.

The remaining step thus consists in proving that the sequence of masses $(m(\alpha_n))_{n \in \mathbb{N}^*}$ is indeed uniformly bounded by $M > 0$ for large enough $n$. Note that for any $(\alpha, \beta) \in \mathcal{M}_+(\mathbb{R}^d)$, one has $\mathrm{UOT}(\alpha, \beta) \geq \rho(\sqrt{m(\alpha)} - \sqrt{m(\beta)})^2$. Indeed one has $\mathrm{UOT}(\alpha, \beta) \geq \mathcal{D}(\lambda, -\lambda)$, where $\mathcal{D}$ denotes the dual functional (3) and $\lambda = \frac{\rho}{2}\log\frac{m(\alpha)}{m(\beta)}$. Note that the pair $(\lambda, -\lambda)$ are feasible dual potentials for the constraint $f \oplus g \leq C_d$, because the cost $C_d$ is positive in our setting. The property of push-forwards measures means that for any $\theta \in \mathbb{S}^{d-1}$, one has $m(\theta_\sharp^\star \alpha) = m(\alpha)$. Therefore, we obtain the following bounds for $n$ large enough.

$$\mathrm{USOT}(\alpha_n, \alpha) \geq \mathrm{SUOT}(\alpha_n, \alpha) \geq \int_{\mathbb{S}^{d-1}} \rho\left(\sqrt{m(\theta_\sharp^\star \alpha_n)} - \sqrt{m(\theta_\sharp^\star \alpha)}\right)^2 d\boldsymbol{\sigma}(\theta),$$
$$= \rho(\sqrt{m(\alpha_n)} - \sqrt{m(\alpha)})^2.$$

Hence, $\lim_{n \to +\infty} \mathrm{SUOT}(\alpha_n, \alpha) = 0$ or $\lim_{n \to +\infty} \mathrm{USOT}(\alpha_n, \alpha) = 0$ implies $\lim_{n \to +\infty} m(\alpha_n) = m(\alpha)$. In other terms the mass of sequence converges and is thus uniformly bounded for large enough $n$. Since we proved that $m(\alpha_n) < M$ and $m(\alpha)$ is finite, it ends the proof. $\square$

## A.7 Properties of sliced partial OT

We provide in this subsection the proofs of Proposition 3.3, Theorems 3.11 and 3.12 for the setting of sliced partial OT. To this end, we rely on a formulation for SUOT and USOT when $D_{\varphi_1} = D_{\varphi_2} = \rho TV$, which we prove below. Equation 76 is proved in (Piccoli & Rossi, 2014) and can then be applied to SUOT: we include it for completeness. Equation 77 is our contribution and is specific to USOT.

**Lemma A.14.** *Let $\rho > 0$ and assume $D_{\varphi_1} = D_{\varphi_2} = \rho\mathrm{TV}$ and $\mathrm{C}_d(x, y) = ||x - y||$. Then, for any $(\alpha, \beta) \in \mathcal{M}_+(\mathbb{R}^d)$,*

$$\mathrm{UOT}(\alpha, \beta) = \sup_{f \in \mathcal{E}} \int f(x)\mathrm{d}(\alpha - \beta)(x), \tag{76}$$

*where $\mathcal{E} = \{f : \mathbb{R}^d \to \mathbb{R}, ||f||_{Lip} \leq 1, ||f||_\infty \leq \rho\}$, $||f||_\infty \triangleq \sup_{x \in \mathbb{R}^d} |f(x)|$ and $||f||_{Lip} \triangleq \sup_{(x,y) \in \mathbb{R}^d} \frac{|f(x) - f(y)|}{\mathrm{C}_d(x,y)}$.*

*Furthermore, for $\mathrm{C}_1(x, y) = |x - y|$ and an empirical approximation $\hat{\boldsymbol{\sigma}}_N = \frac{1}{N} \sum_{i=1}^N \delta_{\theta_i}$ of $\boldsymbol{\sigma}$, one has*

$$\mathrm{USOT}(\alpha, \beta) = \sup_{(f_\theta) \in \mathcal{E}} \int_{\mathbb{R}^d} \left( \int_{\mathbb{S}^{d-1}} f_\theta(\theta^\star(x))\mathrm{d}\hat{\boldsymbol{\sigma}}_N(\theta) \right) \mathrm{d}(\alpha - \beta)(x), \tag{77}$$

*where*

$$\mathcal{E} = \{\forall \theta \in \mathrm{supp}(\hat{\boldsymbol{\sigma}}_N), \ f_\theta : \mathbb{R} \to \mathbb{R}, \ ||f_\theta||_{Lip} \leq 1, \ || \int_{\mathbb{S}^{d-1}} f_\theta \circ \theta^\star \mathrm{d}\hat{\boldsymbol{\sigma}}_N(\theta)||_\infty \leq \rho\},$$

*and the Lipschitz norm here is defined w.r.t. $\mathrm{C}_1$ as $||f||_{Lip} \triangleq \sup_{(x,y) \in \mathbb{R}^d} \frac{|f(x) - f(y)|}{\mathrm{C}_1(x,y)}$*

*Proof.* We start with the formulation of Equation 3 and Theorem 3.9. For USOT one has

$$\mathrm{USOT}(\alpha, \beta) = \sup_{f_\theta(\cdot) \oplus g_\theta(\cdot) \leq \mathrm{C}_1} \int \varphi_1^\circ \Big( \int_{\mathbb{S}^{d-1}} f_\theta(\theta^\star(x))\mathrm{d}\sigma_N(\theta) \Big) \mathrm{d}\alpha(x)$$
$$+ \int \varphi_2^\circ \Big( \int_{\mathbb{S}^{d-1}} g_\theta(\theta^\star(y))\mathrm{d}\sigma_N(\theta) \Big) \mathrm{d}\beta(y).$$

When $D_\varphi = \rho\mathrm{TV}$, the function $\varphi^\circ$ reads $\varphi^\circ(x) = x$ for $x \in [-\rho, \rho]$, $\varphi^\circ(x) = \rho$ when $x \geq \rho$, and $\varphi^\circ(x) = -\infty$ otherwise. Noting $f_{avg}(x) = \int_{\mathbb{S}^{d-1}} f_\theta(\theta^\star(x))\mathrm{d}\sigma_N(\theta)$ and $g_{avg}(x) = \int_{\mathbb{S}^{d-1}} g_\theta(\theta^\star(x))\mathrm{d}\sigma_N(\theta)$. This formula on $\varphi^\circ$ imposes $f_{avg}(x) \geq -\rho$ and $g_{avg}(x) \geq -\rho$. Furthermore, since we perform a supremum w.r.t. $(f_{avg}, g_{avg})$ where $\varphi^\circ$ attains a plateau, then without loss of generality, we can impose the constraint $f_{avg}(x) \leq \rho$ and $g_{avg}(x) \geq \rho$, as it will have no impact on the optimal dual functional value. Thus we have that $||f_{avg}||_\infty \leq \rho$ and $||g_{avg}||_\infty \leq \rho$. To obtain the Lipschitz property, we use the constraint that $f_\theta(\cdot) \oplus g_\theta(\cdot) \leq \mathrm{C}_1$ for any $\theta \in \mathrm{supp}(\sigma_N)$, as well as (Santambrogio, 2015, Proposition 3.1). Thus by using c-transform for the cost $\mathrm{C}_1(x, y) = |x - y|$, we can take w.l.o.g $f_\theta(\cdot) = -g_\theta(\cdot)$ with $f_\theta(\cdot)$ a 1-Lipschitz function. Thus w.l.o.g we can perform the supremum over $(f_\theta)_\theta \in \mathcal{E}$, and rephrase the functional as desired, since we have that $\varphi^\circ(f_{avg}) = f_{avg}$.

The proof for UOT is exactly the same, except that our inputs are $(f, g)$ instead of $(f_\theta, g_\theta)$. $\qquad\square$

We can now prove Proposition 3.3, Theorems 3.11 and 3.12 in the setting of sliced Partial OT. All those results are summarized in the following statement.

**Theorem A.15.** *(Properties of Sliced Partial OT) Assume $\mathrm{C}_1(x, y) = |x - y|$ and $D_{\varphi_1} = D_{\varphi_2} = \rho\mathrm{TV}$. Then, USOT satisfies the triangle inequality. Additionally, for any $(\alpha, \beta) \in \mathcal{M}_+(\mathsf{X})$ where $\mathsf{X} \subset \mathbb{R}^d$ is compact with radius $R$, $\mathrm{UOT}(\alpha, \beta) \leq c(\rho, R) \mathrm{SUOT}(\alpha, \beta)^{1/(d+1)}$, and USOT and SUOT both metrize the weak\* convergence.*

*Proof of Sliced Partial OT properties.* First we prove that in that setting USOT is a metric. Reusing Lemma A.14, we have that for any measures $(\alpha, \beta, \gamma)$

$$
\begin{aligned}
\mathrm{USOT}(\alpha, \gamma) &= \sup_{(f_\theta)_\theta \in \mathcal{E}} \int \left( \int_{\mathbb{S}^{d-1}} f_\theta(\theta^\star(x)) \mathrm{d}\sigma_N \right) \mathrm{d}(\alpha - \gamma)(x) \\
&= \sup_{(f_\theta)_\theta \in \mathcal{E}} \int \left( \int_{\mathbb{S}^{d-1}} f_\theta(\theta^\star(x)) \mathrm{d}\sigma_N \right) \mathrm{d}(\alpha - \beta + \beta - \gamma)(x) \\
&\leq \sup_{(f_\theta)_\theta \in \mathcal{E}} \int \left( \int_{\mathbb{S}^{d-1}} f_\theta(\theta^\star(x)) \mathrm{d}\sigma_N \right) \mathrm{d}(\alpha - \beta)(x) \\
&\quad + \sup_{(f_\theta)_\theta \in \mathcal{E}} \int \left( \int_{\mathbb{S}^{d-1}} f_\theta(\theta^\star(x)) \mathrm{d}\sigma_N \right) \mathrm{d}(\beta - \gamma)(x) \\
&= \mathrm{USOT}(\alpha, \beta) + \mathrm{USOT}(\beta, \gamma).
\end{aligned}
$$

Note that reusing Lemma A.14, we have that SUOT is a sliced integral probability metric over the space of bounded and Lipschitz functions. More precisely, we satisfy the assumptions of (Nadjahi et al., 2020b, Theorem 3), so that one has $\mathrm{UOT}(\alpha, \beta) \leq c(\rho, R)(\mathrm{SUOT}(\alpha, \beta))^{1/(d+1)}$.

To prove that USOT and SUOT metrize the weak* convergence, the proof is very similar to that of Theorem 3.12 detailed above. Assuming that $\alpha_n \rightharpoonup \alpha$ implies $\mathrm{SUOT}(\alpha_n, \alpha) \to 0$ and $\mathrm{USOT}(\alpha_n, \alpha) \to 0$ is already proved in Appendix A.6. To prove the converse, the proof is also the same, *i.e.*, we use the property that SUOT, USOT and UOT are equivalent metrics, which holds as we assumed that supports of $(\alpha, \beta)$ are compact in a ball of radius $R$. Note that since the bound $\mathrm{UOT}(\alpha, \beta) \leq c(\rho, R)(\mathrm{SUOT}(\alpha, \beta))^{1/(d+1)}$ holds independently of the measure's masses, we do not need to uniformly bound $m(\alpha_n)$, compared to the KL setting of Theorem 3.12.

$\square$

# B Additional details for Section 4

## B.1 Postponed Proofs for Section 4

*Proof of Proposition 4.1.* Our goal is to compute the first order variation of the SUOT functional. Given that $\mathrm{SUOT}(\alpha, \beta) = \int_{\mathbb{S}^{d-1}} \mathrm{UOT}(\theta^\star_\sharp \alpha, \theta^\star_\sharp \beta) \mathrm{d}\boldsymbol{\sigma}(\theta)$, one can apply Proposition B.1 slice-wise. Since measures are assumed to have compact support, one can apply the dominated convergence theorem and differentiate under the integral sign. Furthermore, the translation-invariant formulation in the setting of SUOT reads

$$
\mathrm{SUOT}(\alpha, \beta) = \int_{\mathbb{S}^{d-1}} \sup_{f_\theta \oplus g_\theta \leq \mathrm{C}_1} \left[ \sup_{\lambda_\theta \in \mathbb{R}} \int \varphi^\circ \Big( f_\theta(\cdot) + \lambda_\theta \Big) \mathrm{d}\theta^\star_\sharp \alpha \right. \tag{78}
$$

$$
\left. + \int \varphi^\circ \Big( g_\theta(\cdot) - \lambda_\theta \Big) \mathrm{d}\theta^\star_\sharp \beta \right], \tag{79}
$$

In the setting where $\varphi^\circ$ is smooth and strictly concave (such as $\mathrm{D}_\varphi = \rho \mathrm{KL}$), there always exists a unique optimal $\lambda^\star_\theta$. Furthermore, one can apply the envelope theorem such that the Fréchet differential w.r.t. to a perturbation $(r_\theta, s_\theta)$ of $(f_\theta, g_\theta)$ reads

$$
\int_{\mathbb{S}^{d-1}} \left[ \int r_\theta(\cdot) \times \nabla\varphi^\circ \Big( f_\theta(\cdot) + \lambda^\star_\theta(f_\theta, g_\theta) \Big) \mathrm{d}\theta^\star_\sharp \alpha \right. \tag{80}
$$

$$
\left. + \int s_\theta(\cdot) \times \nabla\varphi^\circ \Big( g_\theta(\cdot) - \lambda^\star_\theta(f_\theta, g_\theta) \Big) \mathrm{d}\theta^\star_\sharp \beta \right] \tag{81}
$$

Setting

$$
\alpha_\theta = \nabla\varphi^\circ \Big( f_\theta(\cdot) + \lambda^\star(f_\theta, g_\theta) \Big) \alpha, \qquad \beta_\theta = \nabla\varphi^\circ \Big( g_\theta(\cdot) - \lambda^\star(f_\theta, g_\theta) \Big) \beta,
$$

yields the desired result, *i.e.*, the first order variation is

$$\int_{\mathbb{S}^{d-1}} \left[ \int r_\theta(\cdot) \mathrm{d}(\theta_\sharp^\star \alpha_\theta) + \int s_\theta(\cdot) \mathrm{d}(\theta_\sharp^\star \beta_\theta) \right]. \tag{82}$$

$\square$

*Proof of Proposition 4.2.* Our goal is to compute the first order variation of the USOT functional. First, we leverage Theorem 3.9 such that USOT reads

$$\mathrm{USOT}(\alpha, \beta) = \sup_{f_\theta(\cdot) \oplus g_\theta(\cdot) \leq \mathrm{C}_1} \int \varphi_1^\circ \Big( \int_{\mathbb{S}^{d-1}} f_\theta(\theta^\star(x)) \mathrm{d}\hat{\boldsymbol{\sigma}}_K(\theta) \Big) \mathrm{d}\alpha(x) \tag{83}$$

$$+ \int \varphi_2^\circ \Big( \int_{\mathbb{S}^{d-1}} g_\theta(\theta^\star(y)) \mathrm{d}\hat{\boldsymbol{\sigma}}_K(\theta) \Big) \mathrm{d}\beta(y) \tag{84}$$

$$= \sup_{f_\theta(\cdot) \oplus g_\theta(\cdot) \leq \mathrm{C}_1} \int \varphi_1^\circ \Big( f_{avg}(x) \Big) \mathrm{d}\alpha(x) + \int \varphi_2^\circ \Big( g_{avg}(y) \Big) \mathrm{d}\beta(y), \tag{85}$$

where

$$f_{avg}(x) = \int_{\mathbb{S}^{d-1}} f_\theta(\theta^\star(x)) \mathrm{d}\hat{\boldsymbol{\sigma}}_K(\theta), \qquad g_{avg}(y) = \int_{\mathbb{S}^{d-1}} g_\theta(\theta^\star(y)) \mathrm{d}\hat{\boldsymbol{\sigma}}_K(\theta).$$

From this, we derive the translation-invariant formulation as follows.

$$\mathrm{USOT}(\alpha, \beta) = \sup_{f_\theta(\cdot) \oplus g_\theta(\cdot) \leq \mathrm{C}_1} \sup_{\lambda \in \mathbb{R}} \int \varphi_1^\circ \Big( f_{avg}(x) + \lambda \Big) \mathrm{d}\alpha(x) \tag{86}$$

$$+ \int \varphi_2^\circ \Big( g_{avg}(y) - \lambda \Big) \mathrm{d}\beta(y), \tag{87}$$

For smooth and strictly concave $\varphi^\circ$, there exists a unique $\lambda^\star(f_{avg}, g_{avg})$ attaining the supremum. Furthermore, one can apply the enveloppe theorem and differentiate under the integral sign (since the support is compact). Consider perturbations $(r_\theta(\cdot), s_\theta(\cdot))$ of $(f_\theta(\cdot), g_\theta(\cdot))$. Write

$$r_{avg}(x) = \int_{\mathbb{S}^{d-1}} r_\theta(\theta^\star(x)) \mathrm{d}\hat{\boldsymbol{\sigma}}_K(\theta), \qquad s_{avg}(y) = \int_{\mathbb{S}^{d-1}} s_\theta(\theta^\star(y)) \mathrm{d}\hat{\boldsymbol{\sigma}}_K(\theta).$$

Given that $\varphi_1^\circ(f_{avg} + r_{avg}) = \varphi_1^\circ(f_{avg}) + r_{avg} \nabla \varphi_1^\circ(f_{avg}) + o(\|r_{avg}\|_\infty)$, the first order variation reads

$$\int r_{avg}(x) \nabla \varphi_1^\circ \Big( f_{avg}(x) + \lambda^\star(f_{avg}, g_{avg}) \Big) \mathrm{d}\alpha(x) \tag{88}$$

$$+ \int s_{avg}(y) \nabla \varphi_2^\circ \Big( g_{avg}(y) - \lambda^\star(f_{avg}, g_{avg}) \Big) \mathrm{d}\beta(y). \tag{89}$$

Then we define

$$\bar{\alpha} = \nabla \varphi_1^\circ(f_{avg} + \lambda^\star(f_{avg}, g_{avg}))\alpha, \qquad \bar{\beta} = \nabla \varphi_2^\circ(g_{avg} - \lambda^\star(f_{avg}, g_{avg}))\beta,$$

such that the first order variation reads

$$\int r_{avg}(x) \mathrm{d}\bar{\alpha}(x) + \int s_{avg}(y) \mathrm{d}\bar{\beta}(y). \tag{90}$$

One can then explicit the definition of $(r_{avg}, s_{avg})$, such that it reads

$$\int_{\mathbb{S}^{d-1}} \int r_\theta(\theta^\star(x)) \mathrm{d}\bar{\alpha}(x) + \int_{\mathbb{S}^{d-1}} \int s_\theta(\theta^\star(y)) \mathrm{d}\bar{\beta}(y) \tag{91}$$

$$= \int_{\mathbb{S}^{d-1}} \int r_\theta \mathrm{d}\theta_\sharp^\star \bar{\alpha}(x) + \int_{\mathbb{S}^{d-1}} \int s_\theta \mathrm{d}\theta_\sharp^\star \bar{\beta}(y). \tag{92}$$

By optimizing the above over the constraint set $\{r_\theta \oplus s_\theta \leq \mathrm{C}_1\}$, we identify the computation of $\mathrm{SOT}(\bar{\alpha}, \bar{\beta})$, which concludes the proof.

$\square$

## B.2 Frank-Wolfe methodology for computing UOT

**Background: FW for UOT.** Our approach to compute SUOT and USOT takes inspiration from the construction of (Séjourné et al., 2022b). It consists in applying a Frank-Wolfe (FW) procedure over the dual formulation of UOT. Such approach is equivalent to solve a sequence of balanced OT problems between measures $(\tilde{\alpha}, \tilde{\beta})$ which are iterative renormalizations of $(\alpha, \beta)$. While the idea holds in wide generality, it is especially efficient in 1D where OT has low algorithmic complexity, and we reuse it in our sliced setting.

FW algorithm consists in optimizing a functional $\mathcal{H}$ over a compact, convex set $\mathcal{C}$ by optimizing its linearization $\nabla\mathcal{H}$. Given a current iterate $x^t$ of FW algorithm, one computes $r^{t+1} \in \arg\max_{r \in \mathcal{C}} \langle \nabla\mathcal{H}(x^t), r \rangle$, and performs a convex update $x^{t+1} = (1 - \gamma_{t+1})x^t + \gamma_{t+1}r^{t+1}$. One typically chooses the learning rate $\gamma_t = \frac{2}{2+t}$. This yields the routine `FWStep` of Section 4 which is detailed below.

---

**Algorithm 4** – `FWStep`$(f, g, r, s, \gamma)$

**Input:** $\alpha$, $\beta$, $f$, $g$, $\gamma$
**Output:** Normalized measures $(\alpha, \beta)$ as in (96)
   $f(x) \leftarrow (1 - \gamma)f(x) + \gamma r(x)$
   $g(y) \leftarrow (1 - \gamma)g(y) + \gamma s(y)$
   Return $(f, g)$

---

In the setting of UOT, one would take $\mathcal{C} = \{f \oplus g \leq C_d\}$. However, this set is not compact as it contains $(\lambda, -\lambda)$ for any $\lambda \in \mathbb{R}$. Thus, Séjourné et al. (2022b) propose to optimise a *translation-invariant* dual functional $\mathcal{H}(f, g; \alpha, \beta) \triangleq \sup_{\lambda \in \mathbb{R}} \mathcal{D}(f + \lambda, g - \lambda; \alpha, \beta)$, with $\mathcal{D}$ defined in (3). Similar to the balanced OT dual, one has $\mathcal{H}(f + \lambda, g - \lambda; \alpha, \beta) = \mathcal{H}(f, g; \alpha, \beta)$, thus one can apply (Santambrogio, 2015, Proposition 1.11) to assume w.l.o.g. that, *e.g.*, $f(0) = 0$ and restrict to a compact set of functions. We emphasize that FW algorithm is well-posed to optimize $\mathcal{H}$, but not $\mathcal{D}$.

Note that once we have the dual variables $(f, g)$ maximizing $\mathcal{H}$, we retrieve optimal dual variables maximizing $\mathcal{D}$ as $(f + \lambda^\star(f, g), g - \lambda^\star(f, g))$, where $\lambda^\star(f, g) \triangleq \arg\max_{\lambda \in \mathbb{R}} \mathcal{D}(f + \lambda, g - \lambda; \alpha, \beta)$. The KL setting where $D_{\varphi_1} = \rho_1 \text{KL}$ and $D_{\varphi_2} = \rho_2 \text{KL}$ is especially convenient, because $\lambda^\star(f, g)$ admits a closed form, which avoids iterative subroutines to compute it. In that case, it reads

$$\lambda^\star(f, g) = \frac{\rho_1 \rho_2}{\rho_1 + \rho_2} \log\left(\frac{\int e^{-f(x)/\rho_1} \mathrm{d}\alpha(x)}{\int e^{-g(y)/\rho_2} \mathrm{d}\beta(y)}\right). \tag{93}$$

We summarize the FW algorithm for UOT in the proposition below. We refer to (Séjourné et al., 2022b) for more details on the algorithm and pseudo-code. We adapt this approach and result for SUOT and USOT.

**Proposition B.1.** *(Séjourné et al., 2022b) Assume $\varphi^\circ$ is smooth. Given current iterates $(f^{(t)}, g^{(t)})$, the linear FW oracle of $\text{UOT}(\alpha, \beta)$ is $\text{OT}(\bar{\alpha}^{(t)}, \bar{\beta}^{(t)})$, where $\bar{\alpha}^{(t)} = \nabla\varphi^\circ(f^{(t)} + \lambda^\star(f^{(t)}, g^{(t)}))\alpha$ and $\bar{\beta}^{(t)} = \nabla\varphi^\circ(g^{(t)} - \lambda^\star(f^{(t)}, g^{(t)}))\beta$. In particular, one has $m(\bar{\alpha}^{(t)}) = m(\bar{\beta}^{(t)})$, thus the balanced OT problem always has finite value. More precisely, the FW update reads*

$$(f^{(t+1)}, g^{(t+1)}) = (1 - \gamma^{(t+1)})(f^{(t)}, g^{(t)}) + \gamma^{(t+1)}(r^{(t+1)}, s^{(t+1)}), \tag{94}$$

$$\text{where} \qquad (r^{(t+1)}, s^{(t+1)}) \in \arg\max_{r \oplus s \leq C_d} \int r(x) \mathrm{d}\bar{\alpha}^{(t)}(x) + \int s(y) \mathrm{d}\bar{\beta}^{(t)}(y). \tag{95}$$

Recall that the in KL setting one has $\varphi_i^\circ(x) = \rho_i(1 - e^{-x/\rho_i})$, thus $\nabla\varphi_i^\circ(x) = e^{-x/\rho_i}$. Thus in that case one normalizes the measures as

$$\bar{\alpha} = \exp\left(-\frac{f + \lambda^\star(f, g)}{\rho_1}\right)\alpha, \qquad \bar{\beta} = \exp\left(-\frac{g - \lambda^\star(f, g)}{\rho_2}\right)\beta, \tag{96}$$

where $\lambda^\star$ is defined in (93).

This defines the `Norm` routine in Algorithm 1.

### B.3  Implementation of Sliced OT to return dual potentials

Recall from Section 4, Algorithms 2 and 3 and more precisely, Propositions 4.1 and 4.2, that FW linear oracle is a sliced OT program, *i.e.*, a set of OT problems computed between univariate distributions of $\mathcal{M}_+(\mathbb{R})$. Therefore, a key building block of our algorithm is to compute the loss and dual variables of these univariate OT problems. We explain below how one can compute the sliced OT loss and dual potentials. The computation of the loss consists in implementing closed formulas of OT between univariate distributions, as detailed in (Santambrogio, 2015, Proposition 2.17). More precisely, when $C_1(x, y) = |x - y|^p$ and $(\mu, \nu) \in \mathcal{M}_+(\mathbb{R})$, then

$$\text{OT}(\mu, \nu) = \int_0^1 |F_\mu^{[-1]}(t) - F_\nu^{[-1]}(t)|^p \mathrm{d}t, \tag{97}$$

where $F_\mu^{[-1]}$ denotes the inverse cumulative distribution function (ICDF) of $\mu$.

---
**Algorithm 5** – $\texttt{SlicedOTLoss}(\alpha, \beta, \{\theta\}, p)$

---
**Input:** $\alpha$, $\beta$, projections $\{\theta\}$, exponent $p$
**Output:**  $\text{OT}(\theta_\sharp^\star \alpha, \theta_\sharp^\star \beta)$ as in (97)
  **for** $\theta \in \{\theta\}$ **do**
    Project support of $\theta_\sharp^\star \alpha$ and $\theta_\sharp^\star \beta$
    Sort weights of $(\theta_\sharp^\star \alpha, \theta_\sharp^\star \beta)$ and support $(\theta^\star(x))$, $(\theta^\star(y))$ s.t. support is non-decreasing
    Compute ICDF of $\theta_\sharp^\star \alpha$ and $\theta_\sharp^\star \beta$
    Compute $\text{OT}(\theta_\sharp^\star \alpha, \theta_\sharp^\star \beta)$ as in (97) with exponent $p$
  **end for**

---

To compute dual potentials using backpropagation, one computes the sliced OT losses (using Algorithm 5) then calls the backpropagation w.r.t to inputs $(\alpha, \beta)$, because their gradients are optimal dual potentials (Santambrogio, 2015, Proposition 7.17). We describe this procedure in Algorithm 6.

---
**Algorithm 6** – $\texttt{SlicedOTPotentialsBackprop}(\alpha, \beta, \{\theta\}, p)$

---
**Input:** $\alpha$, $\beta$, projections $\{\theta\}$, exponent $p$
**Output:**  Dual potentials $(f_\theta, g_\theta)$ solving $\text{OT}(\theta_\sharp^\star \alpha, \theta_\sharp^\star \beta)$
  Enable gradients w.r.t. $(\theta_\sharp^\star \alpha, \theta_\sharp^\star \beta)$
  Call $\texttt{SlicedOTLoss}(\alpha, \beta, \{\theta\}, p)$
  Sum (but do not average) losses $\mathcal{L} = \sum_\theta \text{OT}(\theta_\sharp^\star \alpha, \theta_\sharp^\star \beta)$.
  Backpropagate $\mathcal{L}$ w.r.t. $(\alpha, \beta)$
  Return $(f_\theta, g_\theta)$ as gradients of $\mathcal{L}$ w.r.t. $(\alpha, \beta)$.

---

The implementation of the dual potentials using 1D closed forms relies on the north-west corner rule principle, which can be vectorized in PyTorch in order to be computed in parallel. The contribution of our implementation thus consists in making such algorithm GPU-compatible and allowing for a parallel computation for every slice simultaneously. We stress that this constitutes a non-trivial piece of code, and we refer the interested reader to the code in our supplementary material for more details on the implementation.

### B.4  Output optimal sliced marginals

In all our algorithms, we focus on dual formulations of SUOT and USOT, which optimize the dual potentials. However, one might want the output variables of the primal formulation (See Definition 3.6). In particular, the marginals of optimal transport plans are interesting because they are interpreted as normalized versions of inputs $(\alpha, \beta)$ where geometric outliers have been removed. We detail where this interpretation comes from in the setting of UOT, and then give how it is adapted to SUOT and USOT. In particular, we justify that the $\texttt{Norm}$ routine suffices to compute them.

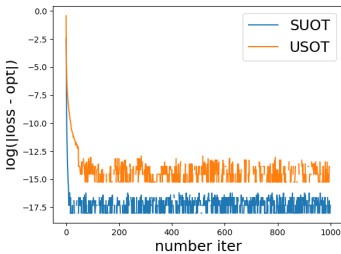 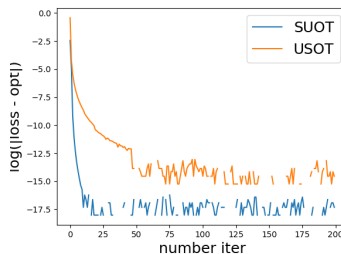 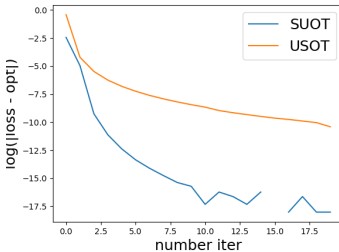

Figure 5: $|\text{SUOT}(\alpha,\beta) - \widehat{\text{SUOT}}_t|$ and $|\text{USOT}(\alpha,\beta) - \widehat{\text{USOT}}_t|$ against iteration $t$, where $\widehat{\text{SUOT}}_t, \widehat{\text{USOT}}_t$ are the estimated SUOT, USOT using $t$ FW iterations. Plots are in log-scale. All figures are issued from the same run, but zoomed on a subset of first iterations: *(left)* 1000 iterations of FW, *(middle)* 200 iterations, *(right)* 20 iterations.

**Case of UOT.** We focus on the $\text{D}_{\varphi_i} = \rho_i \text{KL}$. As per (Liero et al., 2018, Equation 4.21), we have at optimality that the optimal transport $\pi^\star$ plan solving $\text{UOT}(\alpha,\beta)$ as in (2) has marginals $(\pi_1^\star, \pi_2^\star)$ which read $\pi_1^\star = e^{-f^\star/\rho_1}\alpha$ and $\pi_2^\star = e^{-g^\star/\rho_2}\beta$, where $(f^\star, g^\star)$ are the optimal dual potentials solving (3). Since on $\text{supp}(\pi^\star)$ one also has $f^\star(x) + g^\star(y) = \text{C}_d(x,y)$, if the transportation cost $\text{C}_d(x,y)$ is large (*i.e.*, we are matching a geometric outlier), so are $f^\star(x)$ and $g^\star(y)$, and eventually the weights $\pi_1^\star(x)$ and $\pi_2^\star(y)$ are small, hence the interpretation of the geometric normalization of the measures. Note that in that case, one obtain $(\pi_1^\star, \pi_2^\star)$ by calling $\text{Norm}(\alpha,\beta,f^\star,g^\star,\rho_1,\rho_2)$.

**Case of SUOT.** Since $\text{SUOT}(\alpha,\beta)$ consists in integrating $\text{UOT}(\theta_\sharp^\star\alpha, \theta_\sharp^\star\beta)$ w.r.t. $\boldsymbol{\sigma}$, it shares many similarities with UOT. For any $\theta$, we consider $\pi_\theta$ and $(f_\theta, g_\theta)$ solving the primal and dual formulation of $\text{UOT}(\theta_\sharp^\star\alpha, \theta_\sharp^\star\beta)$. The marginals of $\pi_\theta$ are thus given by $(e^{-f_\theta/\rho_1}\alpha, e^{-g_\theta/\rho_2}\beta)$. In particular, we retrieve the observation made in Figure 1 that the optimal marginals change for each $\theta$. In that case we call for each $\theta$ the routine $\text{Norm}(\alpha,\beta,f_\theta,g_\theta,\rho_1,\rho_2)$.

**Case of USOT.** Recall that the optimal marginals $(\pi_1, \pi_2)$ in $\text{USOT}(\alpha,\beta)$ do not depend on $\theta$, contrary to $\text{SUOT}(\alpha,\beta)$. Leveraging the dual formulation of Theorem 3.9, and looking at the Lagrangian which is defined in the proof of Theorem 3.9 (see Appendix A.2), we have the optimality condition that $\pi_1 = e^{-f_{avg}/\rho_1}\alpha$ and $\pi_2 = e^{-g_{avg}/\rho_2}\beta$. Thus in that case, calling $\text{Norm}(\alpha,\beta,f_{avg},g_{avg},\rho_1,\rho_2)$ yields the desired marginals.

### B.5 Convergence of Frank-Wolfe iterations: Empirical analysis

We display below an experiment on synthetic dataset to illustrate the convergence of Frank-Wolfe iterations. We also provide insights on the number of iterations that yields a reasonable approximation: a few iterations suffices in our practical settings, typically $F = 20$.

The results are displayed in Figure 5. We consider the empirical distributions $(\alpha,\beta)$ computed over respectively, $N = 400$ and $M = 500$ samples over the unit hypercube $[0,1]^d$, $d = 10$. Moreover, $\beta$ is slightly shifted by a vector of uniform coordinates $0.5 \times \mathbf{1}_d$. We choose $\rho = 1$ and report the estimation of $\text{SUOT}(\alpha,\beta)$ and $\text{USOT}(\alpha,\beta)$ through Frank-Wolfe iterations. We estimate the true values by running $F = 5000$ iterations, and display the difference between the estimated score and the 'true' values. Appendix B.5 shows that numerical precision is reached in a few tens of iterations. As learning tasks do not usually require an estimation of losses up to numerical precision, we think that it is hence reasonable to take $F \approx 20$ in numerical applications.

Table 2: Dataset characteristics.

|                     | BBCSport      | Movies       | Goodreads genre | Goodreads like |
| ------------------- | ------------- | ------------ | --------------- | -------------- |
| Doc                 | 737           | 2000         | 1003            | 1003           |
| Train               | 517           | 1500         | 752             | 752            |
| Test                | 220           | 500          | 251             | 251            |
| Classes             | 5             | 2            | 8               | 2              |
| Mean words by doc   | $116 \pm 54$  | $182 \pm 65$ | $1491 \pm 538$  | $1491 \pm 538$ |
| Median words by doc | 104           | 175          | 1518            | 1518           |
| Max words by doc    | 469           | 577          | 3499            | 3499           |

## C   Additional details on Section 5

### C.1   Document classification: Technical details and additional results

#### C.1.1   Datasets

We sum up the statistics of the different datasets in Table 2.

**BBCSport.**   The BBCSport dataset (Kusner et al., 2015) contains articles between 2004 and 2005, and is composed of 5 classes. We average over the 5 same train/test split of (Kusner et al., 2015). The dataset can be found in `https://github.com/mkusner/wmd/tree/master`.

**Movie Reviews.**   The movie reviews dataset (Pang et al., 2002) is composed of 1000 positive and 1000 negative reviews. We take five different random 75/25 train/test split. The data can be found in `http://www.cs.cornell.edu/people/pabo/movie-review-data/`.

**Goodreads.**   This dataset, proposed in (Maharjan et al., 2017), and which can be found at `https://ritual.uh.edu/multi_task_book_success_2017/`, is composed of 1003 books from 8 genres. A first possible classification task is to predict the genre. A second task is to predict the likability, which is a binary task where a book is said to have success if it has an average rating $\geq 3.5$ on the website Goodreads (`https://www.goodreads.com`). The five train/test split are randomly drawn with 75/25 proportions.

#### C.1.2   Technical Details

All documents are embedded with the Word2Vec model (Mikolov et al., 2013) in dimension $d = 300$. The embedding can be found in `https://drive.google.com/file/d/0B7XkCwpI5KDYNlNUTTlSS21pQmM/view?resourcekey=0-wjGZdNAUop6WykTtMip3Og`.

In this experiment, we report the results averaged over 5 random train/test split. For discrepancies which are approximated using random projections, we additionally average the results over 3 different computations, and we report this standard deviation in Table 1. Furthermore, we always use 500 projections to approximate the sliced discrepancies. For Frank-Wolfe based methods, we use 10 iterations, which we found to be enough to have a good accuracy. We added an ablation of these two hyperparameters in Figure 7. We report the results obtained with the best $\rho$ for USOT and SUOT computed among a grid $\rho \in \{10^{-4}, 5 \cdot 10^{-4}, 10^{-3}, 5 \cdot 10^{-3}, 10^{-2}, 10^{-1}, 1\}$. For USOT, the best $\rho$ is consistently around $5 \cdot 10^{-3}$ for the Movies and Goodreads datasets, and around $5 \cdot 10^{-4}$ for the BBCSport dataset. We used a second finer grid and reported the results obtained with $\rho = 0.00021$ on BBCSport, $\rho = 0.004$ for Goodreads on the likability task and $\rho = 0.003$ for the genre task. For SUOT, the best $\rho$ obtained was 0.01 for the BBCSport dataset, 1.0 for the movies dataset and 0.5 for the goodreads dataset. For UOT, we used $\rho = 1.0$ on the BBCSport dataset. For the movies dataset, the best $\rho$ obtained on a subset was 50, but it took an unreasonable amount of time to run on the full dataset as the runtime increases with $\rho$ (see (Chapel et al., 2021, Figure 3)). On the goodreads dataset, it took too much memory on the GPU. For Sinkhorn UOT, we used $\varepsilon = 0.1$ and $\rho = 1.0$ on the BBCSport and $\varepsilon = 0.001$, $\rho = 0.1$ on the Goodreads dataset, and $\varepsilon = 0.01$ and $\rho = 0.1$ on the Movies dataset. Note

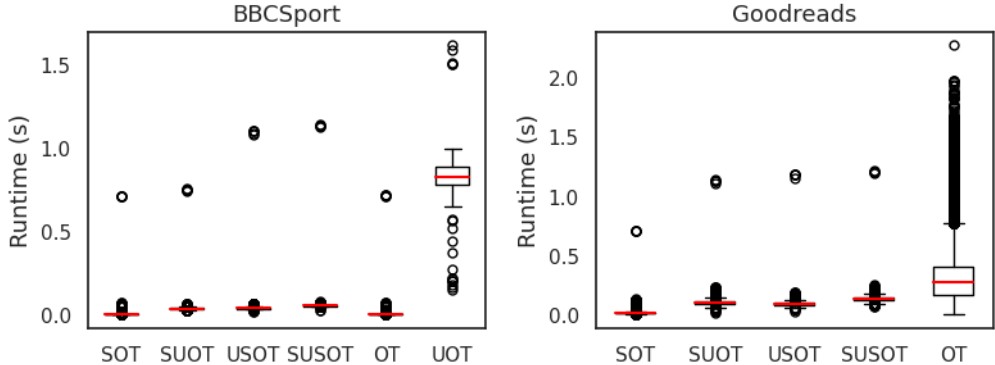

Figure 6: Runtime on the BBCSport dataset *(left)* and on the Goodreads dataset *(right)*.

that we also tested other set of parameters for UOT and Sinkhorn UOT, but on a coarser grid than USOT and SUOT given their computational time. For each method, the number of neighbors used for the k-NN method is obtained via cross-validation.

### C.1.3 Additional experiments

**Runtime.** We report in Figure 6 the runtime of computing the different discrepancies between each pair of documents, and in Table 3 the full runtimes. On the BBCSport dataset, the documents have in average 116 words, thus the main bottleneck is the projection step for sliced OT methods. Hence, we observe that OT runs slightly faster than SOT and the sliced unbalanced counterparts. Goodreads is a dataset with larger documents, with on average 1491 words by document. Therefore, as OT scales cubically with the number of samples, we observe here that all sliced methods run faster than OT, which confirms that sliced methods scale better w.r.t. the number of samples. In this setting, we were not able to compute UOT with the POT implementation in a reasonable time. Computations have been performed with a NVIDIA Tesla V100 GPU.

Table 3: Runtimes on Document Classification

|  |  | **BBCSport** | **Goodreads** |
|---|---|---|---|
| OT | Average ($\cdot 10^{-3}$ s) | $3.29_{\pm 1.61}$ | $440.30_{\pm 259}$ |
|  | Full (s) | 891 | 221252 |
| SOT | Average ($\cdot 10^{-3}$s) | $1.80_{\pm 6.22}$ | $4.49_{\pm 1.44}$ |
|  | Full (s) | 487 | 2256 |
| USOT | Average ($\cdot 10^{-3}$s) | $14.67_{\pm 1.29}$ | $14.45_{\pm 0.88}$ |
|  | Full (s) | 3897 | 7260 |
| SUOT | Average ($\cdot 10^{-3}$s) | $13.9_{\pm 1.21}$ | $14.32_{\pm 0.95}$ |
|  | Full (s) | 3770 | 7193 |

**Ablations.** We plot in Figure 7 accuracy as a function of the number of projections and the number of iterations of the Frank-Wolfe algorithm. We averaged the accuracy obtained with the same setting described in Appendix C.1.2, with varying number of projections $K \in \{4, 10, 21, 46, 100, 215, 464, 1000\}$ and number of FW iterations $F \in \{1, 2, 3, 4, 5, 10, 15, 20\}$. Regarding the hyperparameter $\rho$, we selected the one returning the best accuracy, *i.e.*, $\rho = 5 \cdot 10^{-4}$ for USOT and $\rho = 10^{-2}$ for SUOT.

**Choice of $\rho$.** In Table 4, we also add the results obtained when $\rho$ is chosen by cross validation for USOT and SUOT. In this case, the results are slightly below the best one, but are still better than SOT.

**Unnormalizing measures.** As we have mentioned in Section 5, we have performed document classification experiments by normalizing word histograms to be probabilities. It allows to compare SUOT and USOT with SOT since sliced OT is only well-defined between probabilities. However, it seems reasonnable to compare

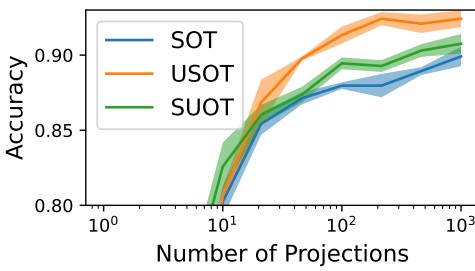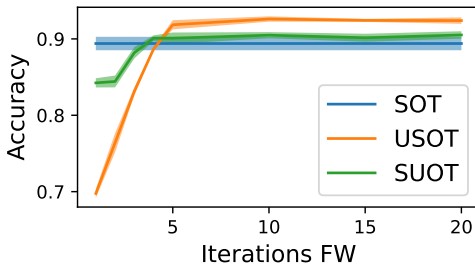

Figure 7: Ablation on BBCSport of the number of projections *(left)* and of the number of Frank-Wolfe iterations *(right)*.

Table 4: Accuracy on document classification. Grid-search over $\rho$ is performed. We report the best accuracy over $\rho$, which corresponds to $\rho = 5.10^{-6}$ for Unnormalized USOT and $\rho = 5.10^{-6}$ for SOPT.

|  | BBCSport | Movies | Goodreads genre | Goodreads like |
|---|---|---|---|---|
| OT | 94.55 | 74.44 | 55.22 | 71.00 |
| UOT | 96.73 | - | - | - |
| Sinkhorn UOT | 95.45 | 72.48 | 53.55 | 67.81 |
| SOT | $89.39_{\pm 0.76}$ | $66.95_{\pm 0.45}$ | $50.09_{\pm 0.51}$ | $65.60_{\pm 0.20}$ |
| SUOT | $90.12_{\pm 0.15}$ | $67.84_{\pm 0.37}$ | $50.15_{\pm 0.04}$ | $66.72_{\pm 0.38}$ |
| USOT | $93.52_{\pm 0.04}$ | $69.21_{\pm 0.37}$ | $52.67_{\pm 0.62}$ | $67.78_{\pm 0.39}$ |
| SUSOT | $92.73_{\pm 0.27}$ | $69.53_{\pm 0.53}$ | $51.93_{\pm 0.53}$ | $67.33_{\pm 0.26}$ |
| SUOT (+CV on $\rho$) | $90.00_{\pm 0.59}$ | $67.40_{\pm 0.64}$ | $49.67_{\pm 0.79}$ | $66.43_{\pm 0.44}$ |
| USOT (+CV on $\rho$) | $92.61_{\pm 0.55}$ | $68.64_{\pm 0.29}$ | $52.06_{\pm 7.20}$ | $66.61_{\pm 0.72}$ |
| USOT (Unnormalized) | $86_{\pm 0.56}$ | - | - | - |
| SOPT (Unnormalized) | $87.27_{\pm 0.20}$ | - | - | - |

with other unbalanced sliced methods such as SOPT (Bai et al., 2023). We chose to compare with this competitor since their code is available in Python. However, a numerical restriction of their algorithm is that it only outputs measures with constants weights, *i.e.*, distributions $\alpha = \sum \alpha_i \delta_{x_i}$ and $\beta = \sum \beta_j \delta_{y_j}$ where $\alpha_i = \beta_j = 1$, but the number of samples in $\alpha$ and $\beta$ may differ. Under this modeling assumption, the total mass of each measure corresponds to the number of words in the sentence. We performed the comparison on the BBC dataset, using 500 projections for both SOPT and USOT. Unfortunately, the quadratic footprint of computing the similarity kernel does not scale reasonably for SOPT for larger datasets such as Movies or Goodreads, especially because their algorithm is not GPU-compatible compared to ours. We cross-validated the parameter $\rho \in \{p.10^{-k}, \ k \in [\![0,6]\!], \ p \in \{1,.5.\}\}$

The result is detailed in the table below. What is noticeable is that the performance degrades for both USOT and SOPT using this parametrization. Furthermore, we observed that the paramater $\rho$ yielding the best accuracy is much smaller for unnormalized measures than for the best one for normalized histograms (*i.e.*, $1e-5$ here compared to $1e-3$ with normalized measures). Our interpretation of this observation is that considering unnormalized measures adds an additional information of the sentence length via the masses of $(\alpha, \beta)$. It seems that this additional information dominates the comparison of measures, instead of focusing on the measures support (*i.e.*, the word embedding) which encodes the semantic information of words. When $\rho$ is large the kernel value of USOT/SOPT is mainly dictated by the mass (*i.e.*, sentence length) comparison. Thus smaller $\rho$ seems to give less importance on sentence length, hence a better performance. We also note that performance of SOPT and USOT on unnormalized measures are rather similar. It means that for the choice of marginal prior $D_\varphi = \rho TV$ or $D_\varphi = \rho KL$ does not significantly matter for this specific task, compared to the preprocessing normalization of measures.

## C.2 Unbalanced sliced Wasserstein barycenters

We define below the formulation of the USOT barycenter which was used in the experiments of Figure 4 to average predictions of geophysical data. We then detail how we computed it.

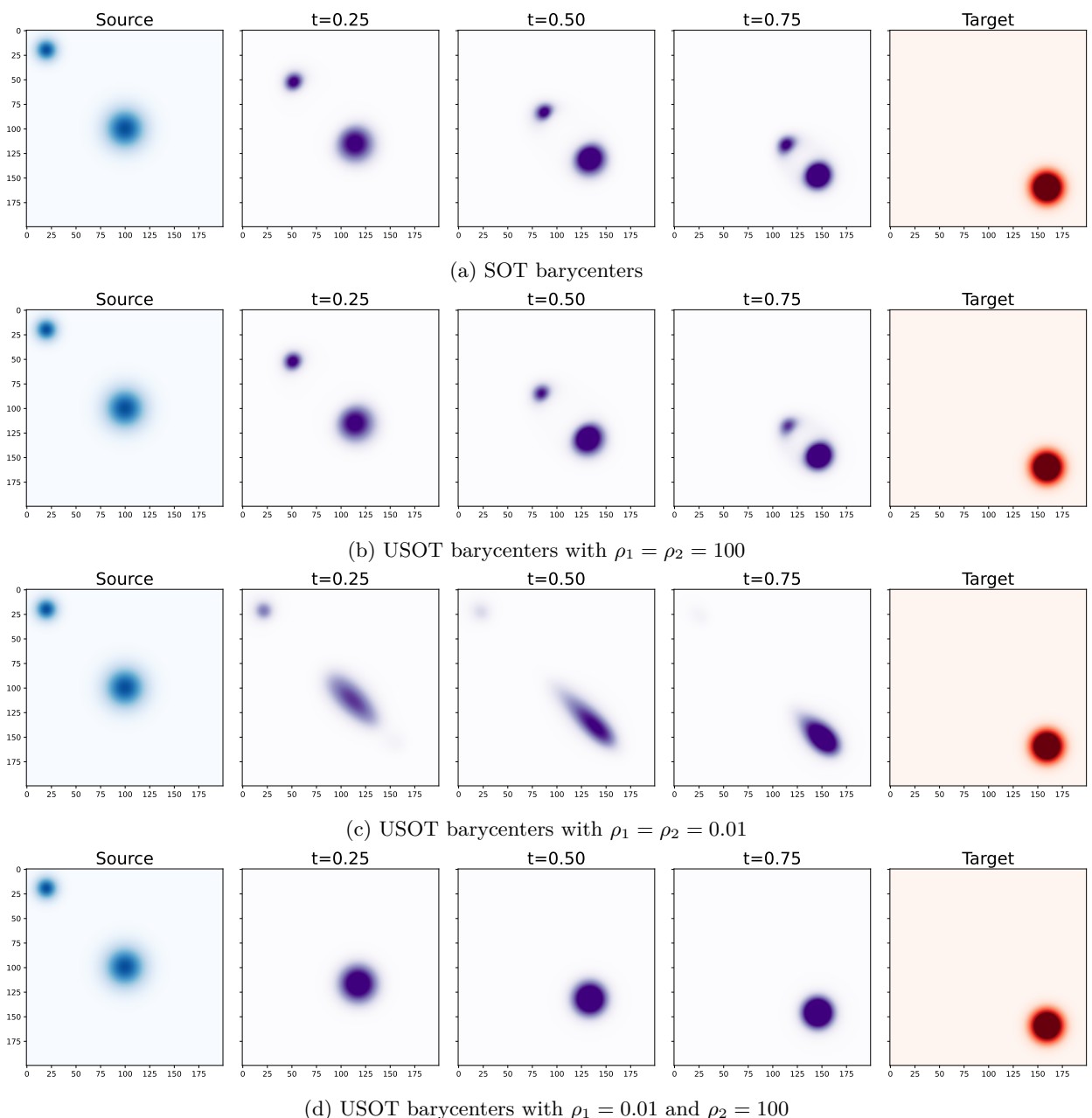

Figure 8: **Interpolation with USOT as a barycenter computation**. We compare different interpolations using SOT or USOT with different settings for the $\rho$ values

**Definition C.1.** *Consider a set of measures* $(\alpha_1, \ldots, \alpha_B) \in \mathcal{M}_+(\mathbb{R}^d)^B$, *and a set of non-negative coefficients* $(\omega_1, \ldots, \omega_B) \geq 0$ *such that* $\sum_{b=1}^B \omega_b = 1$. *We define the barycenter problem (in the* KL *setting) as*

$$\mathcal{B}((\alpha_b)_b, (\omega_b)_b) \triangleq \inf_{\beta \in \mathcal{P}(\mathbb{R}^d)} \sum_{b=1}^B \omega_b \mathrm{USOT}(\alpha_b, \beta), \tag{98}$$

$$= \inf_{\beta \in \mathcal{P}(\mathbb{R}^d)} \sum_{b=1}^B \inf_{(\pi_{b,1}, \pi_{b,2})} \mathrm{SOT}(\pi_{b,1}, \pi_{b,2}) + \rho_1 \mathrm{KL}(\pi_{b,1}|\alpha_b) + \rho_2 \mathrm{KL}(\pi_{b,2}|\beta), \tag{99}$$

*where* $\mathcal{P}(\mathbb{R}^d)$ *denotes the set of probability measures.*

To compute the barycenter, we aggregate several building blocks. First, since we consider that the barycenter $\beta \in \mathcal{P}(\mathbb{R}^d)$ is a probability, we perform mirror descent as in (Beck & Teboulle, 2003; Cuturi & Doucet, 2014b). More precisely, we use a Nesterov accelerated version of mirror descent. We also tried projected gradient descent, but it did not yield consistent outputs (due to convergence speed (Beck & Teboulle, 2003)). Second, we use a Stochastic-USOT version (see Section 4), *i.e.*, we sample new projections at each iteration of the barycenter update (but not a each iteration of the FW subroutines in Algorithm 3). This procedure is described in Algorithm 7.

---

**Algorithm 7** – $\texttt{Barycenter}((\alpha_b)_b, (\omega_b)_b, \rho_1, \rho_2, lr)$

---

**Input:** measures $(\alpha_b)_b$, weights $(\omega_b)_b$, $\rho_1$, $\rho_2$, learning rate $lr$, FW iter $F$
**Output:** Optimal barycenter $\beta$ of (98)

$\quad t \leftarrow 1$
$\quad$ Init $(\beta, \tilde{\beta}, \hat{\beta})$ as uniform distribution over a grid
$\quad$ **while** not converged do **do**
$\quad\quad \gamma \leftarrow \frac{2}{(t+1)}$,
$\quad\quad \beta \rightarrow (1-\gamma)\hat{\beta} + \gamma\tilde{\beta}$
$\quad\quad$ Sample projections $(\theta_k)_{k=1}^K$
$\quad\quad$ Compute $\mathcal{B}((\alpha_b)_b, (\omega_b)_b)$ by calling $\text{USOT}(\alpha_b, \beta, F, (\theta_k)_{k=1}^K, \rho_1, \rho_2)$ in Algorithm 3 for each $b$
$\quad\quad$ Compute $g$ as the gradient of $\mathcal{B}((\alpha_b)_b, (\omega_b)_b)$ w.r.t. variable $\beta$
$\quad\quad \tilde{\beta} \leftarrow \exp(-lr \times \gamma^{-1} \times g)\beta$
$\quad\quad \tilde{\beta} \leftarrow \tilde{\beta}/m(\tilde{\beta})$
$\quad\quad \hat{\beta} \leftarrow (1-\gamma)\hat{\beta} + \gamma\tilde{\beta}$
$\quad\quad t \leftarrow t+1$
$\quad$ **end while**

---

We illustrate this algorithm with several examples of interpolation in Figure 8. We propose to compute an interpolation between two measures located on a fixed grid of size $200 \times 200$ with different values of $\rho_i$ in $\text{D}_{\varphi_i} = \rho_i\text{KL}$. For illustration purposes, we construct the *source* distribution as a mixture of two Gaussians with a small and a larger mode, and the *target* distribution as a single Gaussian. Those distributions are normalized over the grid such that both total norms are equal to one (which is not required by our unbalanced sliced variants but grants more interpretability and possible comparisons with SOT). Figure 8a shows the result of the interpolation at three timestamps ($t = 0.25$, $0.5$ and $0.75$) of a SOT interpolation (within this setting, $\omega_1 = 1 - t$ and $\omega_2 = t$). As expected, the two modes of the source distribution are transported over the target one. We verify in Figure 8b that for a large value of $\rho_1 = \rho_2 = 100$, the USOT interpolation behaves similarly as SOT, as expected from the theory. When $\rho_1 = \rho_2 = 0.01$, the smaller mode is not moved during the interpolation, whereas the larger one is stretched toward the target (Figure 8c). Finally, in Figure 8d, an asymmetric configuration of $\rho_1 = 0.01$ and $\rho_2 = 100$ allows to get an interpolation when only the big mode of the source distribution is displaced toward the target. In all those cases, the mirror-descent algorithm 7 is run for 500 iterations. Even for a large grid of $200 \times 200$, those different results are obtained in a $2 - 3$ minutes on a commodity GPU, while the OT or UOT barycenters are untractable with a limited computational budget.

### C.3 Unbalanced version of hyperbolic SOT.

To illustrate the modularity of our FW algorithm, we aim at comparing synthetic mixtures of Wrapped Normal Distribution on the 2-hyperbolic manifold $\mathbb{H}$ (Nagano et al., 2019), so that the FW oracle is hyperbolic sliced OT (Bonet et al., 2023c). The parameter $\theta$ characterizes on $\mathbb{H}$ any geodesic curve passing through the origin, and each sample is projected by taking the shortest path to such geodesics. Once projected on a geodesic curve, we sort data and compute SOT w.r.t. hyperbolic metric $d_{\mathbb{H}}$. We consider the 2-hyperbolic manifold on the Poincaré disc. As illustrated in Figure 9, the input measure $\alpha$ (in red) is a mixture of 3 isotropic normal distributions, with a mode at the top of the disc playing the role of an outlier. The measure $\beta$ is a mixture of two anisotropic normal distributions, whose means are close to two modes of $\alpha$, but are slightly shifted at the disc's center. We show on Figure 9 the impact of the parameter $\rho = \rho_1 = \rho_2$ on the optimal marginals of USOT.

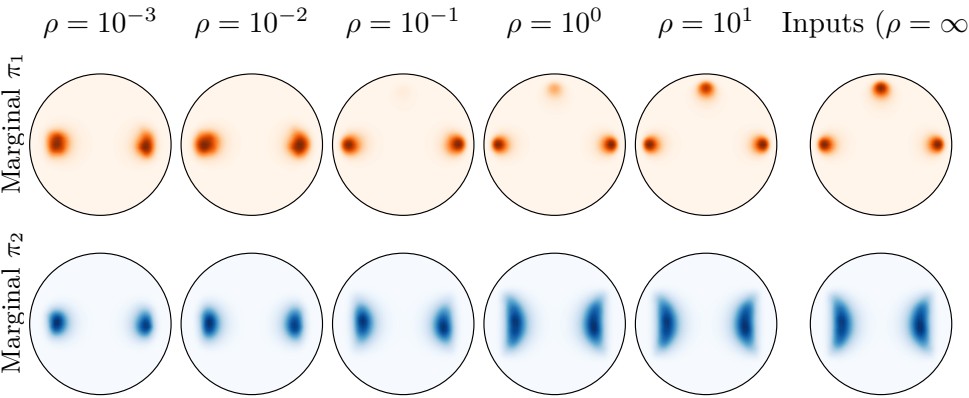

Figure 9: KDE estimation (kernel $e^{-d_{\mathbb{H}}^2/\sigma}$) of optimal $(\pi_1, \pi_2)$ of USOT$(\alpha, \beta)$ when $D_{\varphi_i} = \rho$KL.

This experiment illustrates several take-home messages, mentioned in Section 3. First, the optimal marginals $(\pi_1, \pi_2)$ are renormalisation of $(\alpha, \beta)$ accounting for their geometry, which are able to remove outliers for properly tuned $\rho$. When $\rho$ is large, $(\pi_1, \pi_2) \simeq (\alpha, \beta)$ and we retrieve SOT. When $\rho$ is too small, outliers are removed, but we see a shift of the modes, so that modes of $(\pi_1, \pi_2)$ are closer to each other, but do not exactly correspond to those of $(\alpha, \beta)$. Second, note that such plot cannot be made with SUOT, since the optimal marginals depend on the projection $\theta$ (see Figure 1). Third, we emphasize that we are indeed able to reuse any variant of SOT.

### C.4 Choice and interpretation of hyperparameter $\rho$

An immediate drawback of our framework is the induced additional computational cost w.r.t. SOT. While the above experimental results show that SUOT and USOT improve performance significantly over SOT, and though the complexity is still sub-quadratic in number of samples, our FW approach uses SOT as a subroutine, rendering it necessarily more expensive. Additionally, another practical burden comes from the introduction of extra hyperparameters $(\rho_1, \rho_2)$, which may be tuned using cross-validation. Therefore, a future direction would be to derive efficient strategies to tune $(\rho_1, \rho_2)$, maybe w.r.t. the applicative context, and further complement the possible interpretations of $\rho$ as a 'threshold' for the geometric information encode by $C_1$, $C_d$. While we leave the automation of tuning $(\rho_1, \rho_2)$ for future works, we provide below some details and intuitions on the choice of $\rho$ for the previous experiments. We hope these insights will help the practitioner on how they should chose tune this additional parameter.

**General intuition on $\rho$.** The parameter $\rho$ when $\rho_1 = \rho_2 = \rho$ can be understood as a *characteristic distance* to decide whether or not two sample should be matched by the coupling $\pi$ in the primal formulation of (2). Typically, transportation happens for samples $(x, y)$ such that $C_d(x, y) \leq \rho$, while samples such that $C_d(x, y) \geq \rho$ are interpreted as geometric outliers, and are discarded in the matching $\pi(x, y)$. In the case of SUOT and USOT, there is somehow a similar interpretation, but not for the same quantities, and we rely on their definitions (Equations 7 and 10), as well as the constraint set $\mathcal{E}$ in Theorem 3.9.

One sees that for SUOT$(\alpha, \beta)$ we have a set of 1D-UOT problems between $(\theta_\sharp^\star \alpha, \theta_\sharp^\star \beta)$, thus the threshold interpretation holds depending on whether $C_1(\theta^\star(x), \theta^\star(y)) \leq \rho$ or $C_1(\theta^\star(x), \theta^\star(y)) \geq \rho$. In particular the dependence in $\theta$ explains why the outlier threshold depends on the considered projection. Note also we consider $C_1$ instead of $C_d$.

For USOT$(\alpha, \beta)$ it is different because the marginals $(\pi_1, \pi_2)$ which we optimize in Equation (9) are independent of $\theta$, and common to all projections. Informally speaking, we interpret that the threshold value to discard a matching between $(x, y)$ depends on whether some quantity proportional to $\int_{\mathbb{S}^{d-1}} C_1(\theta^\star(x), \theta^\star(y)) d\pi_\theta(x, y) d\sigma(\theta)$ is larger or smaller than $\rho$. This quantity is not properly defined as it depends on the optimized variables $(\pi_\theta)_\theta$, hence the informality of our intuition. However, we wish to emphasize that the parameter $\rho$ should be interpreted differently between SUOT and USOT. As highlighted

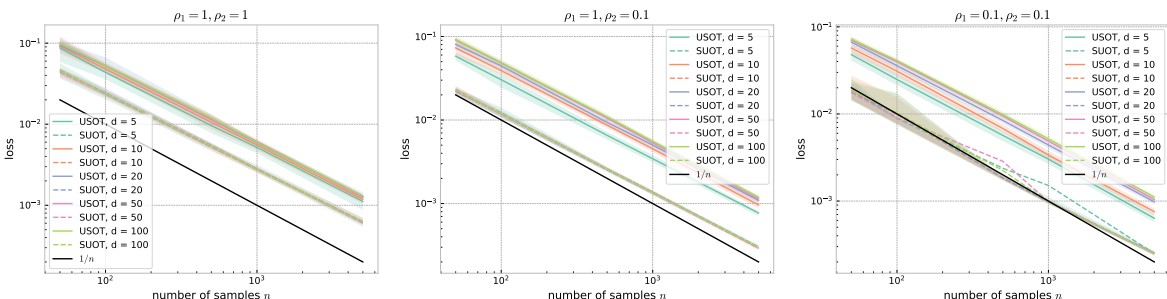

Figure 10: **Sample complexity:** $\mathbb{E}[|\mathcal{L}(\hat{\alpha}_n, \hat{\beta}_n) - \mathcal{L}(\alpha, \beta)|]$ against $n$, with $\mathcal{L} = $ SUOT or USOT and $\alpha = \beta = \mathcal{N}(\mathbf{0}_d, \mathbf{I}_d)$ (thus $\mathcal{L}(\alpha, \beta) = 0$). Results are averaged over 20 runs and the shaded areas represent the 10th and 90th percentiles. All curves exhibit the same convergence rate for any $d$, that is $1/n$ (see solid line black curve).

experimentally for document classification in Figure 2, we observe that the value of $\rho$ yielding the best performance is not the same for each loss.

**Choice of $\rho$ for hyperbolic data.** In Figure 9, the hyperbolic distance between overlapping modes is 0.96, while distance from side modes to the top red outlier is 2.83. Thus, a proper choice of should lie in between, which seems consistent with the observation of Figure 9 for $\rho = 1$. Indeed we see that we have a satisfying trade-off between removing the top mode and preserving the crescent shape structure of main blue modes.

**Choice of $\rho$ for barycenter experiments.** For the barycenter, we used insights from Figure 8 which interpolates circular blobs using asymmetric $(\rho_1, \rho_2)$, where $\rho_1$ is the parameter penalizing the input measures fidelity, and $\rho_2$ the parameter of the barycenter. For Figure 4 (especially line (d)), we also took assymetric $(\rho_1, \rho_2)$ with large $\rho_2 = 1e4$ for the barycenter to force data matching. Then for inputs $\rho_1 = 1e1$ is roughly the distance between cyclones (see Figure 4), to keep them in the barycenter. All in all, we force the barycenter to match the cyclone structure which matters most, while any structure who would be beyond this $\rho_1$ distance between input measure would be discarded.

**Interpretation of $\rho$ for document classification.** In this task the measures' support are given by word embedding in high dimension, for which we have no intuition of what is for instance the characteristic distance between different semantic clusters, and thus no idea on how $\rho$ should be tuned. For this reason (and more generally in ML tasks), we need to perform a cross-validation over this hyperparameter. We would like to comment the dependence of the document classification accuracy w.r.t. $\rho$, which can be observed in Figure 2. One can notice that as $\rho$ increases, the accuracy increases until it reaches a 'peak', until then it decreases to reach a plateau as $\rho \to \infty$. When $\rho \to \infty$, SUOT and USOT converge to SOT (see Definitions 2.4 and 3.6), and we get similar performances. As $\rho \to 0$, marginals $(\pi_1, \pi_2)$ are allowed to differ significantly from inputs $(\alpha, \beta)$, meaning that SUOT/USOT almost ignore input data. Therefore, $\rho$ should be tuned to extract information from inputs while removing noise. In Figure 2, the 'peaks' correspond to such optimal $\rho$, and the gain in performance justify the use of SUOT/USOT over SOT.

### C.5 Illustration of the sample complexity

We investigate the sample complexity of SUOT and USOT in practice and report the results in Figure 10. Our goal is to empirically verify Theorem 3.4 for SUOT, and explore the convergence rate for USOT. To this end, we consider $\alpha = \beta = \mathcal{N}(\mathbf{0}_d, \mathbf{I}_d)$ and compute $\text{SUOT}(\alpha_n, \beta_n)$ and $\text{USOT}(\alpha_n, \beta_n)$ for different number of samples $n$ and dimension $d$. This allows us to explore the convergence rate of $\text{SUOT}(\alpha_n, \beta_n)$ to $\text{SUOT}(\alpha, \beta) = 0$ (respectively, of $\text{USOT}(\alpha_n, \beta_n)$ to $\text{USOT}(\alpha, \beta) = 0$) as a function of $n$ and $d$.

Figure 10 shows that all curves share the same slope w.r.t $n$, for any $d$ and for both SUOT and USOT. This experiment is consistent with the dimension-free rate we established in Theorem 3.4 for SUOT. Interestingly, it also reveals that the dimension-free rate holds for USOT as well in that specific setting. More experiments and/or theoretical justification are needed to verify if this holds for more general distributions.

