# OpenReview forum: "Slicing Unbalanced Optimal Transport"
_TMLR — Accepted by TMLR_

### Review · Reviewer_vipQ · 2024-09-24

**Summary Of Contributions:**

This paper investigates two versions of the sliced unbalanced optimal transport problem, which applies the slicing technique to reformulate unbalanced optimal transport. Some theoretical explorations and comparisons have been explored. The Frank-Wolfe algorithms have also been adapted to solving the sliced unbalanced optimal transport problem. Numerical experiments have been conducted to validate the efficiency of the proposed formulations.

**Audience:**

Yes

**Claims And Evidence:**

Yes

**Requested Changes:**

Please address the major concerns in the previous section. Also, the following are some other questions or requested changes:
1. Page 1: Computing UOT ....in the number if n samples O(n^3logn). Please add a reference for this claim.
2. Page 2: while being statistically more efficient...Why is it statistically efficient?
3. Page 3: $\alpha\otimes\beta$, what does $\otimes$ mean?
4. Could Theorem 3.11 be extended to USOT?
5. Page 8: adapting the reasoning...? What does this mean?

**Strengths And Weaknesses:**

Strength: The paper is well-written and easy to follow. I enjoy reading this paper. The paper is theoretically interesting and rigorous. The authors have made a good try to explore the slicing unbalanced optimal transport.

Weakness:
1. The contributions are not clear

a: From my understanding, the SUOT has been proposed in the literature, while the USOT is newly proposed by the authors. I wonder if the authors could discuss the additional advantages of USOT over SUOT. For example, the authors mention that outliers could be removed by USOT. Does that mean USOT is more robust towards outliers? Such discussions could increase the contributions of this work. Otherwise, the authors are making non-surprising extensions from SUOT to USOT.

b. The goal of slicing unbalanced optimal transport should not be simply combining these two notions but what new insights could be brought. One of them could be: new frameworks of OT which could achieve robustness and computational efficiency at the same time.

2. The slicing technique is applied to accelerate the computation of UOT problems. This is good, but can the properties of UOT could be preserved? The authors are suggested to discuss from this perspective. For example, in the numerical experiments, in addition to evaluating the ACC and time, other properties like the robustness of SUOT and USOT could be explored and compared with UOT.

---

> ### Author Response · Authors · 2024-11-04
> **Response to Reviewer vipQ (Part 1)**
>
> Thank you for your feedback. We appreciate that you enjoyed reading our paper and that you found our theoretical contributions "interesting" and "rigorous".
>
> - "From my understanding, the SUOT has been proposed in the literature, while the USOT is newly proposed by the authors."
>
> We would like to precise that SUOT, as defined in our submission, has *not* been proposed in the literature: as underlined in Section 3.1, it has been defined for $D_\varphi = TV$ only. Therefore, one of our contributions is to define a more generic framework, by allowing the use of any $\varphi$-divergences and studying their impact on the theoretical properties.
>
>
> - "I wonder if the authors could discuss the additional advantages of USOT over SUOT. For example, the authors mention that outliers could be removed by USOT. Does that mean USOT is more robust towards outliers?."
>
> This is a relevant question which we address in Sections 3 and 4, and particularly in Section 3.3. Our empirical results indeed suggest that USOT is more robust towards outliers.
>
> - "Such discussions could increase the contributions of this work. Otherwise, the authors are making non-surprising extensions from SUOT to USOT"
>
> We agree that our contributions are consistent with prior related work, which support their soundness. However, they do not follow immediately from the literature: while we rely on previous important relevant works on unbalanced OT, notably to adapt some proof techniques, these adaptations are not straightforward. We have added the following discussion to clarify this point.
>
> First, Sejourne et al. (2022) [6] focus on optimization and introduce a Frank-Wolfe method to compute UOT between univariate measures. Adapting their approach to compute our losses does not follow from a direct application of their work: we made several theoretical and methodological contributions which are non-trivial, even in light of [1, 2, 6]. Specifically, we derived the duals of SUOT and USOT, which are the core elements of our FW algorithms (Theorems 3.5 and 3.9).
>
> Moreover, the literature on slicing in unbalanced OT is very limited. To the best of our knowledge, it was reduced to Bonneel & Coeurjolly (2019) [1] and Bai et al. (2022) [2], whose main contribution are methodological and focused on a specific choice of divergence. Our analysis complements this line of work, as we examine a broader set of properties (e.g., sample complexity, weak convergence metrization) for a larger class of divergences. Additionally, our USOT loss introduces a novel strategy to combine slicing with unbalancing OT, which has never been studied before.
>
> - "The goal of slicing unbalanced optimal transport should not be simply combining these two notions but what new insights could be brought. One of them could be: new frameworks of OT which could achieve robustness and computational efficiency at the same time."
>
> We agree, and argue that our work aligns well with the goal of developing "new frameworks of OT which could achieve robustness and computational efficiency at the same time": see Section 4.3 for a discussion on computational efficiency, and Section 3.3 and 5 for empirical results illustrating the improved robustness and complexity of our frameworks.
>
> We are happy to address any additional comments on this point that could help clarify and further support this message.
>
> - "The slicing technique is applied to accelerate the computation of UOT problems. This is good, but can the properties of UOT be preserved? The authors are suggested to discuss from this perspective."
>
> We agree that conserving the properties of UOT such as being robust to outliers is important. In Figure 1, we showed experimentally that USOT has a similar behaviour than UOT, as it is able to get rid from the outliers of the original marginals (in contrast with SUOT). We also argue that it is why USOT outperforms SUOT on the Document Classification task.
>
> - "Page 1: Computing UOT ....in the number if n samples O(n^3logn). Please add a reference for this claim."
>
> We have added the references [3, 4].
>
> - "Page 2: while being statistically more efficient...Why is it statistically efficient?"
>
> The statistical efficiency should be understood in terms of sample complexity: SUOT/USOT enjoys a dimension-free sample complexity even for multivarite measures (Theorem 3.4, Section C.5), while the rates for UOT suffer from an exponential dependence on the dimension [5].
>
> - "Page 3: $\alpha\otimes\beta$, what does $\otimes$ mean?"
>
> $\alpha\otimes\beta$ denotes the product measure.
>
> - "Could Theorem 3.11 be extended to USOT?"
>
> Yes: by Theorem 3.10, for any $\varphi$-divergences and $(\alpha,\beta) \in\mathcal{M}\_{+}(\mathbb{R}^d) \times \mathcal{M}\_{+}(\mathbb{R}^d)$, $\mathrm{SUOT}(\alpha,\beta)\le \mathrm{USOT}(\alpha, \beta)$. Therefore, Theorem 3.11 also holds for USOT. We have added a remark in our submission to clarify this.

---

> ### Author Response · Authors · 2024-11-04
> **Response to Reviewer vipQ (Part 2)**
>
> - "Page 8: adapting the reasoning...? What does this mean?"
>
> This means that to develop a Frank-Wolfe algorithm for computing SUOT (or USOT), we followed this strategy:
> 1. We formulated a translation-invariant version of the duals of SUOT (or USOT)
> 2. We then applied the Frank-Wolfe algorithm to this translation-invariant dual version.
>
> This strategy was proposed in [6] but for a different purpose, i.e., to compute UOT between univariate measures, thus resulting in a different dual and algorithm than ours. We have clarified this in page 8.
>
>
> [1] Bonneel, N., & Coeurjolly, D. (2019). Spot: sliced partial optimal transport. ACM Transactions on Graphics (TOG), 38(4), 1-13.
>
> [2] Bai, Y., Schmitzer, B., Thorpe, M., & Kolouri, S. (2023). Sliced optimal partial transport. In Proceedings of the IEEE/CVF Conference on Computer Vision and Pattern Recognition (pp. 13681-13690).
>
> [3] Peyré, G., & Cuturi, M. (2019). Computational optimal transport: With applications to data science. Foundations and Trends® in Machine Learning, 11(5-6), 355-607.
>
> [4] Pele, O., & Werman, M. (2009, September). Fast and robust earth mover's distances. In 2009 IEEE 12th international conference on computer vision (pp. 460-467). IEEE.
>
> [5] Manupriya, P., Nath, J. S., & Jawanpuria, P. (2020). MMD-Regularized Unbalanced Optimal Transport. arXiv preprint arXiv:2011.05001.
>
> [6] Séjourné, T., Vialard, F. X., & Peyré, G. (2022, May). Faster unbalanced optimal transport: Translation invariant sinkhorn and 1-d frank-wolfe. In International Conference on Artificial Intelligence and Statistics (pp. 4995-5021). PMLR.

---

> ### Comment · Reviewer_vipQ · 2024-11-11
> **Response to the Author**
>
> Thanks for the authors' clarifications. It would be better if the authors could also make these messages more direct and clear in their paper.

---

### Review · Reviewer_u6CG · 2024-10-16

**Summary Of Contributions:**

This paper merges the two variants of relaxing optimal transports (OT) -- unbalanced OT (UOT) and sliced OT (SOT).
Depending on which relaxation (U or S) is applied first, we can consider SUOT or USOT as possible outcomes of their combination.
The authors study theoretical properties of both SUOT and USOT and give algorithms for computing them.
They test SUOT/USOT and their algorithm on several datasets and confirm the computational efficiency of them, especially compared to the usual OT or UOT.

**Audience:**

Yes

**Broader Impact Concerns:**

I do not have any concerns in this regard.

**Claims And Evidence:**

Yes

**Requested Changes:**

Major:
- My strong concern is on the presentation of Theorem 3.4. I can see what the authors mean by this statement, but it should be specified how the sequences $\hat{\alpha}, \hat{\beta}, \hat{\mu}$ are generated, how they are related, if they are random, etc. The current statement does not make sense. Also, I believe that such constants $\kappa(n)$ does not exist unless you consider a proper subset of $\mathcal{M}_+$. If my intuition is wrong, the authors should prove in which sense we can expect such constant exists; otherwise the conclusion might be meaningless.

Minor:
- Many of the notations are incrementally added in the theorem statements. A summary at the early stage of the draft would be helpful.
- "characteristic radius" is written in Italic but not much mentioned afterwards. What is the intuition/importance of this?

**Strengths And Weaknesses:**

After defining SUOT and USOT, this paper extensively studies their theoretical properties including duality, equivalence, and so on. Especially, the equivalence among UOT, SUOT, USOT shown in Theorem 3.10 & 3.11 is important in that we can effectively use SUOT/USOT instead of UOT for fatster computation. They also give Frank-Wolfe algorithms for SUOT and USOT, and confirm that the proposed algorithm can work efficiently in practice. Overall, I think this paper provides a good theoretical/algorithmic contribution to the field and has many potential readers.

A weakness might be in its relatively limited description of related works. Since the resulting SUOT/USOT objectives are straightforward if one wants to merge them, it would be better to explain some strong motivation for merging them and/or the differences from existing studies. For example, we cannot see what a partial OT is and how its limitation is crucial (unit mass points), especially when we compute relatives of OT on empirical measures. The significance of this paper's contribution is thus a bit unclear.
This being said, the paper seems to provide sound theoretical analyses and should be a good fit to the scope of TMLR.

---

> ### Author Response · Authors · 2024-11-04
> **Response to Reviewer u6CG (Part 1)**
>
> Thank you for your feedback and your positive comments about our contributions.
>
> - "A weakness might be in its relatively limited description of related works."
>
> We aimed to provide a comprehensive review of relevant literature, for instance by comparing our general framework to sliced partial OT, but we would be grateful to include any additional references that you believe would enhance our discussion.
>
> - "Since the resulting SUOT/USOT objectives are straightforward if one wants to merge them, it would be better to explain some strong motivation for merging them and/or the differences from existing studies."
>
> We clarify the motivation and novelty of our work.
>
>
> The concepts of sliced and unbalanced optimal transport have generated significant interest in the machine learning community. Combining these two approaches for improved robustness and computational complexity is a natural question that has only been partially addressed: prior studies were limited to specific choices of divergences or cost functions [1, 2]. To expand the current literature, we propose two novel unbalanced OT losses that incorporate slicing, reducing the computational complexity from cubic to nearly linear, $O(n \log n)$. Our framework is therefore more general than existing methods.
>
> Moreover, while SUOT can be viewed as a straightforward generalization of previous work (specifically, sliced partial OT), we argue that USOT introduces a fundamentally new approach. Our experimental results further suggest that USOT is more effective than SUOT at filtering outliers (see Section 3.3).
>
> - "we cannot see what a partial OT is and how its limitation is crucial (unit mass points), especially when we compute relatives of OT on empirical measures."
>
> We apologize for the confusion: partial OT corresponds to unbalanced OT regularized with total variation (see Section 3.1).
>
> While partial OT is defined for general positive measures, it has been convenient to consider univariate measures where all points share the same mass (e.g., $\alpha = \frac{1}{n} \sum_{i=1}^n \delta_{x_i}$) to develop efficient approximation algorithms:
>
>
> 1. In [1], this allows optimizing over the set of injection mappings for each slice, justifying the heuristic used to approximate sliced partial OT;
> 2. In [2], this ensures the existence of an optimal plan [2, Theorem 4.1], which inspired their algorithm for approximating sliced optimal partial transport. However, it remains unclear whether their analysis can be adapted to non-uniform measures due to the optimality conditions in [2, Equation (6)].
>
> As a result, unlike USOT/SUOT, these algorithms assign unit weights to particles, which can be limiting in some applications. This is illustrated in our document classification experiment, where sliced partial OT yielded lower accuracy compared to our methods (see Table 4, and Section C.1.3 for our detailed discussion on the unit mass points constraint).
>
> - "My strong concern is on the presentation of Theorem 3.4. I can see what the authors mean by this statement, but it should be specified how the sequences $\hat{\alpha},\hat{\beta},\hat{\mu}$ are generated, how they are related, if they are random, etc."
>
> We considered the classical setting for studying sample complexity: given a continuous measure $\alpha$, its empirical approximation over $n$ i.i.d. samples $(Z_i)\_{i=1}^n$ is defined as $\hat{\alpha}\_n=\frac{1}{n}\sum\_{i_1}^n \delta_{Z\_i}$ (see section "Background"). Note that empirical approximations are random measures. We have added this clarification in the statement of Theorem 3.4.
>
> - "Also, I believe that such constants $\kappa(n)$ does not exist unless you consider a proper subset of $\mathcal{M}_+$. If my intuition is wrong, the authors should prove in which sense we can expect such constant exists; otherwise the conclusion might be meaningless."
>
> The existence of $\kappa(n)$ indeed depends on several factors, such as the choice of $\varphi$-divergences and cost functions, and may require working within a subset of measures. For instance, the rates in [3] apply to unbalanced OT with the Kullback-Leibler divergence, for measures with compact and convex supports and differentiable densities. Note that while their results are presented as upper-bounds on a pseudo-distance between potentials, denoted as $d^{\lambda}_{H^\circ}$, they also give the sample complexity $\mathbb{E}[|UOT(\hat{\alpha}_n, \hat{\beta}_n) - UOT(\alpha, \beta)|]$: see the proof of Proposition 3, and in particular, equation (22).
>
> Finally, we would like to emphasize that the purpose of Theorem 3.4 is to demonstrate the benefits of slicing for unbalanced OT, namely ensuring a dimension-free sample complexity (thereby extending the conclusions of prior work [4]). In this context, establishing more general or refined rates $\kappa(n)$ is not the primary focus of our submission, as it is not required for the conclusion above to hold.

---

> ### Author Response · Authors · 2024-11-04
> **Response to Reviewer u6CG (Part 2)**
>
> - "Minor: Many of the notations are incrementally added in the theorem statements. A summary at the early stage of the draft would be helpful."
>
> Thank you for your careful reading: we have added a summary of our notations in the "Background" section.
>
> - ""characteristic radius" is written in Italic but not much mentioned afterwards. What is the intuition/importance of this?"
>
> The characteristic radius is the term used to denote the hyperparameter $\rho$, which plays an important role in the performance of unbalanced OT methods, for instance by influencing the proportion of outliers that can be discarded. To avoid any confusion around this terminology, we have removed the italics.
>
>
> [1] Bonneel, N., & Coeurjolly, D. (2019). Spot: sliced partial optimal transport. ACM Transactions on Graphics (TOG), 38(4), 1-13.
>
> [2] Bai, Y., Schmitzer, B., Thorpe, M., & Kolouri, S. (2023). Sliced optimal partial transport. In Proceedings of the IEEE/CVF Conference on Computer Vision and Pattern Recognition (pp. 13681-13690).
>
> [3] Vacher, A., & Vialard, F. X. (2023, July). Semi-Dual Unbalanced Quadratic Optimal Transport: fast statistical rates and convergent algorithm. In International Conference on Machine Learning (pp. 34734-34758). PMLR.
>
> [4] Nadjahi, K., et al. Statistical and Topological Properties of Sliced Probability Divergences. NeurIPS 2020.

---

> ### Comment · Reviewer_u6CG · 2024-11-04
> **On Theorem 3.4**
>
> Thank you for your detailed comments. I am satisfied except for the presentation of Theorem 3.4. I generally understand the intention that this theorem is just for demonstration. However, if you would like to write "Theorem" rather than "Remark", I believe  at least some mathematical fomality should be satsified. My comments for the revised version are as follows:
> - What is the relation between $\hat{\mu_n}$ and $\mu$, and $\hat{\alpha_n}$ and $\alpha$? Does this notation suggest an implicitly defined function $(\ \hat{}, n) :\mathcal{M_+} \to \mathcal{M_+}; \mu \mapsto \hat{\mu_n}$? If so, this is not a very good notation, since there is no intuition explained before the theorem statement or agreement on the use of such a notation -- it is a lot better defining a sequence of mappings $H_n$ and discuss the convergence of $H_n(\mu) \to \mu$. If you do not like the functional notation, you should at least say something about how $\hat{\mu_n}$ is defined (like given by a discretization scheme parametrized by $n$ or whatever) and how it is consistent when applied to other measures ($\alpha, \beta$).
> - For the cited corollary in the revised manuscript, the published version of the paper should be referred to; it is Corollary 3.4 in https://proceedings.mlr.press/v202/vacher23a/vacher23a.pdf , right? Also, the conference infomation is missing in the reference.
> - Finally, I recommend simply taking a subset $M\subset \mathcal{M_+}$ and argue the convergence over $M$ or $M\times M$ -- it is simply stronger (more general) than the original statement as $M$ can take $\mathcal{M_+}$, and you can avoid the risk of confusion on the existence of such constants for each specific $(\ \hat, n)$ over the whole space. Also, the current statement does not cover the case you mentioned: ".. may require working within a subset of measures."
>
> Edit) I noticed that, since we are considering $d$-dimensional case upon the one-dimensional assumption, the situation is more complicated -- I am not sure how we should understand the correspondence of mapping $H_n$ (I wrote above) between 1-dim and $d$-dim measures. Also, taking a subset $M$ might be a bad idea, because there is not a canonical way to consider a $d$-dimensional counterpart. But then, how can it be formalized?
>
> For example, for the first part (the correspondence of $H_n$ between 1-dim and $d$-dim; let me write $H_n^1$ and $H_n^d$), you are implicitly assuming $\theta_\sharp^* H_n^d(\alpha) = H_n^1(\theta_\sharp^* \alpha_n)$. I think this is more like a definition/condition that should be stated in the theorem statement.

---

> ### Comment · Reviewer_u6CG · 2024-11-04
> **So sorry that I misread the manuscript - Theorem 3.4**
>
> I deeply apologize for that I missed you already pointed out $\hat{\alpha_n}$ is the empirical measure.. Most of my concern/misunderstandings got resolved. So please just ignore the point on the notation regarding $H_n$ etc in the previous comment including the (Edit) part. But one thing:
>
> - Since we are considering positive Radon measures, should we adapt empirical measures to non-probability measures? Maybe just by scaling when the measure is finite, but there is an infinite Radon measure: Lebesgue measure on R is such an example. How do we define $\hat{\alpha_n}$ for all the measures in $\mathcal{M_+}$?
>
> The remaining parts are the last two points in the previous comment:
> > For the cited corollary in the revised manuscript, the published version of the paper should be referred to; it is Corollary 3.4 in https://proceedings.mlr.press/v202/vacher23a/vacher23a.pdf , right? Also, the conference infomation is missing in the reference.
>
> > Finally, I recommend simply taking a subset $M\subset \mathcal{M_+}$ and argue the convergence over $M$ or $M\times M$ -- it is simply stronger (more general) than the original statement as $M$ can take $\mathcal{M_+}$, and you can avoid the risk of confusion on the existence of such constants for each specific $(\ \hat, n)$ over the whole space. Also, the current statement does not cover the case you mentioned: ".. may require working within a subset of measures."
>
> For the latter, while the extention of such an $M \subset \mathcal{M_+}(R)$ to $d$-dimensions is not straightforward, I think we can consider measures with all its marginal is in $M$?.
>
> Also, if the inequality is on the random sample, the inequality is usually in high probability or expectation regarding the draws of empirical measures, it should certainly be mentioned in the statement or around.
>
> Edit) I am again sorry, I noticed this last part has already been reflected.

---

> > ### Author Response · Authors · 2024-11-05
> > **Response to Reviewer u6CG on Theorem 3.4**
> >
> > Thank you for your prompt reply: we are glad to read that most of your concerns, in particular regarding the definition of the empirical measures, have been resolved.
> >
> > We appreciate your additional feedback on the exposition of Theorem 3.4 and have revised it. We provide detailed explanations below. Please let us know if there are any further aspects that need clarification.
> >
> > - "Since we are considering positive Radon measures, should we adapt empirical measures to non-probability measures? Maybe just by scaling when the measure is finite, but there is an infinite Radon measure: Lebesgue measure on R is such an example. How do we define $\hat{\alpha_n}$ for all the measures in $\mathcal{M}_+$?"
> >
> > As you correctly pointed out, sample complexity results for balanced OT can be extended to unbalanced OT for positive measures with finite mass (not necessarily equal to 1) by simply applying a rescaling. Extending our study to the setting of infinite measures would be interesting but is very challenging, as the (balanced) optimal transport problem between infinite measures is already very technical and a research topic on its own right: see e.g., [1, 2]. Moreover, it is unclear whether the particular case of infinite measures is of practical interest for the machine learning community. Therefore, our paper is restricted to the set of positive measures with finite mass -- a common assumption in the literature on OT. We have precised it in Section 2.1.
> >
> >
> > - "For the cited corollary in the revised manuscript, the published version of the paper should be referred to; it is Corollary 3.4 in https://proceedings.mlr.press/v202/vacher23a/vacher23a.pdf , right? Also, the conference infomation is missing in the reference."
> >
> > Thank you for noticing; we have updated the reference.
> >
> >
> > - "Finally, I recommend simply taking a subset $M \in \mathcal{M}_+$ and argue the convergence over $M$ or $M \times M$ ... I think we can consider measures with all its marginal is in $M$?"
> >
> > We agree and propose the following slight revision to Theorem 3.4(i): if for $\mu, \nu \in M \subset \mathcal{M}_+(\mathbb{R})$,
> >
> > $\mathbb{E}| UOT(\mu,\nu) - UOT(\hat{\mu}_n,\hat{\nu}_n) | \leq \kappa(n)$,
> >
> > then, for $\alpha, \beta \in \tilde{M} \subset \mathcal{M}_+(\mathbb{R}^d)$ such that
> >
> > $\theta_\sharp^\star \alpha, \theta_\sharp^\star \beta \in M$ for all $\theta \in \mathbb{S}^{d-1}$,
> >
> > $\mathbb{E}| \text{SUOT}(\alpha,\beta) - \text{SUOT}(\hat{\alpha}_n,\hat{\beta}_n) | \leq \kappa(n)$.
> >
> > This follows directly from equation (18). We have also revised Theorem 3.4(ii) accordingly, which is justified by equation (22).
> >
> > Thank you for these suggestions which made the exposition of Theorem 3.4 more rigorous, while preserving our conclusion on the statistical efficiency of our framework. This revised statement also makes the application of Theorem 3.4 to the GHK setting clearer: by choosing $\tilde{M}$ as the set of $d$-dimensional measures satisfying the constraints of [3, Corollary 3.4], one can show that the projected measures $(\theta_\sharp^\star \mu)_{\theta \in \mathbb{S}^{d-1}}$ also satisfy these assumptions, therefore $\kappa(n)$ is given by [3, Corollary 3.4] with $d = 1$.
> >
> > - "Also, if the inequality is on the random sample, the inequality is usually in high probability or expectation regarding the draws of empirical measures, it should certainly be mentioned in the statement or around."
> >
> > The inequality is on the expectation with respect to the samples of empirical measures. We have mentioned it below Theorem 3.4.
> >
> > [1] Huesmann, M., Sturm, K-T. "Optimal transport from Lebesgue to Poisson." Ann. Probab. 41 (4) 2426 - 2478, July 2013. https://doi.org/10.1214/12-AOP814
> >
> > [2] Huesmann, M. "Optimal transport between random measures." Ann. Inst. H. Poincaré Probab. Statist. 52 (1) 196 - 232, February 2016. https://doi.org/10.1214/14-AIHP634
> >
> > [3] Vacher, A., & Vialard, F. X. (2023, July). "Semi-Dual Unbalanced Quadratic Optimal Transport: Fast Statistical Rates and Convergent Algorithm." In International Conference on Machine Learning (pp. 34734-34758). PMLR.

---

> ### Comment · Reviewer_u6CG · 2024-11-06
> **Response to the revision of Theorem 3.4**
>
> Thank you for your immediate update. I think the exposition of the theorem is now very good, together with explanations before and after the statement. The only (minor) remaining thing from my side would be the definition of the empirical measure in Section 2.1, where you can just add the scaling factor to adapt to non-probability finite measures. But overall, all my essential concerns got resolved, thank you.

---

> > ### Author Response · Authors · 2024-11-06
> > **Thank you for your feedback**
> >
> > Thank you for carefully reviewing our update and for your additional comment. We have revised the notation for empirical measures in Section 2.1, replacing the uniform weights $\frac{1}{n}$​ with arbitrary weights $w_i​>0$.

---

### Review · Reviewer_xwi2 · 2024-10-26

**Summary Of Contributions:**

Optimal transport introduces a distance on probability measures but has limitations for high dimensional settings and measures with non-equal mass. Sliced optimal transport (SOT) (i.e., an average of the OT distance over uniform random 1d projections) and unbalanced optimal transport (UOT) try to address these drawbacks.
This paper combines those two techniques and investigates variants of sliced unbalanced optimal transport which has previously only been done in special cases. They propose two formulations that differ in the order in which the operations are performed, either unbalanced optimal transport of the projected measures is considered (SUOT) or they consider unbalanced optimal transport where the OT distance is replaced by its sliced version (USOT).
Leveraging results for SOT and UOT the authors then show that their variants enjoy favorable statistical properties and they propose an Frank-Wolfe type algorithm to compute these metrics in practice. Experiments show that the introduced methods and algorithms are competetive.

**Audience:**

Yes

**Claims And Evidence:**

Yes

**Requested Changes:**

Mostly questions:

- Does such a function $\kappa$ (and $\xi$) exist such that the rate holds uniformly for all measures?
- Weak convergence holds if for any bounded, continuous function ... (p. 7)
- Are the runtimes really only $10^{-3}s$ (in Table 1)? What do they correspond to, a single distance evaluation?
- In my understanding the hyperparameters are tuned differently for UOT and the sliced variant which appears a bit unfair, also I think that the cross validation based $\rho$ should be reported in the main text.
- Theorem 3.11 seems to be a bit more involved than most of the other results. It would be good if some proof intuition can be provided and if the authors could clarify what parts of the proof are identical to prior work and where the proof is adapted to the new setting.
- What is shown below Eq. (21) in the proof of Theorem 3.4

**Strengths And Weaknesses:**

- The paper is very well written and clear.
- The definitions are natural and the results are interesting
- Theory appears to be solid and experiments are OK.

- The paper does not seem to introduce seminal new ideas but mostly combines previous approaches (which is completely fine given the acceptance criteria)


Overall I enjoyed reading the paper and think it is a valuable contribution to TMLR.

---

> ### Author Response · Authors · 2024-11-04
> **Response to Reviewer xwi2**
>
> Thank you for your positive comments on our contributions. We appreciate your feedback and have addressed your questions below.
>
> - "Does such a function $\kappa$ (and $\xi$) exist such that the rate holds uniformly for all measures?"
>
> Yes, for instance, [1] establishes such rates for unbalanced OT based on quadratic costs and the Kullback-Leibler divergence as penalty (see their Corollary 3). Note that while their results are presented as upper-bounds on a pseudo-distance between potentials, denoted as $d^{\lambda}_{H^\circ}$, they also correspond to the sample complexity $\mathbb{E}[|UOT(\hat{\alpha}_n, \hat{\beta}_n) - UOT(\alpha, \beta)|]$: see the proof of Proposition 3, and in particular, equation (22).
>
> - "Weak convergence holds if for any bounded, continuous function ... (p. 7)"
>
> Thank you for spotting this: we have fixed the definition.
>
> - "Are the runtimes really only $10^{-3}s$ (in Table 1)? What do they correspond to, a single distance evaluation?"
>
> We reported in Table 1 the average runtime of an evaluation between two documents, which are in $10^{-3}s$.  These runtimes are low as the sliced UOT methods are fast to compute, and the documents contain in average 116 words for the BBC dataset and 1491 words for the Goodreads dataset (see Table 2 in Appendix C for a summary of the statistics of the datasets).
>
>
> - "In my understanding the hyperparameters are tuned differently for UOT and the sliced variant"
>
> We applied the same methodology (grid search) for hyperparameter tuning, but since the runtimes are significantly longer for UOT than for the sliced variants, we could only evaluate the performance of UOT on a coarser grid.
>
> Therefore, the only difference lies in the granularity of the grid search. Based on our empirical observations, this does not compromise the fairness of the comparison, as the grid was still chosen to cover a wide range of plausible values, ensuring that the (sub-)optimal performance of UOT is accurately captured within a reasonable range. We will clarify this in the paper.
>
>
> - "also I think that the cross validation based $\rho$ should be reported in the main text"
>
> We agree: we added the results for the cross validation in the main text.
>
>
>
> - "Theorem 3.11 seems to be a bit more involved than most of the other results. It would be good if some proof intuition can be provided and if the authors could clarify what parts of the proof are identical to prior work and where the proof is adapted to the new setting."
>
> Thank you for reviewing the appendix. Following your suggestion, we have added in the main text the following elements to clarify the proof of Theorem 3.11. Our result is obtained by adapting the proof of Lemma 5.1.4 in [2] to our setting; specifically, by considering the dual of UOT/SUOT instead of the dual of OT/SOT. Most arguments in [2] adapt well to our setting, but establishing a Lipschitz condition on the integrand of the dual required a more technical approach: to this end, we proved Lemma A.12, which led to equations (58) and (62). This also resulted in a different constant value, denoted as $c(m(\alpha), m(\beta), \rho, R)$.
>
> - "What is shown below Eq. (21) in the proof of Theorem 3.4"
>
> Below eq. (21), we considered a slightly more general setting than Theorem 3.4 by proving the sample complexity for $\mathrm{SUOT}^{1/p}$ with $p \in [1, +\infty)$. We apologize for the confusion and added this result in a corollary.
>
>
>
> [1] Vacher, A., & Vialard, F. X. (2023, July). Semi-Dual Unbalanced Quadratic Optimal Transport: fast statistical rates and convergent algorithm. In International Conference on Machine Learning (pp. 34734-34758). PMLR.
>
> [2] Bonnotte, N. (2013). Unidimensional and evolution methods for optimal transportation (Doctoral dissertation, Université Paris Sud-Paris XI; Scuola normale superiore (Pise, Italie)).

---

> ### Comment · Reviewer_xwi2 · 2024-11-10
>
> Thank you for the careful response which addressed my (minor) concerns.
> I only have a couple of smaller remarks.
>
> - Theorem 3.4 looks much more rigorous now. The double use of $\xi$ should be avoided. It might be worth mentioning that a suitable class $\tilde{M}$ exists in the specific example that you give.
>
> - I would not write '+CV' in the table with results but just 'CV' because it is not really an additional feature but a more sound choice of model selection
>
> - Where exactly did you add the explanations for Theorem 3.11, are they coloured?

---

> > ### Author Response · Authors · 2024-11-12
> >
> > Thank you for carefully reviewing our update and for your additional comments.
> >
> > We have revised Theorem 3.4, using $\eta$ for the measures in the definition of $\tilde{M}$ instead of $\xi$. We also mentioned the existence of such class, removed the "+" in the Table, and added the explanations for Theorem 3.11 above the Theorem (end of page 7).

---

### Author Response · Authors · 2024-11-04
**General Response**

We thank the reviewers for their positive comments and their relevant feedback and questions. We have uploaded a revised version of our manuscript which incorporate their suggestions (in red).


We summarize the main changes in the new version below.
- We added a notation subsection in the "Background" section, as requested by Reviewer u6CG.
- We included a more detailed description of related works (in particular of the sliced partial optimal transport methods proposed in [1,2]), as requested by Reviewer u6CG.
- We added the rate $\kappa(n)$ for the particular case of the GHK setting derived in [3]
- We reported the results of the classification task using a cross validation on $\rho$, as requested by Reviewer xwi2.



[1] Bonneel, N., & Coeurjolly, D. (2019). Spot: sliced partial optimal transport. ACM Transactions on Graphics (TOG), 38(4), 1-13.

[2] Bai, Y., Schmitzer, B., Thorpe, M., & Kolouri, S. (2023). Sliced optimal partial transport. In Proceedings of the IEEE/CVF Conference on Computer Vision and Pattern Recognition (pp. 13681-13690).

[3] Adrien Vacher and François-Xavier Vialard. Stability of semi-dual unbalanced optimal transport: fast statistical rates and convergent algorithm. 2022.

---

### Decision · Action_Editor_1yxo · 2024-11-27

**Recommendation:** Accept as is

**Comment:**

All reviewers praise the paper after the (stimulating!) discussion phase, and the authors have successfully answered all queries and requests for clarification from the initial reviews. The paper effectively introduces interesting results in optimal transport and makes a fine contribution to the field of machine learning. A clear accept. Congratulations!

**Audience:**

I am confident this paper will be of interest to the TMLR audience.

**Claims And Evidence:**

Claims are all appropriately supported by evidence.